# Non-canonical start codons confer context-dependent advantages in carbohydrate utilization for commensal *E. coli* in the murine gut

Yassine Cherrak [1,3] ✉, Miguel Angel Salazar [1,3], Nicolas Näpflin [2], Lukas Malfertheiner [2], Mathias K.-M. Herzog[1], Christopher Schubert [1], Christian von Mering[2] & Wolf-Dietrich Hardt [1] ✉

Resource competition is a driver of gut microbiota composition. Bacteria can outcompete metabolically similar rivals through the limitation of shared growth-fuelling nutrients. The mechanisms underlying this remain unclear for bacteria with identical sets of metabolic genes. Here we analysed the lactose utilization operon in the murine commensal *Escherichia coli* 8178. Using in vitro and in vivo approaches, we showed that translation of the lactose utilization repressor gene *lacI* from its native non-canonical GTG start codon increases the basal expression of the lactose utilization cluster, enhancing adaptation to lactose consumption. Consequently, a strain carrying the wild type *lacI* GTG start codon outperformed the *lacI* ATG start codon mutant in the mouse intestine. This advantage was attenuated upon limiting host lactose intake through diet shift or altering the mutant frequency, emphasizing the context-dependent effect of a single nucleotide change on the bacterial fitness of a common member of the gut microbiota. Coupled with a genomic analysis highlighting the selection of non-ATG start codons in sugar utilization regulator genes across the *Enterobacteriaceae* family, our data exposed an unsuspected function of non-canonical start codons in metabolic competition.

The gastrointestinal tract harbours a dense and complex microbial community termed the gut microbiota. At the phylum level, the composition of the microbiota tends to remain relatively constant over time[1]. However, when examined at the strain diversity level, the gut proves to be a highly dynamic environment governed by cooperation and more predominantly exclusionary relationships between bacterial species[2]. Freter's nutrient niche theory posits that bacteria consuming identical nutrients with equal efficiency cannot coexist stably within the same ecological niche[3,4]. This principle holds particularly true in nutrient-limited environments such as the gastrointestinal tract, where a variety of bacterial adaptive mechanisms and competitive interactions have been selected and studied over the past years[5–7]. In this context, overlapping rivals can engage in a direct competition by using antagonistic strategies that favour the dominance of a single bacterium excelling in consuming shared resources[8]. This strategy, referred to as 'exploitation', is exemplified by certain commensal *Escherichia coli* and

[1]Institute of Microbiology, Department of Biology, ETH Zurich, Zurich, Switzerland. [2]Department of Molecular Life Sciences and Swiss Institute of Bioinformatics, University of Zurich, Zurich, Switzerland. [3]These authors contributed equally: Yassine Cherrak, Miguel Angel Salazar. ✉e-mail: ycherrak@biol.ethz.ch; hardt@micro.biol.ethz.ch

**Fig. 1 | Start codon distribution in metabolic gene regulators across *E. coli* genomes. a**, The start codon sequence of 32 carbohydrate metabolism regulator genes was analysed across 10,643 *E. coli* genomes. The distribution of the start codon sequence is indicated for each regulator. Non-ATG start codons were classified as 'other'. Of the predicted start codons in the *lacI* gene, 99% are non-canonical in *Escherichia* genomes. **b**, Among the different non-ATG combinations, GTG was exclusively found to initiate *lacI* translation in *E. coli*

strains. **c**, The *lacI* sequence of two *E. coli* strains (SMN152SH1 and K-12 MG1655) is provided to emphasize cases with multiple ambiguous start codons. In the top half of the boxes, the ribosome binding site (RBS), and the stop and predicted start codons are indicated. The prodigal output exposing the prediction score for each potential start codon, as well as information about the length of proteins and ribosome binding sites are depicted in the bottom half.

*Klebsiella* strains, and can directly affect pathogen fitness by depleting the environment of galactitol and beta-glycosides[9–11]. Furthermore, the probiotic *E. coli* Nissle can actively use oxygen and iron, thereby limiting the growth of pathogens such as *Salmonella enterica* serovar Typhimurium (*S.* Tm)[12,13]. Despite numerous examples of bacterial exploitation strategies documented in the literature, we still lack a mechanistic understanding of how niche exclusion occurs among metabolically related rivals. Specifically, it is unknown how bacteria with identical fitness-associated genes can metabolically compete, or what factors determine the outcome of exploitation interactions between strains whose growth relies on equivalent sets of nutrient utilization genes.

Protein translation is a complex cellular process shared by all living organisms. This multilayered phenomenon starts with an initiation phase orchestrated by a dynamic interplay between mRNAs, ribosomal subunits, the initiator tRNA and initiation factors. Together, they ensure the precise recognition of the start codon and delineate the open reading frame of the transcripts[14]. The nucleotide triplet ATG is the most predominant start codon in all kingdoms of life and is considered as the universal initiation codon in every known genetic code. The strong evolutionary pressure to use ATG as start codon relates to its intrinsic properties leading to the highest ribosome binding strength and translation initiation rate compared with other codon sequences[15,16]. Notably, recent advances in proteomic and ribosome profiling techniques highlighted thousands of eukaryotic, archaeal and prokaryotic ORFs harbouring non-ATG start codons[17–20]. In bacteria, GTG and TTG are respectively found in ~12% and ~8%, which makes them

some of the most common alternative start codons[18]. Mechanistically, and despite recruiting the same *N*-formyl methionyl-tRNA[21,22], the use of GTG or TTG start codons leads to a suboptimal translation efficiency reflected by an 8- to 12-fold decrease in the expression rate compared with the use of ATG[15,23]. Considering the significant effect of a single nucleotide change in the start codon on the gene expression level, what benefit organisms get from such a suboptimal translation system and how this is reflected in vivo remain unclear.

Here we aimed to explore the role of the non-canonical start codons in bacterial metabolic adaptation within the context of gut colonization.

## Results

### Non-ATG start codons in *E. coli* carbohydrate regulators

Given their direct function in metabolic adjustment, we focused our attention on the start codon sequences found in metabolic gene regulators. We analysed the occurrence of non-ATG start codons within characterized carbohydrate utilization regulator genes in 10,643 *E. coli* genomes (Extended Data Fig. 1a, Supplementary Table 1 and Methods)[24]. Out of the 32 regulator genes subjected to analysis, 13 featured non-ATG start codons (Fig. 1a). Notably, *rbsR*, *murR*, *mltR*, *malT*, *gntR*, *gatR*, *fucR*, *araC* and *alsR* showed a dual presence of both ATG and non-ATG start codons, arranged in distinct patterns. By contrast, *lacI*, *rhaR*, *mlc* and *cra* carry almost exclusively a non-ATG start codon, suggesting a strong evolutionary preference for unconventional start codons in these metabolic gene regulators. To explore that phenomenon

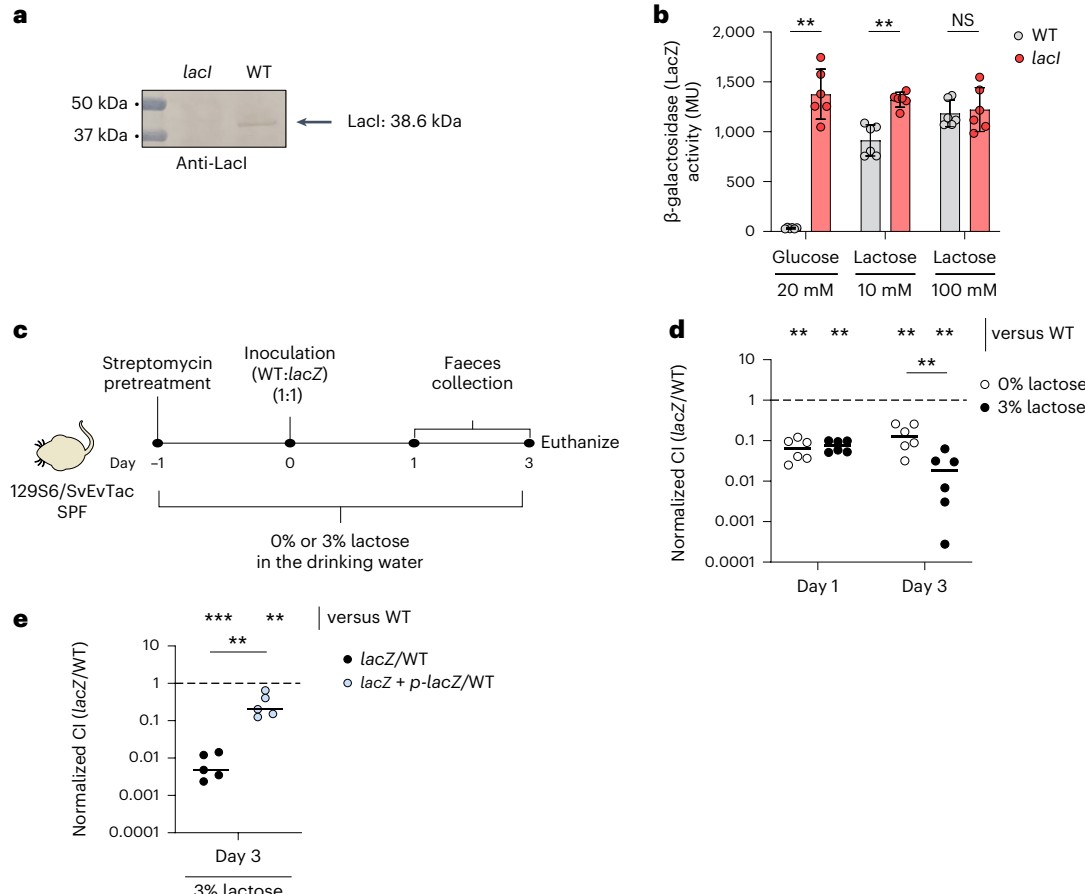

**Fig. 2 | Lactose metabolism facilitates *E. coli* 8178 growth in the mouse intestine. a**, Detection of LacI expression in *E. coli* 8178. Soluble extracts of *E. coli* 8178 strains (WT and *lacI* mutant) were subjected to a 12.5% acrylamide SDS-PAGE analysis followed by an immunodetection step using an anti-LacI antibody. Detected LacI is indicated on the right. The molecular weight marker (in kDa) is indicated on the left. The western blot was performed independently twice, and a representative experiment is shown. **b**, β-galactosidase activity assays. WT and *lacI*-deficient *E. coli* 8178 strains were grown for 8 h on a minimal medium supplemented with glucose (20 mM) or lactose (10 mM or 100 mM). Bacterial cells were collected and subjected to a β-galactosidase activity assay. The β-galactosidase activity is expressed in Miller units (MU). Activities are the mean of six biological replicates, and the error bars indicate standard deviation. **c**, Experimental scheme. Streptomycin-pretreated SPF 129S6/SvEvTac mice were inoculated with an equal mixture of the *E. coli* 8178 WT and *lacZ*-deficient strains. The drinking water was supplemented (3% w/v) or not (0%) with lactose.

**d**, Competitive experiment. The WT and *lacZ*-deficient strains were used to inoculate streptomycin-pretreated 129S6/SvEvTac mice supplemented (3%) or not (0%) with lactose in the drinking water. The WT and *lacZ* counts were determined by selective plating. The normalized C.I is calculated as the ratio between the mutant (*lacZ*) and WT divided by the ratio of both strains in the inoculum. **e**, Complementation experiments. The *lacZ* mutant was transformed with a plasmid constitutively expressing the *lacZ* gene. The fitness of the resulting strain (*lacZ + p-lacZ*) was analysed through a competitive experiment in streptomycin-pretreated 129S6/SvEvTac mice supplemented with lactose (3%) in the drinking water. **d,e**, The *x* axis represents the time post-inoculation (in days). The bars represent the median, and the dotted lines represent the C.I expected for a fitness-neutral mutation. The results from at least two independent replicates are shown. **b,d,e**, Two-tailed Mann–Whitney *U* tests were used to compare two groups in each panel. NS, not significant ($P \geq 0.05$); **$P < 0.01$; ***$P < 0.001$.

and considering the established literature on the lactose operon, we focused our effort on studying the function of the non-canonical start codon in the *E. coli lacI* gene. *lacI* codes for a negative regulator tightly regulating the lactose utilisation operon *lacZYA* by binding to its promoter and preventing gene transcription in the absence of lactose[25,26]. Upon closer examination of the *lacI* start codon sequences encoded in *E. coli* strains, we found that more than 99% are predicted to be GTG (Fig. 1b). Specifically, the vast majority of analysed *E. coli* strains harbour multiple potential GTG start codons, located in frame and within ten nucleotides (Fig. 1c). To our surprise, an additional scenario arose in which ATG and GTG codons were in close vicinity (Fig. 1c). For the analysis of these ambiguous cases, we selected the start codons with the highest prediction scores and calculated their relative fractions (Methods). Genomes with a predicted *lacI* ATG start codon were spread throughout the phylogenetic diversity within a subset of *E. coli* strains (Extended Data Fig. 1b), suggesting that the preference for the GTG or ATG start codon in *lacI* is attributable to selection and not genetic drift.

This prompted us to mechanistically investigate the benefit associated with the *lacI* GTG start codon in the natural habitat of a gut commensal *E. coli* strain. For this purpose, we selected the recently isolated *E. coli* 8178 mouse strain as our model organism[27].

### Functional *lacZYA* promotes *E. coli* 8178 growth in vivo

*E. coli* 8178 is a murine commensal strain (NCBI reference sequence: NZ_JAEFCJ010000001.1)[28] that encodes for a complete and functional *lacZYA* operon (Extended Data Fig. 2a,b). The *lacZYA* operon repressor *lacI* (NCBI reference sequence: WP_000805859) harbours an ambiguous and rare ATGTG sequence composed of both the ATG and GTG start sites, out of frame (Extended Data Fig. 2c). Translation from the commonly universal ATG codon leads to immediate termination (Extended Data Fig. 2c). Conversely, translation from the GTG reading frame generates a full and functional protein that actively represses the *lacZYA* operon in the absence of lactose, with this repression relieved by increasing concentrations of lactose (Fig. 2a,b and

Extended Data Fig. 2c). To study the advantage conferred by lactose metabolism in vivo, we used 129S6/SvEvTac mice harbouring a complex specific pathogen-free (SPF) microbiota. The microbiota of these mice is devoid of endogenous commensal *E. coli*, allowing us to monitor *E. coli* 8178 gut colonization properties without related endogenous strains[29]. Mice were treated with a single dose of streptomycin (25 mg, by gavage) 1 day before inoculation to temporally disrupt the gut microbiota and facilitate *E. coli* 8178 colonization (Fig. 2c)[30]. Inoculation with a 1:1 mixture of the wild type (WT) and *lacZ*-deficient *E. coli* 8178 strains followed by bacterial enumeration from faecal samples revealed a growth advantage of the WT over the *lacZ* mutant (normalized competitive index (C.I) < 1) (Fig. 2d). To further probe the role of lactose metabolism in the competitive growth of *E. coli* 8178, we inoculated an additional group of mice that were provided drinking water supplemented with lactose. For this purpose, we opted for a concentration of 3% (w/v) lactose, which falls within the range of lactose concentrations found in human (6–8%), cow (5%) and mouse (3–4%) milk[31–33]. While lactose supplementation did not impact the total load of *E. coli* in the gut (Extended Data Fig. 2d), the increased lactose intake aggravated the growth disadvantage of the *lacZ*-deficient strain as reflected by a reduced C.I (Fig. 2d). Expression of *lacZ* in *trans* partially restored the optimal growth of the *lacZ* mutant (Fig. 2e), which ruled out any indirect negative effect of the gut microbiota and indicated that lactose metabolism in *E. coli* 8178 plays an important role during intestinal colonization. Overall, these data show that the *lacZYA* operon of *E. coli* 8178 functions effectively, promoting gut luminal growth in the murine gut under these conditions.

## Microbiota and diet composition modulate *E. coli* 8178 *lacZ* fitness in vivo

Recent studies have presented inconclusive findings on the role of lactose in *E. coli* gut colonization[34–37]. To assess whether lactose-related fitness of *E. coli* 8178 was a general trait or context dependent, we analysed the *E. coli* 8178 lactose-dependent growth competitiveness in C57BL/6 mice. This commonly used mouse harbours a different *E. coli*-free SPF microbiota compared with 129S6/SvEvTac animals[29]. C57BL/6 mice were pretreated with streptomycin and inoculated with a 1:1 mixture of the WT and *lacZ* strains (Fig. 3a). Similar to our findings in 129S6/SvEvTac animals, *E. coli* 8178 reached a high level of gut colonization, with the *lacZ* mutant exhibiting a fitness defect (Fig. 3b (black symbols) and Extended Data Fig. 3a). However, the addition of lactose (3%) to the drinking water did not exacerbate the competitiveness of the WT over the *lacZ* strain in this mouse model (Fig. 3b). This was achieved at a higher lactose concentration (8%), showing a different response to lactose supplementation in C57BL/6 compared with 129S6/SvEvTac mice (Fig. 3c and Extended Data Fig. 3a). To explore the potential role of the gut microbiota in the lactose-dependent fitness defect of a *lacZ* mutant, we used germ-free C57BL/6 mice (Fig. 3a). Inoculated germ-free C57BL/6 mice showed a competitive defect of the *lacZ* mutant, which, in contrast to SPF colonized C57BL/6 animals, was significantly exacerbated upon 3% lactose supplementation (Fig. 3b (orange symbols) and Extended Data Fig. 3b). As opposed to SPF C57BL/6 mice, germ-free mice are inherently permissive to gut colonization and, as a result, were not subjected to antibiotic pretreatment. To verify that the difference in the *lacZ*-associated fitness between germ-free and antibiotic-pretreated SPF C57BL/6 mice arises from the microbiota rather than the antibiotic effect on the host, we evaluated the C.I of the *lacZ* mutant in streptomycin-pretreated germ-free mice. As expected, the C.I of the *lacZ* mutant remained unchanged between antibiotic-pretreated and non-pretreated germ-free animals when exposed to 3% lactose in the drinking water (Extended Data Fig. 3c,d). Combined, these observations show that the reliance of *E. coli* 8178 on the *lacZYA* operon varies in a microbiota-dependent manner.

In contrast to antibiotic-pretreated SPF 129S6/SvEvTac and C57BL/6 mice, the importance of lactose utilization by *E. coli* 8178 in germ-free animals became evident only at day 3 post-inoculation (Fig. 3b and Extended Data Fig. 3e). To maintain their status, C57BL/6 germ-free mice were housed in a controlled environment (germ-free facility) until the experiment began, at which point they were transferred to the SPF facility (day 0). Considering that the diet composition in our germ-free facility (germ-free facility diet) was slightly different from that in our SPF facility (standard diet; Methods), we wondered whether this would be the origin of the observed *E. coli* 8178 lactose-dependent fitness discrepancy between SPF and germ-free animals. We measured the level of lactose in both diets and found a significantly higher amount of lactose in SPF chow than in the germ-free diet (Fig. 3d). To confirm the diet effect on *E. coli* 8178 *lacZ* fitness, we inoculated 129S6/SvEvTac mice fed with the germ-free facility diet and found that the fitness advantage of the WT over the *lacZ* mutant strain was lost compared with mice that were kept on the standard diet (Fig. 3e). By contrast, we could restore the competitive advantage of the WT over the *lacZ* mutant strain by providing lactose-supplemented water to mice fed with germ-free-facility diet (Fig. 3e). These findings indicate that variation in lactose content between diets can lead to different *lacZ* fitness outcomes.

## *lacI* GTG start codon enhances lactose utilization in vitro

We next deciphered the fitness of the non-canonical GTG start codon in *lacI* and compared it with its more common counterpart, ATG. We selected the *lacI* start codon based on the sequence similarity to *E. coli* K-12 and the extensive body of functional and structural validation in the literature[38–40]. By targeting the historically established LacI start codon, we expected to alter the protein expression level without generating isoforms. The selected native GTG *lacI* start codon was subsequently replaced by ATG in *E. coli* 8178, resulting in the ATG$_{lacI}$ strain. In addition, we created a *lacI*-deficient mutant and a GTG$_{lacI}$ strain from the ATG$_{lacI}$ mutant by reverting the *lacI* ATG start codon into the original GTG, serving as a complementation step to further validate our functional studies. Western blot analyses on the native LacI protein indicated a higher abundance of LacI in the ATG$_{lacI}$ mutant compared with the WT and GTG$_{lacI}$ (Fig. 4a), confirming the effect of the start codon mutation on protein expression. To assess the fitness associated with this codon change, we performed in vitro growth assays. The WT, ATG$_{lacI}$, GTG$_{lacI}$ and *lacI*-deficient strains were initially cultured in a defined medium supplemented with glucose, followed by incubation under similar conditions using either glucose or lactose as the sole carbon source. No fitness defect was observed when glucose was added to the medium, indicating that none of these mutations has a global effect on bacterial growth (Extended Data Fig. 4). In the presence of lactose, all strains showed similar growth rates. However, we observed distinct lag phases after the switch from glucose to lactose, which were markedly shortened in the *lacI*, WT and complemented GTG$_{lacI}$ mutants, compared with the ATG$_{lacI}$ strain (Fig. 4b). This effect was further exacerbated when bacteria were incubated in a nutrient-rich lysogeny broth (LB) medium before transitioning to minimal media supplemented with lactose (Fig. 4c). Under these conditions, the *lacI* mutant adapted and grew the fastest. The WT and GTG$_{lacI}$ strains exhibited a significantly longer lag phase that was further extended in the ATG$_{lacI}$ strain. To mechanistically explain the observed variations in the lactose-adaptation phase, we designed a *gfp*-based transcriptional reporter system and measured the expression level of the *lacZYA* operon in the WT, GTG$_{lacI}$, ATG$_{lacI}$ and *lacI* strains. Bacterial cells were incubated with glycerol and isopropyl-β-D-thiogalactopyranoside (IPTG), an allolactose analogue. Among all tested strains, the *lacI* mutant showed a high and a consistently unchanged *gfp* signal across all tested IPTG concentrations, which illustrated the derepression of the *lacZYA* operon in this strain (Fig. 4d). At the highest IPTG concentration tested (500 µM), the WT, GTG$_{lacI}$ and ATG$_{lacI}$ strains showed reporter signals similar to the *lacI* background, indicating that the full derepression of the *lacZYA* operon is achieved under these conditions. Notably, the *gfp* signal resulting

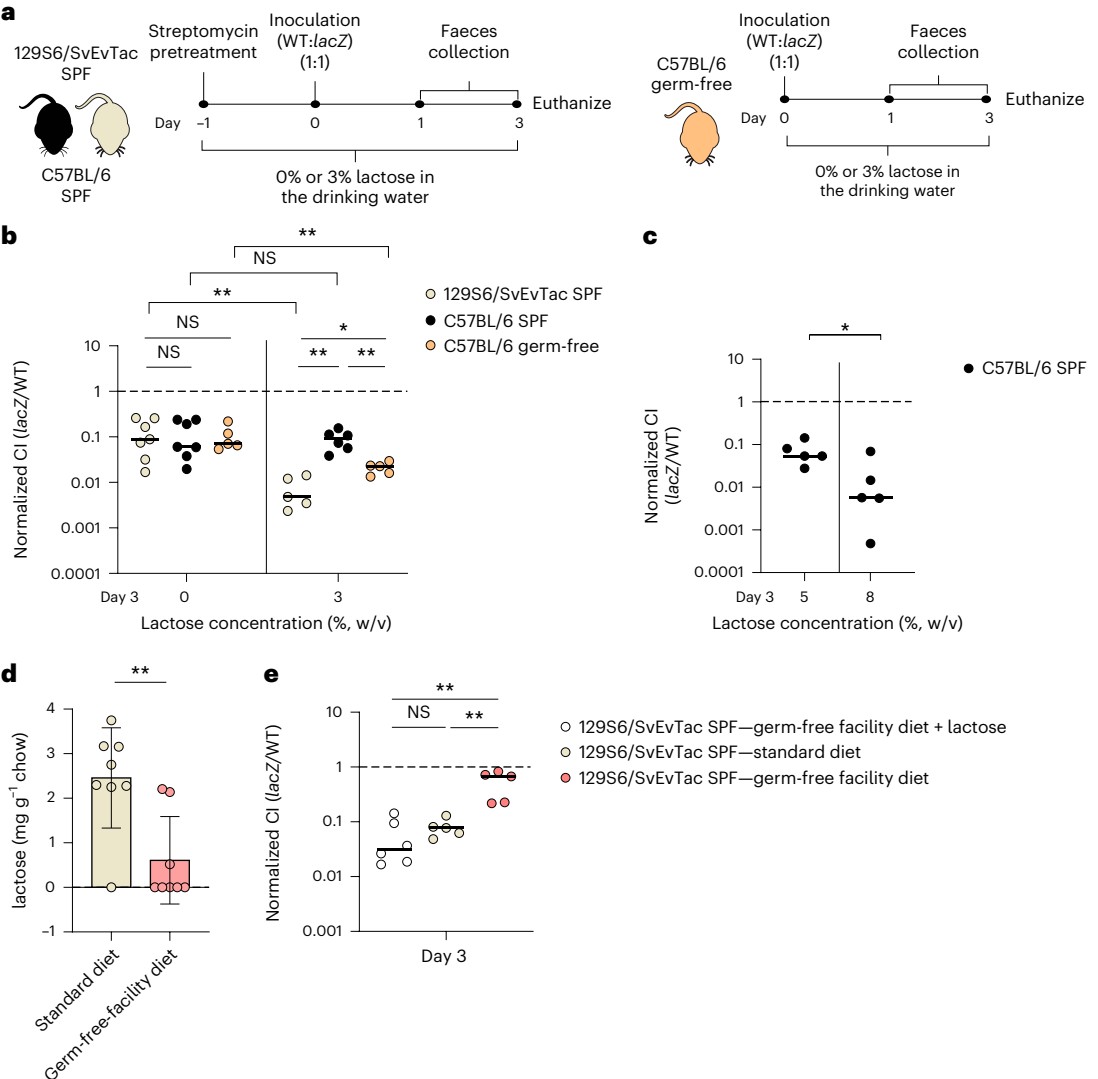

**Fig. 3 | Lactose-dependent *E. coli* 8178 growth in the murine gut is microbiota and diet dependent. a**, Experimental scheme. Streptomycin-pretreated SPF 129S6/SvEvTac or C57BL/6 mice were colonized with an equal mixture of the *E. coli* 8178 WT and *lacZ*-deficient strains. No antibiotic pretreatment was performed on C57BL/6 germ-free mice before inoculation. The drinking water was supplemented (3%) or not (0%) with lactose. **b**, Influence of the mouse microbiota on *E. coli* 8178 *lacZ* mutant fitness. The *E. coli* 8178 WT and *lacZ* mutant strains were used to colonize streptomycin-pretreated SPF 129S6/SvEvTac, streptomycin-pretreated SPF C57BL/6 or C57BL/6 germ-free mice. The counts of the WT and *lacZ* mutant strains were determined by selective plating from faecal samples and used to calculate the C.I at day 3 post-inoculation. **c**, Increased lactose supplementation exacerbates the *lacZ*-associated growth defect in streptomycin-pretreated SPF C57BL/6 mice.

The C.I of the *E. coli* 8178 *lacZ* strain in antibiotic-pretreated and inoculated SPF C57BL/6 mice is indicated at day 3 post-inoculation. The drinking water was supplemented with lactose at a concentration of 5% or 8%. **d**, Lactose proportion is heterogeneous across diets. SPF and germ-free facility chows ($n = 8$ pieces) were subjected to lactose measurement assays. The mean and error (standard deviation) are represented for each diet. **e**, *E. coli* 8178 lactose-dependent fitness is influenced by the diet composition. The C.I of the *lacZ* mutant at day 3 post-inoculation was assessed in streptomycin-pretreated SPF 129S6/SvEvTac mice fed with different mouse chows (standard or germ-free facility diets). **b,c,e**, The bars represent the median, and the dashed lines represent the CI expected for a fitness-neutral mutation. Two-tailed Mann–Whitney *U* tests were used to compare two groups in each panel. NS, not significant ($P \geq 0.05$); *$P < 0.05$; **$P < 0.01$. The results from at least two independent replicates are shown.

from the *lacZYA* operon transcription in the WT and GTG$_{lacI}$ strains was consistently higher than in the ATG$_{lacI}$ mutant at any tested intermediate IPTG concentration (1 μM, 5 μM, 20 μM, 100 μM). This trend held true even in the absence of IPTG, showing that the utilization of the GTG start codon in *lacI* was sufficient to enhance the sensitivity and basal expression of the *lacZYA* operon (Fig. 4d). These observations indicated that the decreased LacI protein level caused by the GTG start codon alleviated the repression of the *lacZYA* operon, allowing cells to adapt faster to lactose consumption. Whole genome sequencing analysis verified that the observed phenotypes were indeed linked to the mutation of the *lacI* start codon in the ATG$_{lacI}$ strain, ruling out the involvement of other secondary mutations.

## *lacI* GTG start codon increases *E. coli* 8178 fitness in vivo

Given the importance of lactose in supporting *E. coli* 8178 growth in vivo, we speculated that the utilization of a non-canonical *lacI* start codon would similarly enhance *E. coli* 8178 fitness in the murine gut. To test this hypothesis, we colonized streptomycin-pretreated 129S6/SvEvTac mice with an equal mixture of the WT and ATG$_{lacI}$ strains and assessed their C.I. When mice were fed a standard diet supplemented with 3% lactose in the drinking water, we found that the WT strain outperformed the isogenic ATG$_{lacI}$ mutant by approximately 30-fold within 3 days (Fig. 5a). Notably, the ATG$_{lacI}$ mutant did not go extinct but rather stably coexisted (albeit at a lower proportion) with the WT strain when the experiment was prolonged to 2 weeks (Extended Data Fig. 5a,b).

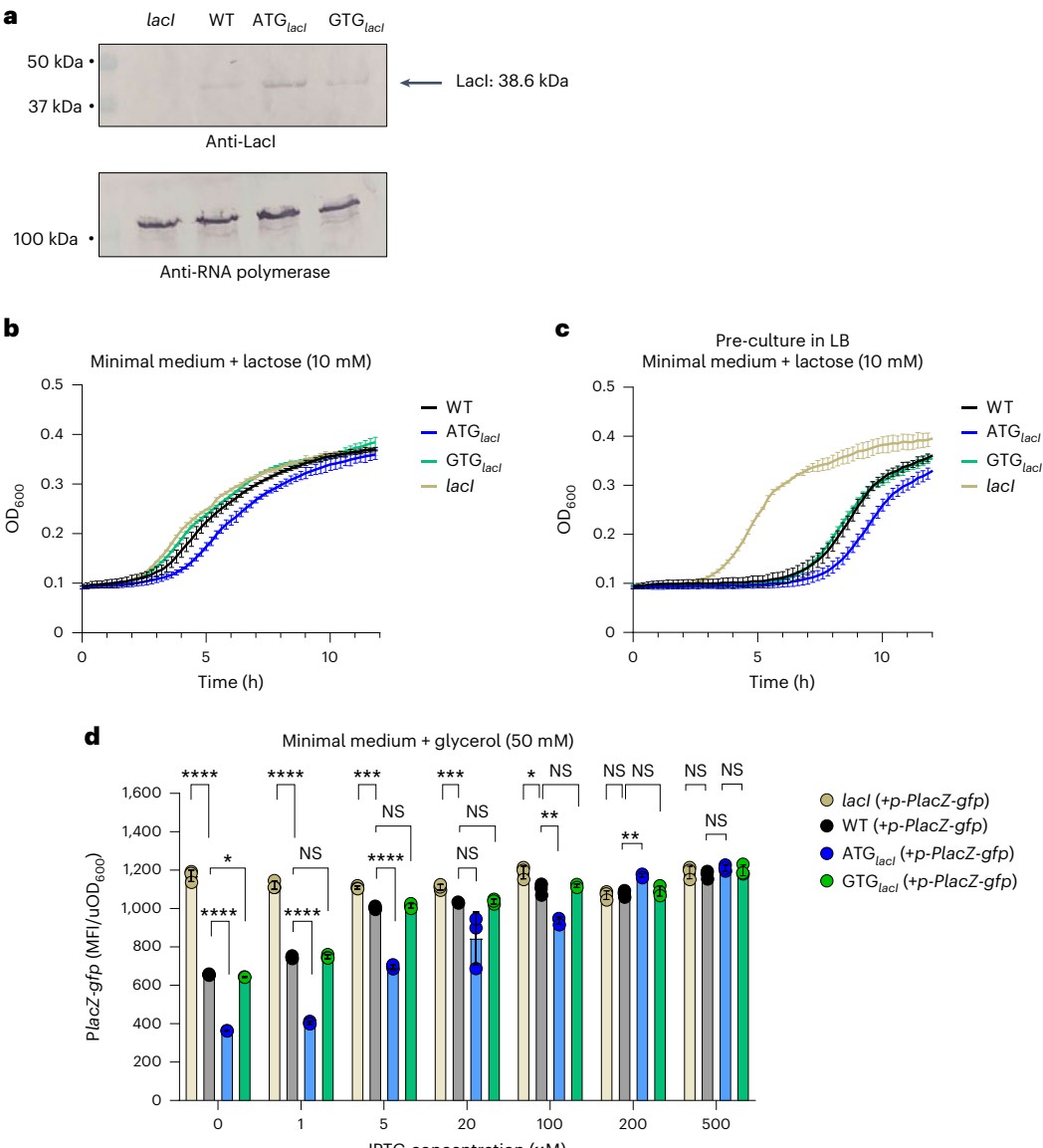

**Fig. 4 | The *lacI* non-canonical GTG start codon fosters lactose utilization in vitro. a**, Effect of the start codon sequence on the LacI expression level. Soluble extracts of *E. coli* 8178 strains (WT, ATG*lacI*, GTG*lacI* and *lacI* mutant) were subjected to a 12.5% acrylamide sodium dodecyl-sulfate polyacrylamide gel electrophoresis (SDS-PAGE) analysis followed by an immunodetection step using an antibody specific against LacI (upper gel) or the subunit-β of the RNA polymerase (lower gel). The detected LacI protein is indicated on the right. The molecular weight marker (in kDa) is added on the left. The western blot was performed independently twice and a representative experiment is shown. **b**,**c**, In vitro growth assays. The indicated strains were individually grown on a minimal medium supplemented with lactose (10 mM). Bacterial growth was quantified by measuring the optical density (OD) at 600 nm (OD$_{600}$). Strains were previously incubated overnight in a minimal medium supplemented

with glucose (20 mM) (**b**) or in a rich LB medium (**c**). The solid coloured lines represent the means of biological triplicates, and the error bars represent standard deviation. **d**, The *lacZ* expression level is influenced by the *lacI* start codon sequence. The *lacI*, WT, ATG*lacI* and GTG*lacI* strains were transformed with the *p-PlacZ-gfp* reporter plasmid. The strains were incubated in a minimal medium supplemented with glycerol (50 mM) in the presence of the indicated IPTG concentrations. The *gfp* signal resulting from *lacZ* expression at mid-exponential phase represents the mean of a biological triplicate and is expressed as the mean fluorescence intensity (MFI) normalized by the bacterial density (optical density unit: uOD$_{600}$). The error bars represent standard deviation. Student *t*-tests were used to compare two groups in each panel. NS, not significant ($P \geq 0.05$); *$P < 0.05$; **$P < 0.01$; ***$P < 0.001$; ****$P < 0.0001$.

By contrast, the complemented GTG*lacI* mutant had a similar competitive fitness as the WT strain (Fig. 5a and Extended Data Fig. 5a,b). It is worth noting that the load of the GTG*lacI* strain tends to be slightly lower than that of the WT. This can be attributed to the growth defect associated with antibiotic-based selection of the mutant, whereas the WT count is deducted from the total load on antibiotic-free plates. Altogether, we found that a single nucleotide change in the *lacI* start codon is sufficient to alter the fitness of isogenic *E. coli* 8178 strains carrying identical sets of metabolic genes in vivo. Strikingly, the fitness

advantage of the WT over the ATG*lacI* strain diminished in mice that were kept on germ-free chow, showing that the increased competitiveness conferred by the *lacI* GTG start codon is contextual and depends on the presence of lactose in the environment (Fig. 5a and Extended Data Fig. 5a,b). We next explored how the initial mutant frequency influences the impact of the *lacI* ATG start codon mutation. For this aim, we inoculated streptomycin-pretreated 129S6/SvEvTac mice with a mixture of the WT and ATG*lacI* strains at varying ratios and evaluated the mutant fitness over a 2-week experiment course in the presence of

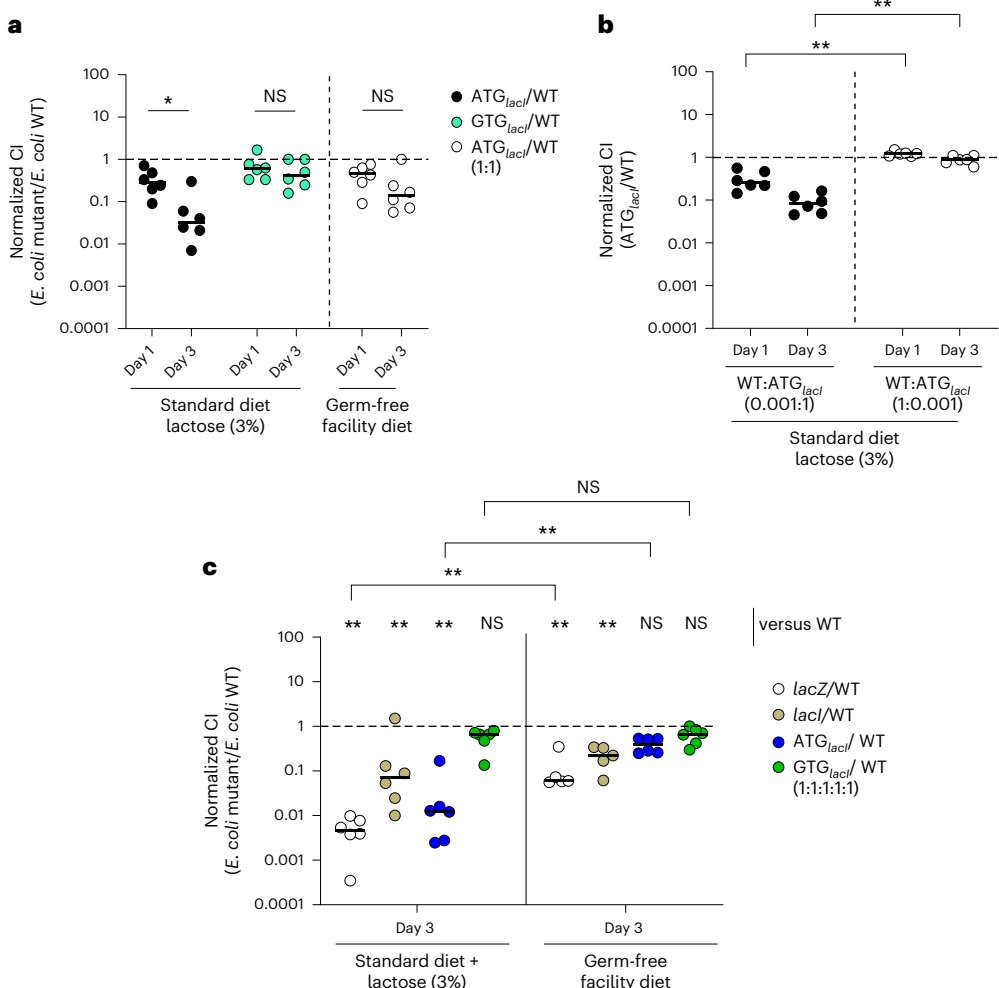

**Fig. 5 | A single nucleotide change within the *lacI* start codon impacts *E. coli* 8178 fitness in the gut. a**, The C.I of the ATG$_{lacI}$ strain (ATG$_{lacI}$/WT) or GTG$_{lacI}$ mutant (GTG$_{lacI}$/WT) in the streptomycin-pretreated 129S6/SvEvTac mouse model is indicated. Mice were kept under different regimes (standard diet coupled with lactose supplementation in the drinking water or germ-free facility diet without supplemented lactose) to analyse the effect of lactose in the bacterial fitness. **b**, Mice were inoculated with the WT and ATG$_{lacI}$ strains at different ratios (indicated). The C.I of the ATG$_{lacI}$ mutant to the WT strain is indicated for each dilution tested. Mice were kept under a standard diet supplemented with lactose (3%) in the drinking water. **c**, Barcoded *E. coli* 8178 strains carrying different genetic backgrounds (WT, *lacI*, *lacZ* ATG$_{lacI}$, GTG$_{lacI}$)

were equally mixed and used to orally inoculate streptomycin-pretreated SPF 129S6/SvEvTac mice kept under a standard diet supplemented with lactose 3% or on a germ-free facility diet. The different tagged *E. coli* 8178 populations were quantified by qPCR from faecal samples. The fitness of each single mutant is represented as a CI. The *x* axis represents the C.I of each indicated mutant compared with that of the WT strain. **a**–**c**, The bars represent the median, and the dotted lines represent the C.I expected for a fitness-neutral mutation. The results from at least two independent replicates are shown. Two-tailed Mann–Whitney *U* tests was used to compare two groups in each panel. NS, not significant ($P \geq 0.05$); *$P < 0.05$; **$P < 0.01$.

lactose (3%) in the drinking water. When placed in excess, the ATG$_{lacI}$ mutant was progressively outnumbered by the WT strain, highlighting a fitness disadvantage conferred by the ATG *lacI* start codon even at a high frequency (Fig. 5b and Extended Data Fig. 5c,d). Interestingly, the fitness of the ATG$_{lacI}$ mutant remained unchanged when diluted a thousand-fold with the WT strain. This indicated that this mutation, at a lower frequency, appears to be no longer detrimental under our conditions (Fig. 5b and Extended Data Fig. 5c,d).

Finally, given the slight alleviation of the *lacZYA* repression in the WT strain compared with the ATG$_{lacI}$ mutant, we reasoned that *E. coli* 8178 fitness in vivo would, similarly to our in vitro observation, be negatively correlated with the degree of *lacZYA* repression. To explore the potential correlation between the bacterial growth fitness and the level of *lacZYA* repression, we inoculated 129S6/SvEvTac mice with a 1:1:1:1:1 mixture of barcoded versions of the *lacI*-deficient strain, WT, GTG$_{lacI}$, ATG$_{lacI}$ and the *lacZ* mutant as an internal control. The relative abundance of each mutant was assessed through the detection

of strain-specific fitness-neutral 40 bp barcodes[41] (Extended Data Fig. 5e). In the lactose-supplemented group, the normalized abundance of the *lacZ*, ATG$_{lacI}$ and *lacI* strains at 3 days post-inoculation was below 1, revealing a growth defect compared with the WT strain under these conditions (Fig. 5c). In contrast to our in vitro results, these data indicate that after 3 days of colonization, the complete derepression of the *lacZYA* operon is as detrimental as the elevated LacI expression resulting from the restoration of the canonical ATG start codon in the ATG$_{lacI}$ strain. The fitness of the *lacI* and ATG$_{lacI}$ mutants stabilized throughout the 2 week duration of the experiment, highlighting the coexistence of multiple lactose-consuming variants, with the WT prevailing under our conditions. Contrasting with the lactose-supplemented group, the fitness defect of the ATG$_{lacI}$ strain was less pronounced under lactose-limited conditions (Fig. 5c and Extended Data Fig. 5f). Taken together, these observations highlight the frequency- and context-dependent metabolic benefit attributed to the *lacI* start codon in *E. coli* 8178.

**a**

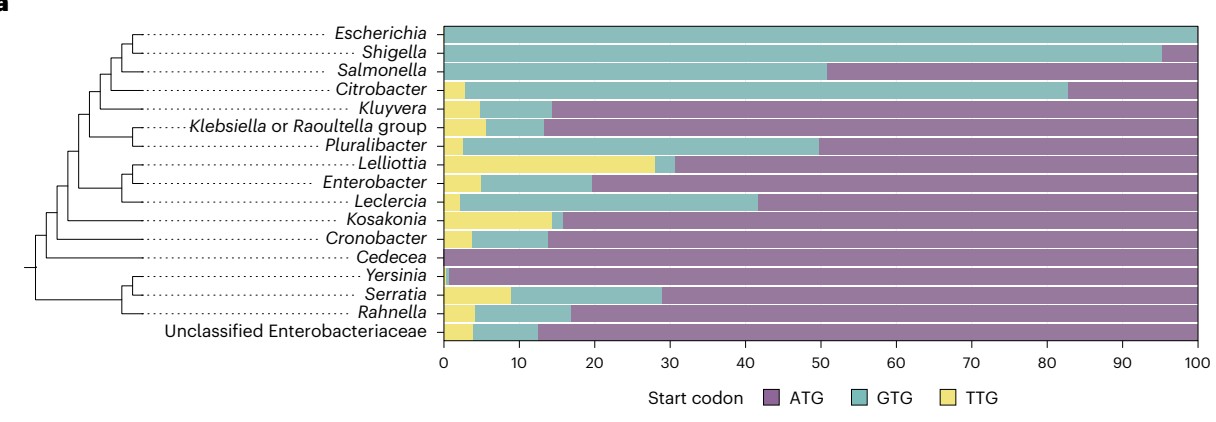

**b**

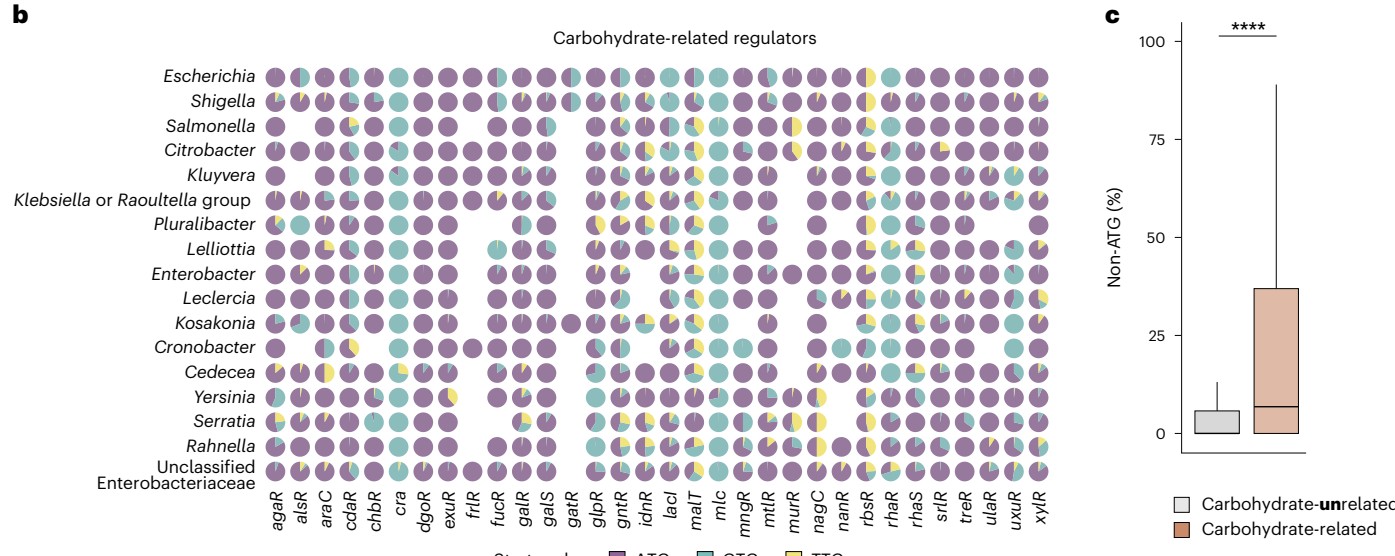

**Fig. 6 | Distribution of metabolic regulator start codons across the *Enterobacteriaceae* family. a**, Non-ATG start codons are conserved among several genera of the *Enterobacteriaceae* family. *Shigella* and *Citrobacter* genomes have a similar level of *lacI* GTG start codon conservation as *Escherichia*. The bacterial families were grouped according to their phylogenetic lineage. **b**, Alternative start codons are found in multiple carbohydrate-related regulators within the *Enterobacteriaceae* family. The data shown in this figure rely on existing annotations, which can be ambiguous or missing. The conservation level is represented by a circle-filled colour code that varies depending on the regulator and genus. **c**, Percentage of non-ATG start codons in carbohydrate-related ($n = 32$) and carbohydrate-unrelated ($n = 29$) transcriptional regulators in *Enterobacteriaceae*. A two-tailed Mann–Whitney *U* test was used to compare the two groups (****$P = 1.5 \times 10^{-11}$). Data are shown as median (black horizontal line) and 25% and 75% percentiles (hinges). Whiskers extend from the hinges to the maxima and minima, no further than 1.5× distance of the interquartile range.

## Non-ATG start codons in *Enterobacteriaceae* carbohydrate regulators

Building upon our functional studies, we next analysed the sequence of the *lacI* start codons across 16 genera within the *Enterobacteriaceae* family to investigate the conservation of similar mutations in related species (Fig. 6a). Despite the conventional characterization of *Salmonella* species as 'Lac-negative', all *Salmonella* species selected for this analysis were 'Lac-positive'. Unusual *lacI* start codons were heterogeneously distributed across nearly all the selected genera. A closer examination at the sequence level indicated that, in most instances, the predominant non-ATG start codon found within the *Enterobacteriaceae lacI* genes was GTG (Fig. 6a). Aside from *Escherichia*, more than 80% of the *lacI* start codons are predicted to be GTG in the *Shigella* and *Citrobacter* genomes analysed, suggesting a benefit associated with the *lacI* GTG start codon that can be extended – in a strain-dependent manner – to other bacterial genera. Examination of the different genome phylogenies revealed that the distribution of the different start codons within the different lineages is not monophyletic. This observation indicated that appearance and distribution of non-canonical start codons in *lacI* resulted from multiple and independent evolutionary events, further emphasizing the notion of contextual fitness associated with the choice of start codon sequences. We expanded our bioinformatic analysis to include genes coding for the main transcriptional regulators of carbohydrate utilization (Supplementary Table 1 and Fig. 6b). Upon scrutinizing the occurrence of non-ATG start codons at the *Enterobacteriaceae* family level, we found that all examined metabolic regulator genes showed alternative start codons in at least a subset of strains within specific genera. Notably, non-canonical start codons were further enriched in *cra* and *mlc*, starting exclusively with a GTG start codon in almost all the genera analysed (Fig. 6b). Given the potential benefit associated with non-canonical start codons in carbohydrate metabolism, we next wondered whether such distribution would show a similar pattern among carbohydrate-unrelated regulators. To explore this, we carried out an analysis of the start codons in 29 additional regulators that do not appear to be exclusively involved in carbohydrate metabolism (Supplementary Table 2). The overall

proportion of non-canonical start codons was significantly lower in carbohydrate-unrelated compared with carbohydrate-related regulators (Fig. 6c and Extended Data Fig. 6a). This trend persisted across positive, negative and bidirectional regulators (Extended Data Fig. 6b). Taken together, these observations suggest that non-canonical start codons have a wider range of functional advantages beyond lactose utilization in *E. coli* 8178, extending to carbohydrate metabolic pathways within the *Enterobacteriaceae* family.

## Discussion

Non-ATG codons are widespread across genomes, and the evolutionary pressures driving their selection remain debated. Translation from unusual and differently placed start codons in eukaryotic genomes has been proposed to contribute to the generation of protein isoforms[42]. In bacteria, it is yet to be determined whether the presence of multiple alternative start codons in close proximity is associated with the generation of protein variants. Recent works in *E. coli* proposed stress-related functions to non-ATG start codons owing to their high prevalence in genes ensuring essential processes[43]. However, this study offered only a partial view of the function of non-canonical start codons, as many of them are also found in non-essential genes[44]. Our computational analysis highlighted a significant conservation level of various non-ATG start codons among known carbohydrate utilization regulators across the *Escherichia* genus and the broader *Enterobacteriaceae* family (Figs. 1a and 6b). Yet, upon closer examination, we also observed cases in which ATG start codons were more prevalent or even exclusively used in certain strains and regulators. From this distribution, it became evident to us that the choice between ATG and non-ATG start codons is influenced by distinct evolutionary pressures determining the preference for the universal or atypical start codons. This specific evolutionary selection presents an intriguing avenue for future research. Focusing on the lactose utilization operon, we found that the GTG start codon was predominantly found in 99% of the analysed start codon sequences of the *E. coli* 8178 *lacI* gene (Fig. 1b). Lactose is primarily found in the mammalian neonate gut where it can benefit *E. coli* intestinal colonization early in life[45]. Supporting this hypothesis, mutations abrogating LacI production are selected for via mother-to-offspring microbiota transmission experiments[46]. Nevertheless, introduction of solid food can be detrimental for lactose-dependent long-term resilience of *E. coli* in adults. In line with this hypothesis, various studies revealed that *E. coli* does not rely on lactose utilization to colonize the mouse intestine[34,36]. By contrast, our competitive experiment data suggested not only that *E. coli* 8178 relies on lactose metabolism to efficiently colonize the mouse gut but also that such a phenotype was highly variable depending on the mouse microbiota and diet composition[37]. This conditional importance may result from lactose heterogeneity in diets, which, along with the variations in the model organism, could explain distinct outcomes observed in studies examining the importance of lactose in the intestinal colonization of *E. coli*.

To study the role of the *lacI* GTG start codon in *E. coli* 8178, we introduced a single point mutation (G→A) precisely at the known *lacI* start codon position. By doing so, we anticipated that both the WT and ATG$_{lacI}$ strains would produce identical LacI protein sequences but in variable amounts. Using a lactose-rich condition, we found that the growth of both the *lacI* and ATG$_{lacI}$ mutants was impeded compared with that of the WT. This indicated that reduction of the LacI protein level by the GTG start codon confers optimal growth parameters in this environment. Indeed, utilization of the canonical ATG codon in *lacI* (ATG$_{lacI}$) generates a higher repression level reducing the responsiveness of the lactose utilization operon when switching to a lactose-dependent growth phase (Fig. 4c,d). Conversely, complete derepression of the *lacZYA* operon (*lacI* mutant) becomes detrimental for bacterial growth as lactose is not the sole and primary carbon source in the adult murine gut[46–48]. The growth advantage of the WT was, albeit at a lesser extent, also visible in the lactose-limited germ-free food condition, implying that the expression of the lactose utilization operon in the WT strain was optimally balanced to prevent any fitness burden. From these observations, we concluded that selection of the GTG start codon in *lacI* results in an optimal metabolic fitness balancing between the full derepression (*lacI*) and stringent repression (ATG$_{lacI}$) of the *lacZYA* operon, allowing cells to transition faster to lactose-dependent growth (Extended Data Fig. 7). Finally, it is worth mentioning that the strong preference for the GTG start codon in *lacI*, and in certain other sugar utilization regulators, may indicate a potential function of non-canonical start codons that expands beyond translation processes. Further study is required to investigate this hypothesis thoroughly. The disadvantage associated with the *lacI* ATG start codon was primarily visible in the early stages of gut colonization (days 1–3 post-inoculation). Beyond this period, the ATG$_{lacI}$ strain did not go extinct but rather persisted at a stable proportion (Extended Data Fig. 5a). When coupled with the observed neutral fitness of the ATG$_{lacI}$ mutant at a lower proportion (Fig. 5b and Extended Data Fig. 5c), we concluded that the *lacI* ATG start codon is subjected to a frequency-dependent selection. This phenomenon, similarly observed for the galactitol and sorbitol metabolism[49,50], may provide a reasonable explanation for the relatively low frequency of the ATG *lacI* start codon observed in the analysed *E. coli* genomes (Fig. 1). Furthermore, the observed long-term coexistence of the WT and ATG$_{lacI}$ and *lacI* strains indicated the maintenance of a polymorphism, with the WT strain prevailing under our conditions. The simultaneous presence of different lactose metabolism variants, albeit at different proportions, concurs with recent findings highlighting the coexistence of different genotypes through negative frequency-dependent selection[50–53]. This suggests that, beyond gene regulation, the long-term preservation of different lactose metabolism variants may play an important role in the bacterial adaptation to the fluctuating conditions of the gastrointestinal tract.

Besides depicting the in vivo advantage associated with the *lacI* GTG start codon, our study provides a conceptual example of how bacteria can fine-tune their metabolic properties to be at an advantage over their closely related competitors. In a similar fashion, recent work on epidemic *Clostridioides difficile* ribotypes highlighted the presence of a single point mutation on the trehalose repressor, which abrogated the repression of the trehalose utilization operon and conferred an increased growth competitiveness under low trehalose concentrations[54]. In contrast to the trehalose utilization, which is completely derepressed in certain *C. difficile* strains, our experimental data show that an alternative strategy has been selected for the lactose operon in which repression is alleviated by an unusual start codon in *E. coli* 8178. These two examples illustrate how gene regulation can affect metabolic capacities and demonstrate the generalizability of such metabolic rewiring strategy. In that perspective, our bioinformatic analysis pinpointed several metabolic regulators in which non-ATG codons were enriched. Some of them, such as UxuR, GatR, FucR and RhaR, play an important function in carbohydrate metabolism and foster *S.* Tm, *Citrobacter rodentium* and *E. coli* growth in the gut[10,55–59]. To a greater extent, the *cra* and *mlc* genes almost exclusively adopted GTG start codons, suggesting a role of non-canonical start codons extending to other metabolic pathways within the Enterobacteriaceae family level.

## Methods

### Strain, media and chemicals

All strains, plasmids and oligonucleotides used in this study are listed in Supplementary Tables 3–5. The physiological function of the non-canonical LacI start codon in intestinal colonization was assessed using the recently isolated murine gut commensal *E. coli* 8178 strain[27]. Bacterial strains were routinely grown in LB supplemented or not with bactoagar (1%). Plasmid maintenance and mutant selections were performed via appropriate antibiotic addition: streptomycin (100 µg ml⁻¹), ampicillin (100 µg ml⁻¹), kanamycin (50 µg ml⁻¹) and chloramphenicol (30 µg ml⁻¹). Insertion of fitness-neutral barcodes and deletion of the

*lacZ* gene into *E. coli* 8178 were achieved using a modified version of the lambda red recombinase-dependent one-step inactivation procedure[60]. Briefly, the antibiotic resistance cassette (kanamycin for gene deletion, ampicillin for barcode insertion) was PCR amplified using primer pairs carrying a 50-nucleotide extension homologous to the adjacent targeted region. Mutants were obtained following the electroporation of the PCR product into *E. coli* 8178 cells expressing the lambda red recombinase from the pSIM5 plasmid and incubated on selective media[61]. Gene deletion was confirmed by colony PCR. The *lacI* point mutation at the native locus was conducted by allelic replacement using the pKO3 suicide vector[62]. Briefly, the *lacI* region including the start codon and 500 base pairs upstream and downstream was cloned into the pKO3 plasmid that carried the *sacB* gene for counter selection on sucrose-containing media. Following the mutagenesis of the start codon by PCR amplification, the plasmid was introduced into *E. coli* 8178 and we selected on ampicillin plates for mutants with a single-crossover integration of the plasmid into the chromosome. A sucrose-based counter selection, colony PCR and sequencing were applied to identify clones carrying the desired marker-less nucleotide exchange at the *lacI* start codon.

### Animals
Male and female 8- to 12-week-old mice were used in this study and randomly assigned to experimental groups. 129S6/SvEvTac (Jackson Laboratory) and C57BL/6 (Jackson Laboratory) mice were held under SPF conditions in individually ventilated cages at the EPIC mouse facility of ETH Zurich (light–dark cycle 12 h:12 h, room temperature $21 \pm 1 \,°C$, humidity $50 \pm 10\%$). C57BL/6 germ-free animals were bred in flexible film isolators under strict exclusion of microbial contamination at the isolator facility (EPIC). All animal experiments were reviewed and approved by Tierversuchskommission, Kantonales Veterinäramt Zürich under licence ZH158/2019, ZH108/2022 and ZH109/2022 complying with the cantonal and Swiss legislation. All mice were maintained on the mouse maintenance diet (SPF diet: Kliba Nafag number 3537). When indicated, mice were shifted to a different diet (germ-free diet: Kliba Nafag number 3302) 1 day before inoculation. For the conditions in which lactose was supplemented, lactose was diluted in 200 ml of drinking water and filtered using syringe filters (0.22 μm). The mouse drinking water was replaced with 3% (w/v) lactose 1 day before inoculation for SPF mice and on the inoculation day for germ-free mice. Lactose was maintained throughout the entire course of the experiment.

### Selection of *Enterobacteriaceae* and *E. coli* genomes
Genomes classified as *Enterobacteriaceae* or *Yersiniaceae* in the progenomes v3 database[63] were selected for the analysis. To ensure that the selected genomes are of high quality, CheckM2 (ref. 64) assembly statistics provided by S. Schmidt were used to remove genomes with completeness below 95% and contamination larger than 10%. From this initial selection, exceptionally small genomes compared with those annotated with the same species were excluded and genomes with excessively high numbers of contigs were removed as well. In total, 10,643 *E. coli* genomes were downloaded and used in the subsequent analysis. To cover the entire family of *Enterobacteriaceae*, large genera were randomly subset to maximally 100 genomes, while genera with fewer than 10 genomes were discarded. This resulted in a final dataset of 1,158 genomes distributed among 16 genera that remained to be analysed. The identifiers and taxonomic classification of all genomes are provided in Supplementary Table 6.

### Start codon analysis in *E. coli* carbohydrate regulators
A total of 10,643 *E. coli* genomes were downloaded from the progenomes3 database as described previously[63]. The sequences of established metabolic regulator genes from *E. coli* K-12 MG1665 were obtained from the Biocyc database[24] and converted to a blast database

for each regulatory gene with the command -makeblastdb. A homology search using nucleotide blast (blastn) was performed for each genome against every database, with parameters as follows: -max_target_seqs 100 -evalue 1e-5 -outfmt '6 qseqid qstart qend bitscore'[65]. The best hit and 75 nucleotides on either end were cut out for each genome. Prodigal version 2.6.3 (ref. 66) trained on the *E. coli* K-12 MG1665 genome was then used to identify potential start codons for each gene. In certain cases, the predicted start codon was ambiguous, meaning that two or more potential start codons were situated in frame and close proximity to each other. In this scenario, we selected all start codons whose scores were within 80% of the best scoring hit. In addition, the predicted genes must span at least 90% of the best-hit sequence length, as otherwise those fragmented proteins would most probably not be functional. If the best hit included a predicted ribosome binding site, only alternative start codons that were also predicted to contain a ribosome binding site were considered. We then calculated the relative fractions of potential start codons for each genome (for example, if a gene in one genome has three potential, high-scoring start codons located in close vicinity, ATG, GTG and GTG, they would count as 0.66 GTG and 0.33 ATG). Python 3.7.6 was used to compute the relative percentages of start codons over all genomes, and bar plots were generated using ggplot2 v3.3.5 in R 3.6.3 (ref. 67). For the phylogenetic tree showing the distribution of ATG and GTG start codons in *lacI* within *E. coli*, we chose the two genomes depicted in Fig. 1c and randomly selected an additional 25 genomes with a high-scoring ATG start codon, and 75 genomes without. The phylogenetic tree was constructed using *Salmonella* as an outgroup in gtdbtk v 2.1.0 (ref. 68) with the parameters 'gtdbtk de_novo_wf–genome_dir genomes/–outgroup_taxon g_Salmonella–bacteria–taxa_filter g_Escherichia–out_dir de_novo_output_2–cpus 24' and visualized in iTOL v 6.9.1 (ref. 69).

### Start codon distribution in *Enterobacteriaceae* carbohydrate regulators
Sequences of the selected 61 metabolic regulators (32 carbohydrate-related regulators, 29 carbohydrate-unrelated regulators; Supplementary Tables 1 and 2) were obtained by applying PIRATE v1.0.5 to the dataset of 1,158 *Enterobacteriaceae* genomes[70]. In short, PIRATE generates orthologous gene clusters at different sequence identity thresholds using GFF3 files as an input. GFF3 files were generated using Prodigal v2.6.3 (ref. 66). PIRATE was applied with default settings and an inflation parameter of 3 was set for the Markov Cluster Algorithm. Gene clusters containing sequences with annotations of interest, such as metabolic regulators listed in Supplementary Tables 1 and 2, were manually verified using blast (blastx, -max_target_seqs 10000 –evalue 1e-5) against the UniProtKB/Swiss-Prot database[71]. Manually curated sequences were then merged using the short name resulting in a set of 65,679 sequences for 61 metabolic regulators. Start codon distributions were calculated as described previously with Prodigal v2.6.3 trained on the genome from which the respective sequence originated. Regulators were categorized as negative or positive according to RegulonDB (ref. 72). Regulators influencing gene expression in both positive and negative ways were classified as bidirectional. Exceptions were made only for regulators exhibiting either a positive or a negative effect on a target gene, excluding self-regulation of the transcriptional regulator in this classification. Two-tailed Mann–Whitney *U* tests were used for statistical analysis of alternative start codon usage between carbohydrate-related and unrelated regulators. $P < 0.05$ was considered to indicate statistical significance.

### Mouse colonization experiments
Mice aged 8 weeks to 12 weeks were orally pretreated with streptomycin (25 mg) 24 h before inoculation. Due to the low intestinal colonization resistance properties of germ-free mice, no antibiotic pretreatment was achieved. *E. coli* cultures were grown on LB at 37 °C for 4 h and washed twice with a phosphate-buffered saline solution (PBS: 137 mM

$NaCl$, 2.7 mM $KCl$, 10 mM $Na_2HPO_4$ and 1.8 mM $KH_2PO$). Before colonization, *E. coli* strains were made streptomycin resistant by transferring the streptomycin-resistance-conferring plasmid pRSF1010 from *S. enterica* serovar Typhimurium SL1344 (refs. 30,73). Each mouse was orally given a single 50 µl dose containing ~$5.10^7$ colony-forming units (cfu) of an inoculum mixture composed of an equal ratio of the indicated strains. Faeces samples were collected 24 h and 72 h post-inoculation. Animals were euthanized by $CO_2$ asphyxiation at day 3 post-inoculation. Faecal samples were suspended in 1 ml PBS and homogenized using a TissueLyser (Qiagen). The bacterial load was determined by plating the suspension on MacConkey or LB agar supplemented or not with antibiotics. In competitive experiments with two strains (WT and mutant), the load of the mutant was directly determined on the antibiotic plate. By contrast, the load of the WT strain was calculated from the total amount of bacteria on an antibiotic-free plate ($cfu_{WT} = cfu_{total} - cfu_{mutant}$). The load of every single mutant strain was then normalized to the inoculum and used to calculate the normalized C.I to compare their fitness with that of the WT. The C.I of individual mutant in a competitive experiment was determined as the ratio between $cfu_{mutant}$ and $cfu_{WT}$ divided by the ratio of both strains in the inoculum.

### Fitness measurement of barcoded strains
Faecal *E. coli* cells were inoculated in 3 ml LB (37 °C, overnight) supplemented with ampicillin to select and enrich for living *E. coli* barcoded strains. The bacterial cells were pelleted and stored at −20 °C. DNA was extracted from thawed pellets using commercial kits (Qiagen Mini DNA) according to the manufacturer's instructions. The relative densities of the different barcodes were determined by real-time PCR quantification using tag-specific primers. The obtained ratio was multiplied by the number of cfu recovered from selective plating to calculate the absolute load of each tagged strain.

### Lactose measurement
Mice chow was suspended in PBS and homogenized using a TissueLyser (Qiagen). The lactose concentration in the homogenate was determined using a commercially available lactose assay kit (Sigma-Aldrich, MAK017) following the manufacturer's instructions.

### In vitro growth assays
Individual overnight culture of each strain was initiated 1 day before the experiment, either in LB or minimal media supplemented with glucose (50 mM) at 37 °C. The following day, the cultures were washed with PBS and diluted into the appropriate medium to reach a final concentration of 0.025 u$OD_{600}$ ml$^{-1}$. A volume of 200 µl of every single culture was distributed in a 96-well plate placed in an Infinite 200 Pro plate reader (Tecan) that automatically measured the $OD_{600}$ at 37 °C for 24 h.

### *lacZYA* reporter expression analysis
The WT, ATG$_{lacI}$ and GTG$_{lacI}$ strains were previously transformed with the *p-PlacZ-gfp* plasmid encoding for the *lacZ* promoter fused to the *gfp* reporter gene. The resulting strains were individually incubated overnight in minimal media supplemented with glycerol (50 mM). The day after, bacterial cultures were diluted into an identical and freshly prepared medium to reach a final bacterial concentration of 0.025 u$OD_{600}$ ml$^{-1}$. A volume of 200 µl of every single culture was distributed in a 96-well plate supplemented with increasing concentrations of IPTG. The bacterial growth and *gfp* signal were automatically recorded for 24 h at 37 °C by an Infinite 200 Pro plate reader (Tecan). The output was corrected by subtracting the autofluorescence signal and normalizing to the bacterial density (OD).

### β-galactosidase activity measurement
The activity of the LacZ β-galactosidase protein was assessed in a WT and *lacI* background. Briefly, strains were incubated in minimal media supplemented with either lactose or glucose as sole carbon source at 37 °C. After 8 h of incubation, cells were collected and lysed, and the β-galactosidase activity was measured in the presence of *o*-nitrophenyl β-ᴅ-galactopyranoside (Sigma-Aldrich) as described in ref. 74.

### Analysis of the intracellular LacI protein level
The WT, *lacI*-deficient, ATG$_{lacI}$ and GTG$_{lacI}$ strains were grown on LB until reaching the late-exponential phase. Cells were then pelleted, resuspended into a Laemmli loading buffer supplemented with 1 mM β-mercaptoethanol (Sigma-Aldrich) and heated for 10 min at 96 °C before analysis by sodium dodecyl-sulfate polyacrylamide gel electrophoresis (SDS-PAGE) and immunoblotting.

### SDS-PAGE and immunoblotting
SDS-PAGE was performed on Bio-Rad Mini Protean systems using standard protocols with homemade 12.5% polyacrylamide gels. For immunostaining, proteins were transferred onto a 0.2 µm nitrocellulose membrane that was then saturated with 5% milk and probed with primary antibodies (anti-LacI: Sigma-Aldrich-05-503-I, dilution 1:1,000; anti-RNA polymerase beta: ThermoFisher-MA125425, dilution 1:1,000). Mouse secondary antibody coupled with alkaline phosphatase (anti-mouse antibody: Sigma-Aldrich AP124A, dilution 1/1,000) was added and developed using a commercialized alkaline phosphatase substrate (Promega, S3841) containing 5-bromo-4-chloro-3-indoyl phosphate and nitroblue tetrazolium.

### Molecular biology
Custom oligonucleotides were synthetized using Microsynth and are listed in Supplementary Table 5. PCRs were performed using the Phusion DNA polymerase (Thermo Scientific), and PCR products were purified using the Nucleospin Gel and PCR clean-up mini kit (Macherey-Nagel). The complementation *p-lacZ* plasmid, suicide plasmids pT2214/pT2215 and *lacZ* reporter *p-PlacZ-gfp* were constructed by restriction-free cloning (E5510S Gibson Assembly, New England Biolabs) and verified by DNA sequencing (Microsynth). The QIAprep Spin Miniprep kit (E5510S; Qiagen) was used to extract plasmids from bacterial pellets. Real-time PCR was done using the FastStart Universal SYBR Green Master (Roche) according to the manufacturer's instructions.

### Whole genome sequencing analysis
The genomic DNA of the WT, ATG$_{lacI}$ and GTG$_{lacI}$ strains was extracted from overnight cultures using the NucleoBond (AXG20; Macherey-Nagel) kit. Library preparation and sequencing (ONT and Illumina) were performed by BMKGENE. Long reads were assembled using Flye 2.9.2, and the draft genome was polished with the Illumina reads using CLC Workbench 23.0.1 ('polish with reads 1.1'; Racon). The closed genomes were annotated with prokka 1.14.5. To find mutations in the ATG$_{lacI}$ and GTG$_{lacI}$ strains, short genomic reads were mapped ('map reads to reference 1.9'; CLC Workbench) to the closed genome of the parental strain. Basic Variant Detection 2.4 (CLC Workbench) was used to detect mutations.

### Statistical analysis
No statistical methods were used to predetermine sample sizes. The current sample sizes ($n \geq 5$) are similar to those reported in previous publications[75,76]. Data collection and analysis were not performed blind to the conditions of the experiments. No animals or data points were excluded from the analyses. Data distribution was assumed to be normal, but this was not formally tested. The statistical analysis and data graphical representation were done using GraphPad Prism 9.2.0 version for Windows (GraphPad Software; www.graphpad. com). When applicable, the unpaired Mann–Whitney *U* test (comparison of ranks) was used to assess statistical significance when two groups were compared. $P < 0.05$ was considered to indicate statistical significance.

## Material availability

Mouse lines used in this study can be obtained from Jackson Laboratories. Gnotobiotic mice are available upon request.

### Reporting summary

Further information on research design is available in the Nature Portfolio Reporting Summary linked to this article.

## Data availability

All data needed to evaluate the conclusions of this study are presented in the Article and Supplementary Information. The sequenced genomes of the *E. coli* 8178 WT, ATG*lacI* and GTG*lacI* strains have been deposited in NCBI (BioProject identifier PRJNA1096466) under accession numbers SAMN40757031, SAMN40757032 and SAMN40757033, respectively. The accession numbers of all downloaded genomes used for bioinformatic analysis are specified in Supplementary Table 6. The 10,643 *E. coli* genomes as well as the genomes used for other families were downloaded from the proGenomes v3 database (https://progenomes. embl.de/). Sequences of *E. coli* K-12 MG1665 metabolic regulator genes were obtained from BioCyc (https://biocyc.org/). Source data are provided with this paper.

## Code availability

The custom code used to select appropriate start codons and calculate their relative frequency per gene is available via GitHub at https://github.com/lukasmalfi/startcodons and via Zenodo at https://doi.org/10.5281/zenodo.12582572 (ref. 77).

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

## Acknowledgements

We thank M. Ackermann, O. Schubert and E. Kussell for helpful discussion. We would also like to thank members of the Wolf-Dietrich Hardt group, especially A. A. Younes and N. Santamaria de Souza, for their comments on the paper. We also thank the EPIC RCHCI staff for support of the animal work. Y.C. is supported by an EMBO long-term fellowship (ALTF-234-2020) and a flexibility grant from the SNF/NCCR Microbiome (51NF40_180575). C.S. is supported by a German Research Foundation fellowship (SCHU 3606/1-1). This work has been further funded by grants from the Swiss National Science Foundation (grant 310030_192569) attributed to N.N., L.M. and C.v.M.; the NCCR Microbiomes funding by the Swiss National Science Foundation to C.v.M. and W.-D.H.; and grant 310030_192567 from the Swiss National Science Foundation to W.-D.H. This project has received funding from the European Union's Horizon 2020 research and innovation programme under the Marie Skłodowska-Curie grant agreement number 956279 to W.-D.H.

## Author contributions

Y.C. and W.-D.H. conceived and designed the experiments. Y.C. trained and supervised M.A.S. Y.C. and M.A.S. performed the in vitro and in vivo experiments. Y.C., M.A.S. and C.S. analysed the data. M.K.-M.H. analysed the genomes of the *E. coli* 8178 strains. L.M., N.N. and C.v.M. performed the bioinformatic analysis. Y.C. wrote the paper with contribution from all authors.

## Funding

## Competing interests

The authors declare no competing interests.

## Additional information

**Extended data** is available for this paper at https://doi.org/10.1038/s41564-024-01775-x.

**Correspondence and requests for materials** should be addressed to Yassine Cherrak or Wolf-Dietrich Hardt.

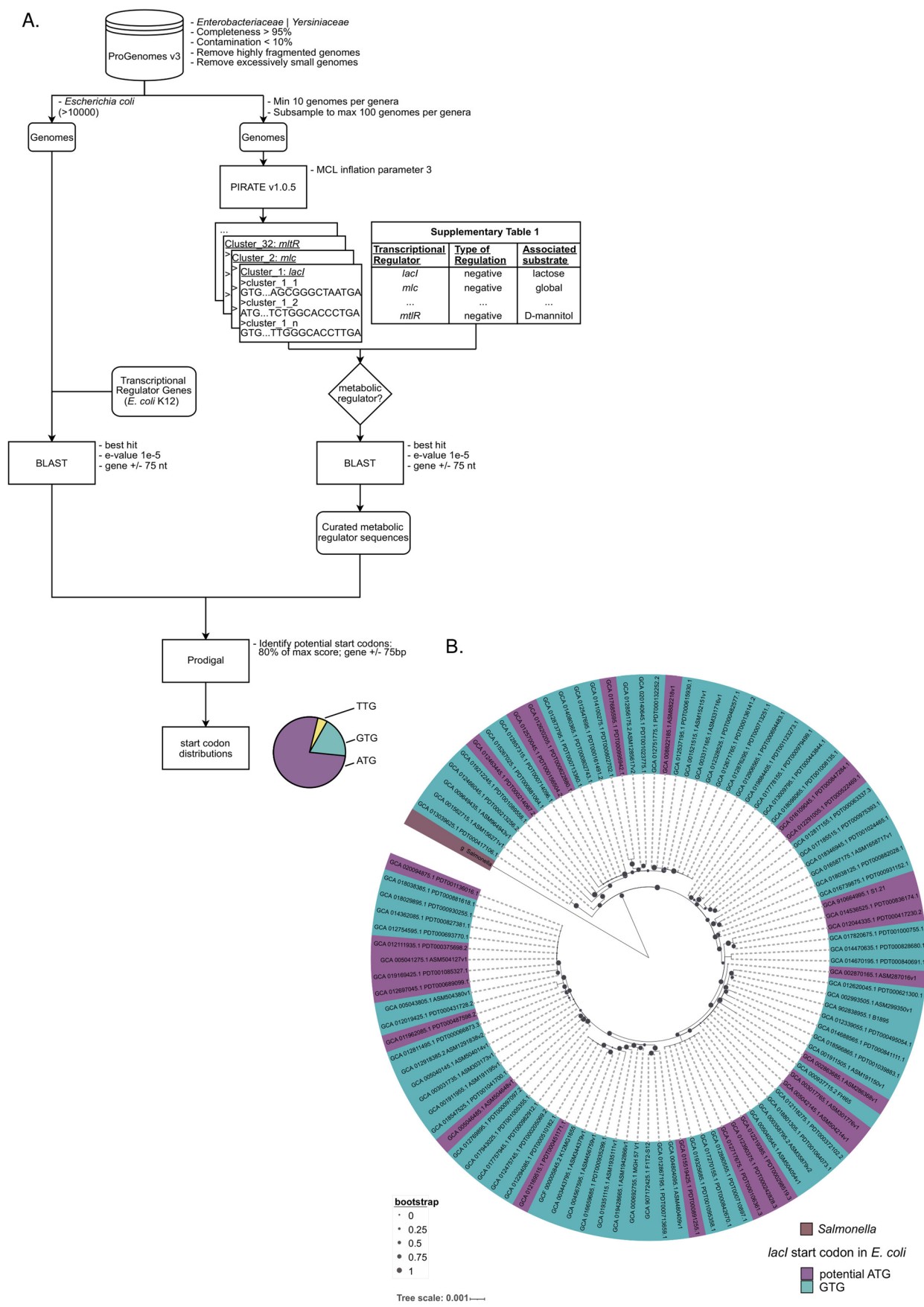

**Extended Data Fig. 1 | See next page for caption.**

**Extended Data Fig. 1 | A flow-chart depiction of the bioinformatic analysis to identify start codons of metabolic regulators in *Enterobacteriaceae* and *E. coli*. (a)** A flow-chart depiction of the bioinformatic analysis to identify start codons of metabolic regulators in *Enterobacteriaceae* genomes were retrieved from the progenomes v3 database. Orthologous gene clusters for the *Enterobacteriaceae* family were created using PIRATE v1.0.5. Gene clusters containing annotations of interest (Supplementary Tables 1, 2) were selected based on their short gene name. Metabolic regulators of the *E. coli* dataset were identified by a homology search against established metabolic regulator sequences of *E. coli* K-12 MG1665. For both datasets, start codons distributions were calculated as described in the method section. **(b)** Distribution of *lacI* ATG start codons within a subset of *E. coli* genomes. 76 genomes displaying exclusively a *lacI* GTG start codon and 26 genomes encoding a potential *lacI* ATG start codon were randomly selected for the analysis. The phylogenetic tree was calculated using 120 bacterial marker genes and *Salmonella* as an outgroup. Bootstrap support is indicated by the size of the circles. Genomes with a predicted *lacI* ATG start codons are non-monophyletic and are distributed throughout the *E. coli* phylogenetic diversity.

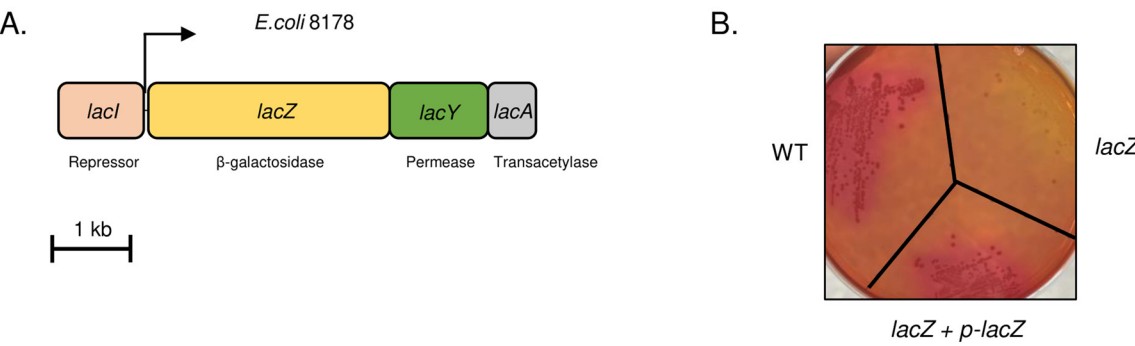

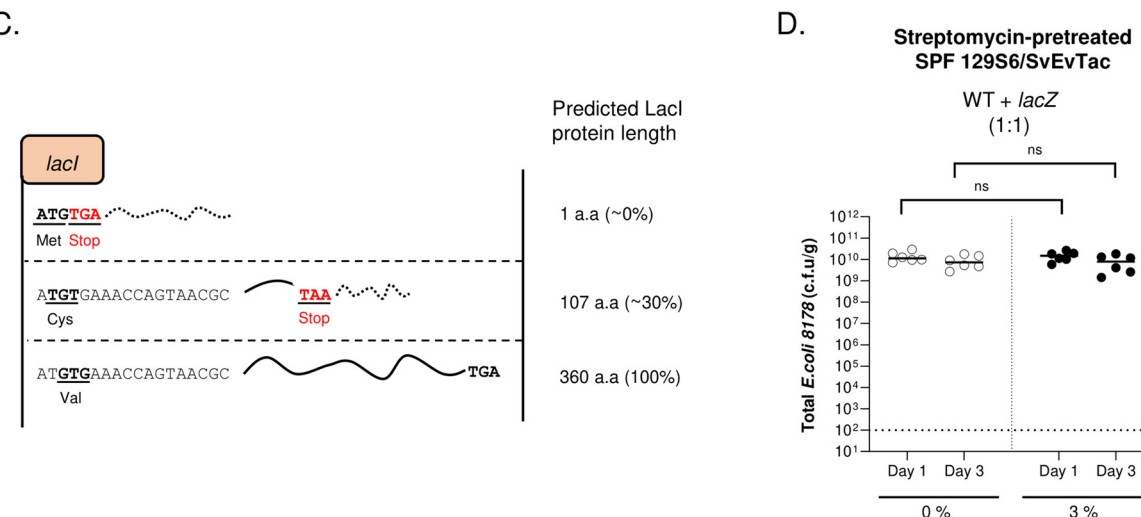

**Extended Data Fig. 2 | Analysis of the lactose utilization cluster in *E. coli* 8178.** (**a**) Schematic representation of the lactose operon in *E. coli* 8178. (**b**) WT and *lacZ*-deficient colonies on MacConkey agar plate highlighting lactose utilisation capacities. Expression of the *lacZ* gene *in trans* restores lactose utilisation properties in the *lacZ* mutant (*lacZ + p-lacZ*). (**c**) Sequence of the *lacI* gene in *E. coli* 8178. Translation from the canonical ATG start codon leads to a premature stop-codon. The predicted length of the LacI protein is indicated for each potential start codon in all 3 reading-frames. The percentage refers to the predicted LacI translated protein length relative to the WT version (**d**) Total *E. coli* 8178 loads (WT + *lacZ*) in colonised 129S6/SvEvTac from Fig. 2d. The mouse models, strains, and lactose concentration (0 % or 3 % w/v) in the drinking water are indicated. The x and y axis represent respectively the time post-inoculation and the load of *E. coli* 8178 in colony forming units (c.f.u) per gramme of faeces. Bars: median, dotted line: limit of detection. The results from at least 2 independent replicates are shown. Two-tailed Mann Whitney-U tests to compare two groups in each panel. p ≥ 0.05 not significant (ns).

A.

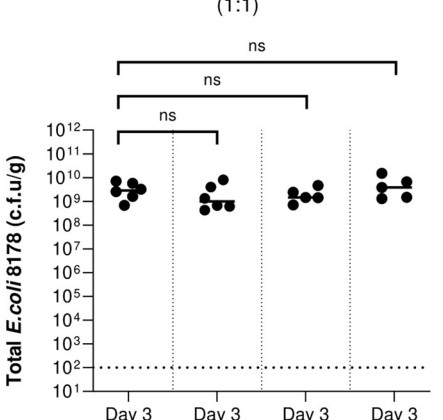

B.

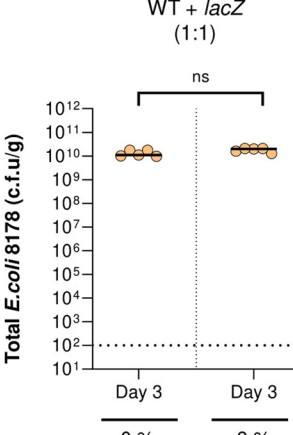

C.

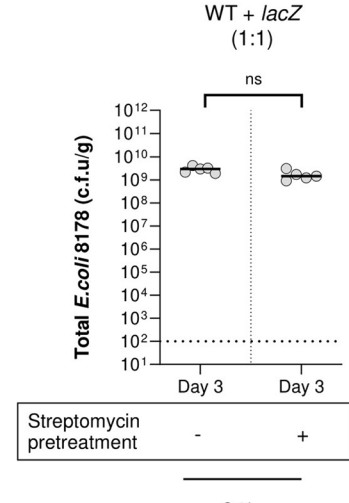

D.

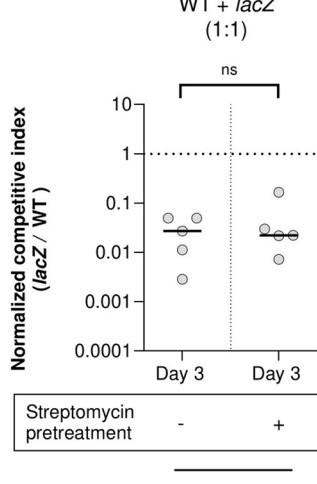

E.

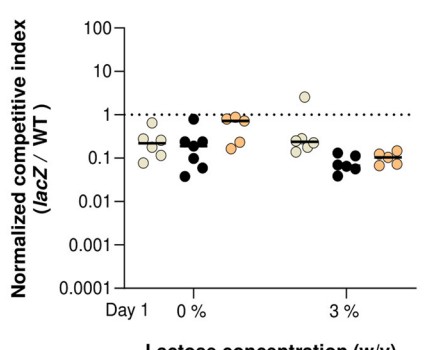

**Extended Data Fig. 3 | See next page for caption.**

**Extended Data Fig. 3 | Level of *E. coli* 8178 gut colonization in different mouse models.** (**a-b**) The total *E. coli* 8178 load (WT + *lacZ*) in colonised mice from Fig. 3b, c. The mouse models, strains, and lactose concentration (w/v) in the drinking water are indicated. The x and y axis represent respectively the time post-inoculation and the load of *E. coli* 8178 in colony forming units (c.f.u) per gramme of faeces. (**c-d**) Influence of the streptomycin-pretreatment on the *lacZ*-associated growth phenotype. C57BL/6 germ-free mice were inoculated with a 1:1 mixture of the WT and *lacZ E. coli* 8178 strains. The total *E. coli* 8178 load (C) and *lacZ*-associated competitive index (C.I) (D) in inoculated mice supplemented with lactose in the drinking water (3 % w/v) are indicated at day 3 post-colonisation. C57BL/6 LCM animals were streptomycin-pretreated (+) or

not (-) prior to inoculation. (**e**) Influence of the mouse microbiota and diet on *E. coli* 8178 *lacZ* mutant fitness. The *E. coli* 8178 WT and *lacZ* mutant strains were used to inoculate streptomycin-pretreated SPF 129S6/SvEvTac, streptomycin-pretreated SPF C57BL/6 or C57BL/6 germ-free mice. The counts of the WT and *lacZ* mutant strains were determined by selective plating from faecal samples at day 1 post-inoculation and used to calculate the C.I. (A-C) Dotted line: limit of detection. (D-E) Dotted line: C.I expected for a fitness-neutral mutation. (**a-e**) Bars: median. The results from at least 2 independent replicates are shown. Two-tailed Mann Whitney-U tests to compare two groups in each panel. p ≥ 0.05 not significant (ns).

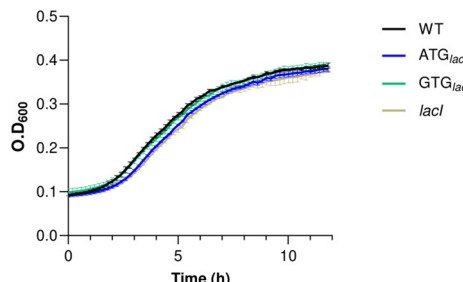

**Extended Data Fig. 4 | Nucleotide change within the *lacI* start codon does not have any global impact on *E. coli* 8178 growth *in vitro*.** The indicated strains were individually grown on a minimal medium supplemented with glucose. The mean and error of a biological replicate is represented.

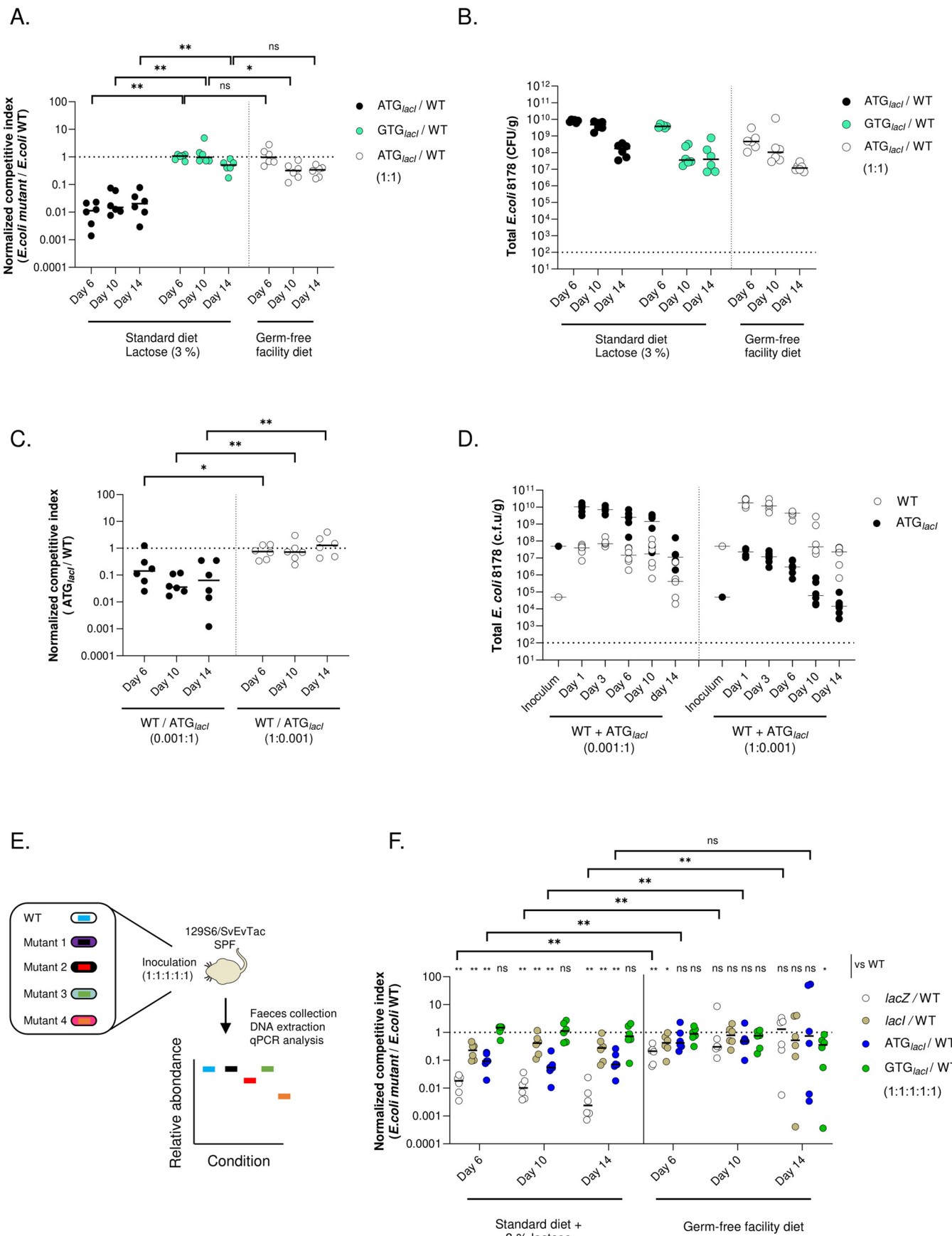

**Extended Data Fig. 5 | See next page for caption.**

**Extended Data Fig. 5 | Diet's influence on the *E. coli* 8178 *lacI* start codon associated fitness.** (**a**) Fitness of the ATG$_{lacI}$ and GTG$_{lacI}$ strains (from Fig. 5a) in streptomycin-pretreated 129S6/SvEvTac mice kept under different regimes at day 6, 10 and 14 post-inoculation. (**b**) Level of *E. coli* 8178 gut colonisation in 129S6/SvEvTac mice fed on different chows. The total *E. coli* 8178 load (WT + *lacZ*) in colonised mice from Fig. 5a and Extended Data Fig. 5a are presented. The diet and lactose concentration (w/v) in the drinking water are indicated. (**c**) The competitive fitness of the ATG$_{lacI}$ from Fig. 5b is presented at each dilution tested for the days 6, 10 and 14 post-colonisation. (**d**) Loads of the *E. coli* 8178 WT and ATG$_{lacI}$ strains from Fig. 5b and Extended Data Fig. 5c. (**e**) Experimental scheme summarising the fitness analysis of barcoded mutants. Barcoded *E. coli* 8178 strains carrying a specific mutation are equally mixed in the presence of the WT and used to inoculate mice. The distribution of barcoded mutants is analysed by qPCR from faecal samples. (**f**) Fitness analysis of barcoded *E. coli* 8178 mutants from Fig. 5c at day at day 6, 10 and 14 post-inoculation. (B-D) Dotted line: limit of detection. (A, C, F): Dotted line: C.I expected for a fitness-neutral mutation. (A-D, F) Bars: median. The results from at least 2 independent replicates are shown. Two-tailed Mann Whitney-U tests to compare two groups in each panel. $p \geq 0.05$ not significant (ns), $p < 0.05$ (*), $p < 0.01$ (**).

A.

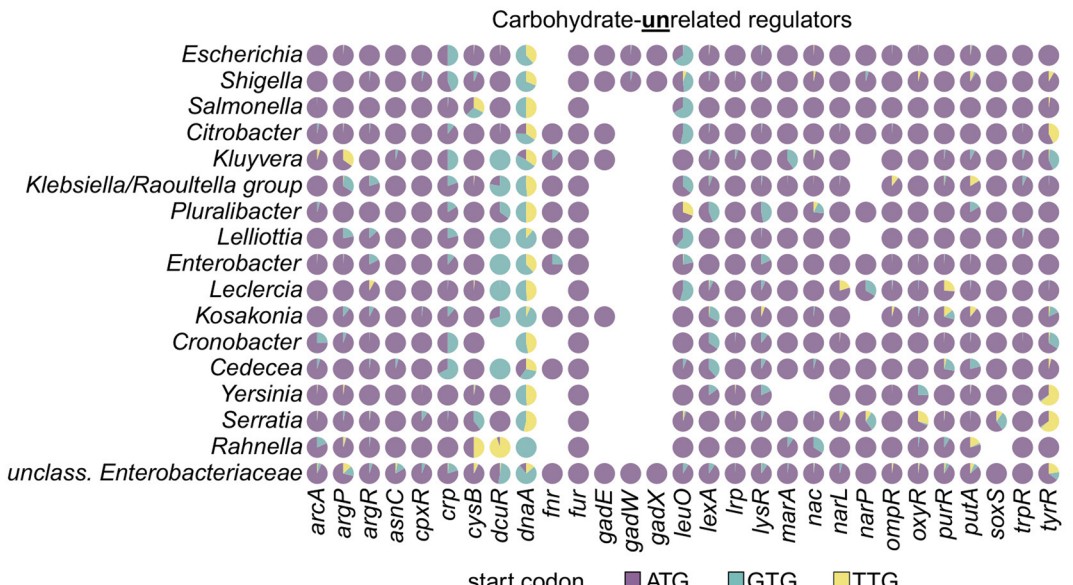

B.

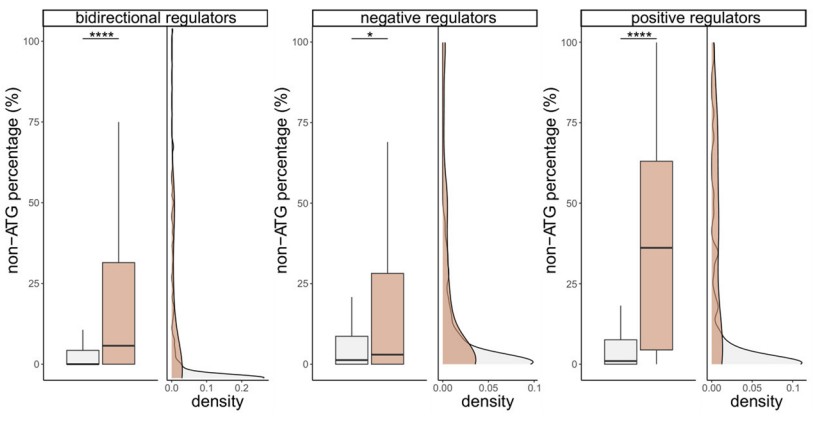

**Extended Data Fig. 6 | Distribution of non-canonical start codons in carbohydrate-unrelated metabolic regulators across the *Enterobacteriaceae* family.** (**a**) Alternative start codons are mainly observed in *dcuR*, *dnaA* and *tyrR*. The conservation level is represented by a circle-filled colour code which varies depending on the regulator and genus. Bacterial families were grouped according to their phylogenetic lineage. (**b**) Comparison of non-ATG start codon percentages between carbohydrate-related and unrelated transcriptional regulators. Regulators are grouped according to the type of regulation (bidirectional, negative, positive and all types combined). The right side of each subpanel shows a kernel density estimate of the non-ATG percentage. Two-tailed

Mann Whitney-U tests were used to compare each type of regulation. $p \geq 0.05$ not significant (ns), $p < 0.05$ (*), $p < 0.01$ (**), $p < 0.001$ (***), $p < 0.0001$ (****). In total, 32 carbohydrate related regulators (10 bidirectional, 17 negative and 5 positive) and 29 carbohydrate unrelated regulators (20 bidirectional, 4 negative and 5 positive) were analysed. Data are shown as median (black horizontal line) as well as the 25% and 75% percentiles (hinges). Whiskers extend from the hinges to the maxima and minima, no further than 1.5x distance of the inter-quartile range. The p-values for the bidirectional, negative and positive regulators are respectively $1.1 \times 10^{-5}$, $2.9 \times 10^{-2}$ and $5.8 \times 10^{-8}$.

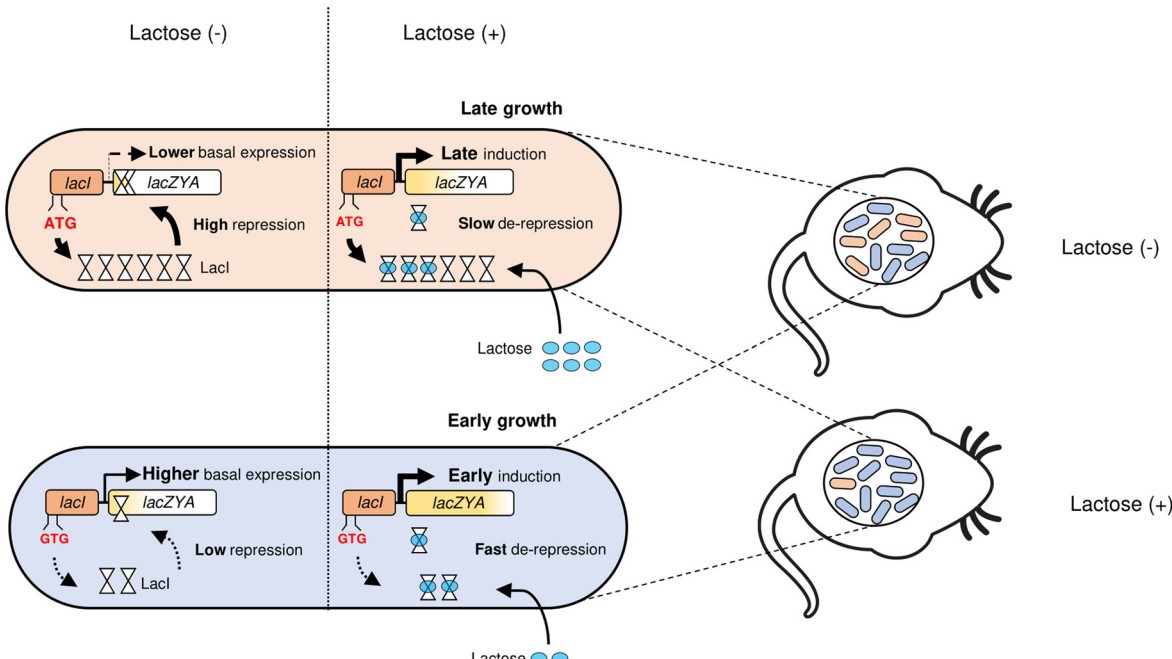

**Extended Data Fig. 7 | Model for the metabolic advantage conferred by the non-canonical start codon in the *E. coli* 8178 *lacI* gene.** Lactose plays a crucial role as a carbohydrate for *E. coli* 8178 growth in the mouse gut. Translation of *lacI* from a GTG start codon is associated with a lower repressor production (bottom). As a result, the strain displays elevated basal expression of the *lacZYA* operon, making it predisposed to lactose utilisation at a faster rate. Conversely, an isogenic strain encoding a *lacI* version with an ATG start codon features a stronger *lacZYA* repression level (top) In this context, the transition to lactose utilisation is further delayed, resulting in a fitness disadvantage in the presence of lactose (+) compared to the GTG$_{lacI}$ strain. The metabolic benefit linked to the *lacI* start codon is contextual and reduces in lactose-limited conditions (−).

# Reporting Summary

## Statistics

For all statistical analyses, confirm that the following items are present in the figure legend, table legend, main text, or Methods section.

| n/a | Confirmed | |
|---|---|---|
| ☐ | ☒ | The exact sample size (_n_) for each experimental group/condition, given as a discrete number and unit of measurement |
| ☐ | ☒ | A statement on whether measurements were taken from distinct samples or whether the same sample was measured repeatedly |
| ☐ | ☒ | The statistical test(s) used AND whether they are one- or two-sided _Only common tests should be described solely by name; describe more complex techniques in the Methods section._ |
| ☐ | ☒ | A description of all covariates tested |
| ☐ | ☒ | A description of any assumptions or corrections, such as tests of normality and adjustment for multiple comparisons |
| ☐ | ☒ | A full description of the statistical parameters including central tendency (e.g. means) or other basic estimates (e.g. regression coefficient) AND variation (e.g. standard deviation) or associated estimates of uncertainty (e.g. confidence intervals) |
| ☐ | ☒ | For null hypothesis testing, the test statistic (e.g. _F_, _t_, _r_) with confidence intervals, effect sizes, degrees of freedom and _P_ value noted _Give P values as exact values whenever suitable._ |
| ☒ | ☐ | For Bayesian analysis, information on the choice of priors and Markov chain Monte Carlo settings |
| ☒ | ☐ | For hierarchical and complex designs, identification of the appropriate level for tests and full reporting of outcomes |
| ☒ | ☐ | Estimates of effect sizes (e.g. Cohen's _d_, Pearson's _r_), indicating how they were calculated |

_Our web collection on statistics for biologists contains articles on many of the points above._

## Software and code

Policy information about availability of computer code

| Data collection | The custom code used to select appropriate start codons and calculate their relative frequency per gene is available at https://github.com/lukasmalfi/startcodons. |
|---|---|
| Data analysis | All data analysis softwares used in this study are described in the Materials and Methods. For mouse experiments, non-parametric statistical testing was performed using GraphPad Prism 8 for Windows. BLAST (blastn) was used to create databases from selected metabolic regulator genes in E. coli, and to search all genomes for them. The metabolic regulator sequences in Enterobacteriaceae families were selected using PIRATE v1.0.5. Prodigal v 2.6.3 was then used to call putative ORFs, and results were plotted with ggplot2 v3.3.5 in R 3.6.3. For statistical comparison of start codon usage between carbohydrate related and unrelated regulators, non-parametric statistical testing was performed in R 3.6.3. Phylogenetic trees were calculated with gtdbtk v 2.1.0 and visualized in iTOL v6.9.1 (https://itol.embl.de/version_history.cgi). The custom code used to select appropriate start codons and calculate their relative frequency per gene is available at https://github.com/lukasmalfi/startcodons. The whole genome assembly and annotation was performed with Flye 2.9.2, CLC workbench 23.0.1 (for polishing) and prokka 1.14.5. CLC workbench was also used for mutation analysis. |

For manuscripts utilizing custom algorithms or software that are central to the research but not yet described in published literature, software must be made available to editors and reviewers. We strongly encourage code deposition in a community repository (e.g. GitHub). See the Nature Portfolio guidelines for submitting code & software for further information.

## Data

Policy information about availability of data

All manuscripts must include a data availability statement. This statement should provide the following information, where applicable:

- Accession codes, unique identifiers, or web links for publicly available datasets
- A description of any restrictions on data availability
- For clinical datasets or third party data, please ensure that the statement adheres to our policy

All data needed to evaluate the conclusions in the article are presented in the paper and the supplementary information. The sequenced genomes of the E. coli 8178 WT, ATGlacI and GTGlacI strains have been deposited in NCBI (Bioproject ID: PRJNA1096466) under the accession number SAMN40757031, SAMN40757032, SAMN40757033 respectively. Source data are available for Figures 2-5 and Extended Data Figs 1-5. The accession numbers of all downloaded genomes used for bioinformatic analysis are specified in Supplementary Table 6.The 10,643 Escherichia coli genomes as well as the genomes used for other families were downloaded from the progenomes3 database (https://progenomes.embl.de/). Sequences of E. coli K-12 MG1665 metabolic regulator genes were obtained from Biocyc (https://biocyc.org/).

## Research involving human participants, their data, or biological material

Policy information about studies with human participants or human data. See also policy information about sex, gender (identity/presentation), and sexual orientation and race, ethnicity and racism.

| | |
|---|---|
| Reporting on sex and gender | *N/A* |
| Reporting on race, ethnicity, or other socially relevant groupings | *N/A* |
| Population characteristics | *N/A* |
| Recruitment | *N/A* |
| Ethics oversight | *N/A* |

Note that full information on the approval of the study protocol must also be provided in the manuscript.

# Field-specific reporting

Please select the one below that is the best fit for your research. If you are not sure, read the appropriate sections before making your selection.

☒ Life sciences ☐ Behavioural & social sciences ☐ Ecological, evolutionary & environmental sciences

For a reference copy of the document with all sections, see nature.com/documents/nr-reporting-summary-flat.pdf

# Life sciences study design

All studies must disclose on these points even when the disclosure is negative.

| | |
|---|---|
| Sample size | *At least 5 mice were used in each group. Sample size was chosen according to institutional directives and in accordance with the 3Rs rules (Replacement, Reduction and Refinement) guiding principles underpinning the humane use of animals in research, but no statistical analyses were performed to predetermine the sample sizes* |
| Data exclusions | *No data were excluded from the analysis.* |
| Replication | *All the experiments were performed at least twice, with all attempts at data replication being successful* |
| Randomization | *Mice were randomly allocated to different treatments* |
| Blinding | *Blinding does not apply to this study because the investigators needed to identify the cages of mice for subsequent treatments/infections with respective bacterial strains.* |

# Behavioural & social sciences study design

All studies must disclose on these points even when the disclosure is negative.

| | |
|---|---|
| Study description | N/A |

| Research sample | N/A |
|---|---|
| Sampling strategy | N/A |
| Data collection | N/A |
| Timing | N/A |
| Data exclusions | N/A |
| Non-participation | N/A |
| Randomization | N/A |

# Ecological, evolutionary & environmental sciences study design

All studies must disclose on these points even when the disclosure is negative.

| Study description | N/A |
|---|---|
| Research sample | N/A |
| Sampling strategy | N/A |
| Data collection | N/A |
| Timing and spatial scale | N/A |
| Data exclusions | N/A |
| Reproducibility | N/A |
| Randomization | N/A |
| Blinding | N/A |

Did the study involve field work?  ☐ Yes  ☒ No

## Field work, collection and transport

| Field conditions | N/A |
|---|---|
| Location | N/A |
| Access & import/export | N/A |
| Disturbance | N/A |

# Reporting for specific materials, systems and methods

We require information from authors about some types of materials, experimental systems and methods used in many studies. Here, indicate whether each material, system or method listed is relevant to your study. If you are not sure if a list item applies to your research, read the appropriate section before selecting a response.

## Materials & experimental systems

| n/a | Involved in the study |
|---|---|
| ☐ | ☒ Antibodies |
| ☒ | ☐ Eukaryotic cell lines |
| ☒ | ☐ Palaeontology and archaeology |
| ☐ | ☒ Animals and other organisms |
| ☒ | ☐ Clinical data |
| ☒ | ☐ Dual use research of concern |
| ☒ | ☐ Plants |

## Methods

| n/a | Involved in the study |
|---|---|
| ☒ | ☐ ChIP-seq |
| ☒ | ☐ Flow cytometry |
| ☒ | ☐ MRI-based neuroimaging |

# Antibodies

| | |
|---|---|
| Antibodies used | Anti-LacI Antibody (Sigma-Aldrich; 05-503-I); RNA polymerase beta Antibody (ThermoFischer Scientific, MA125425); Goat Anti-Mouse IgG Antibody Alkaline Phosphatase conjugate (Sigma-Aldrich, AP124A). |
| Validation | The LacI antibody was validated by western blot by the manufacturer in E. coli BL21 (https://www.sigmaaldrich.com/CH/de/product/mm/05503i). The RNA polymerase beta antibody was used and validated in the following studies : Zhou et al., Structure of the human cGAS-DNA complex reveales enhanced control of immune surveillance, Cell 2018; Lennings et al., Polar localization of the ATPase ClpV-5 occurs independent of type VI secretion system apparatus proteins in Burkholderia thailandansis, BMC Research Notes, 2019). |

# Eukaryotic cell lines

Policy information about cell lines and Sex and Gender in Research

| | |
|---|---|
| Cell line source(s) | N/A |
| Authentication | N/A |
| Mycoplasma contamination | N/A |
| Commonly misidentified lines (See ICLAC register) | N/A |

# Palaeontology and Archaeology

| | |
|---|---|
| Specimen provenance | N/A |
| Specimen deposition | N/A |
| Dating methods | N/A |

☐ Tick this box to confirm that the raw and calibrated dates are available in the paper or in Supplementary Information.

| | |
|---|---|
| Ethics oversight | N/A |

Note that full information on the approval of the study protocol must also be provided in the manuscript.

# Animals and other research organisms

Policy information about studies involving animals; ARRIVE guidelines recommended for reporting animal research, and Sex and Gender in Research

| | |
|---|---|
| Laboratory animals | 8 - to 12-week old C57BL/6 and 129S6/SvEvTac mice (The Jackson Laboratory) mice were held under specific pathogen-free (SPF) conditions at the EPIC facility at ETH Zürich (light/dark cycle 12:12 h, room temperature 21±1 ℃, humidity 50±10%). Germ-free mice were bred in flexible film isolators under strict exclusion of microbial contamination at the isolator facility (EPIC). |
| Wild animals | No wild animals were used |
| Reporting on sex | 8-12 weeks old mice of both sexes were randomly assigned to experimental groups |
| Field-collected samples | This work does not involve samples collection from fields |
| Ethics oversight | All animal experiments were reviewed and approved by Kantonales Veterinäramt Zürich under license ZH158/2019, ZH108/2022; ZH109/2022, complying with the cantonal and Swiss legislation |

Note that full information on the approval of the study protocol must also be provided in the manuscript.

## Clinical data

Policy information about clinical studies

All manuscripts should comply with the ICMJE guidelines for publication of clinical research and a completed CONSORT checklist must be included with all submissions.

| | |
|---|---|
| Clinical trial registration | *N/A* |
| Study protocol | *N/A* |
| Data collection | *N/A* |
| Outcomes | *N/A* |

## Dual use research of concern

Policy information about dual use research of concern

### Hazards

Could the accidental, deliberate or reckless misuse of agents or technologies generated in the work, or the application of information presented in the manuscript, pose a threat to:

No | Yes
- ☒ ☐ Public health
- ☒ ☐ National security
- ☒ ☐ Crops and/or livestock
- ☒ ☐ Ecosystems
- ☒ ☐ Any other significant area

### Experiments of concern

Does the work involve any of these experiments of concern:

No | Yes
- ☒ ☐ Demonstrate how to render a vaccine ineffective
- ☒ ☐ Confer resistance to therapeutically useful antibiotics or antiviral agents
- ☒ ☐ Enhance the virulence of a pathogen or render a nonpathogen virulent
- ☒ ☐ Increase transmissibility of a pathogen
- ☒ ☐ Alter the host range of a pathogen
- ☒ ☐ Enable evasion of diagnostic/detection modalities
- ☒ ☐ Enable the weaponization of a biological agent or toxin
- ☒ ☐ Any other potentially harmful combination of experiments and agents

## Plants

| | |
|---|---|
| Seed stocks | *N/A* |
| Novel plant genotypes | *N/A* |
| Authentication | *N/A* |

# ChIP-seq

## Data deposition

☐ Confirm that both raw and final processed data have been deposited in a public database such as GEO.

☐ Confirm that you have deposited or provided access to graph files (e.g. BED files) for the called peaks.

Data access links
*May remain private before publication.*
> *N/A*

Files in database submission
> *N/A*

Genome browser session
(e.g. UCSC)
> *N/A*

## Methodology

Replicates
> *N/A*

Sequencing depth
> *N/A*

Antibodies
> *N/A*

Peak calling parameters
> *N/A*

Data quality
> *N/A*

Software
> *N/A*

# Flow Cytometry

## Plots

Confirm that:

☐ The axis labels state the marker and fluorochrome used (e.g. CD4-FITC).

☐ The axis scales are clearly visible. Include numbers along axes only for bottom left plot of group (a 'group' is an analysis of identical markers).

☐ All plots are contour plots with outliers or pseudocolor plots.

☐ A numerical value for number of cells or percentage (with statistics) is provided.

## Methodology

Sample preparation
> *N/A*

Instrument
> *N/A*

Software
> *N/A*

Cell population abundance
> *N/A*

Gating strategy
> *N/A*

☐ Tick this box to confirm that a figure exemplifying the gating strategy is provided in the Supplementary Information.

# Magnetic resonance imaging

## Experimental design

Design type
> *N/A*

Design specifications
> *N/A*

Behavioral performance measures
> *N/A*

## Acquisition

Imaging type(s)
<br>*N/A*

Field strength
<br>*N/A*

Sequence & imaging parameters
<br>*N/A*

Area of acquisition
<br>*N/A*

Diffusion MRI ☐ Used ☐ Not used

## Preprocessing

Preprocessing software
<br>*N/A*

Normalization
<br>*N/A*

Normalization template
<br>*N/A*

Noise and artifact removal
<br>*N/A*

Volume censoring
<br>*N/A*

## Statistical modeling & inference

Model type and settings
<br>*N/A*

Effect(s) tested
<br>*N/A*

Specify type of analysis: ☐ Whole brain ☐ ROI-based ☐ Both

Statistic type for inference
<br>*N/A*

(See Eklund et al. 2016)

Correction
<br>*N/A*

## Models & analysis

| n/a | Involved in the study |
|---|---|
| ☒ | ☐ Functional and/or effective connectivity |
| ☒ | ☐ Graph analysis |
| ☒ | ☐ Multivariate modeling or predictive analysis |

Functional and/or effective connectivity
<br>*N/A*

Graph analysis
<br>*N/A*

Multivariate modeling and predictive analysis
<br>*N/A*

