## [Peer Review File · Nature Microbiology]

Peer Review Information

Journal: Nature Microbiology

Manuscript Title: Non-canonical start codons confer context-dependent advantages in carbohydrate utilisation for commensal *E. coli* in the murine gut

Corresponding author name(s): Professor Wolf-Dietrich Hardt

Reviewer Comments & Decisions:Decision Letter, initial version:

Message: 12th December 2023

Dear Professor Hardt,

Thank you for your patience while your manuscript "A non-canonical start codon in the lactose utilization repressor provides a context-dependent growth advantage to a commensal *E. coli* in the mouse gut" was under peer-review at Nature Microbiology. It has now been seen by 4 referees, whose expertise and comments you will find at the end of this email. Although they find your work of some potential interest, they have raised a number of concerns that will need to be addressed before we can consider publication of the work in Nature Microbiology.

In particular, you will see that among the points raised by the referees is a request for further analyses to support the conclusion that non-canonical start codons are specifically associated with carbohydrate utilisation regulators. There was also a request to confirm or provide further statistical analyses for some data sets. Referee #1 asked for further discussion of the alternative reasons for the selective benefits of non-canonical start codons and why certain strains affected in *lacI* expression exhibited given competitive advantages or disadvantages in different contexts. Referee #3 raised the point that polymorphism and frequency-dependent selection for regulator expression has been previously observed, as have differing effects of *lacZ* mutation upon bacterial colonisation of germ-free mice. These points should be discussed in the context of your findings. We feel that these are critical points which would need to be addressed for us to further consider a revised manuscript, alongside the remaining issues outlined in the referees' reports, which are clear and should be straightforward to address.

Should further experimental data allow you to address these criticisms, we would be happy to look at a revised manuscript.

Please include a data availability statement as a separate section after Methods but before references, under the heading "Data Availability". This section should inform readers about

3the availability of the data used to support the conclusions of your study. This information includes accession codes to public repositories (data banks for protein, DNA or RNA sequences, microarray, proteomics data etc...), references to source data published alongside the paper, unique identifiers such as URLs to data repository entries, or data set DOIs, and any other statement about data availability. At a minimum, you should include the following statement: "The data that support the findings of this study are available from the corresponding author upon request", mentioning any restrictions on availability. If DOIs are provided, we also strongly encourage including these in the Reference list (authors, title, publisher (repository name), identifier, year). For more guidance on how to write this section please see:

<http://www.nature.com/authors/policies/data/data-availability-statements-data-citations.pdf>

* If you have not done so already we suggest that you begin to revise your manuscript so that it conforms to our Article format instructions at <http://www.nature.com/nmicrobiol/info/final-submission>. Refer also to any guidelines provided in this letter.

When submitting the revised version of your manuscript, please pay close attention to our [href="https://www.nature.com/nature-portfolio/editorial-policies/image-integrity">Digital Image Integrity Guidelines](https://www.nature.com/nature-portfolio/editorial-policies/image-integrity). and to the following points below:

Note: This url links to your confidential homepage and associated information about

manuscripts you may have submitted or be reviewing for us. If you wish to forward this e-mail to co-authors, please delete this link to your homepage first.

Nature Microbiology is committed to improving transparency in authorship. As part of our efforts in this direction, we are now requesting that all authors identified as 'corresponding author' on published papers create and link their Open Researcher and Contributor Identifier (ORCID) with their account on the Manuscript Tracking System (MTS), prior to acceptance. This applies to primary research papers only. ORCID helps the scientific community achieve unambiguous attribution of all scholarly contributions. You can create and link your ORCID from the home page of the MTS by clicking on 'Modify my Springer Nature account'. For more information please visit please visit www.springernature.com/orcid.

If you wish to submit a suitably revised manuscript we would hope to receive it within 6 months. If you cannot send it within this time, please let us know. We will be happy to consider your revision, even if a similar study has been accepted for publication at Nature Microbiology or published elsewhere (up to a maximum of 6 months).

Yours sincerely,

Reviewer Expertise:

Referee #1: carbohydrate utilisation, microbial evolution

Referee #2: microbiota, metabolic interactions, evolution

Referee #3: microbial evolution, competition in gut

Reviewer Comments:

Reviewer #1 (Remarks to the Author):

This manuscript describes a very interesting study of the effect of LacI expression level on competitive fitness of *E. coli* strains. The authors find that the non-canonical GTG start codon of the LacI protein (the repressor of the lac operon) causes lower expression of the repressor relative to the canonical ATG start codon. This lower expression level is shown to be beneficial in regulating the lac operon. Competition experiments conducted in vivo within mice demonstrate that the wild-type strain outcompetes a strain altered to use the canonical ATG codon for LacI as well as a strain that lacks the repressor altogether. Part of the importance of this work is the close connection of in vitro and in vivo results, which together demonstrate that a single base pair change in a regulator, which increases its expression levels marginally, has a pronounced effect on competitive fitness in vivo. The authors explore phylogenetically the importance of non-canonical start codons, and use the case of LacI as a model system for understanding phylogenetic changes and their selective underpinnings.

Overall, I would recommend publication of this manuscript at Nature Microbiology if the authors could substantially address the following comments:

5Line 211, Fig. 2B: Is there a significant difference between wild type and lacI in the lactose condition? If so, why?

Line 313-314: It is unexpected that the expression level of the lacZYA reporter is significantly lower in the ATG lacI mutant regardless of IPTG concentration. I would expect that saturation with IPTG would cause all molecules of LacI to lose affinity for DNA and there should be no difference in expression levels at sufficiently high IPTG. Could the authors explain this finding? Or do they have data at higher IPTG levels where this residual repression is lost?

Line 343-344: Could the authors point to a specific figure that demonstrates the more optimal growth of wild-type compare to ATG lacI on lactose? It was unclear which of their experimental results they are referring to here.

Line 352: The authors state that wt outperforms by 100-fold but the horizontal bar on Day 3 in Fig. 5A indicates less, perhaps 20 or 30 fold? It is hard to tell exactly.

Lines 454-455: The observation that unusual start codons lead to different protein isoforms is interesting - is this possibility ruled out by the authors' results? Is there any possibility that ATG and GTG result in (slightly) different isoforms?

Lines 511-526: I found the entire discussion of noise in relation to the authors' findings to be highly speculative and largely irrelevant for understanding their results.

Lines 521-526: The authors state that there will be increased gene expression noise for the case of ATG. ATG causes higher expression levels of LacI. How does this lead to higher noise in LacI? Usually higher expression is associated with lower noise. It is not clear whether the authors are considering noise levels in LacI or downstream of LacI. Either way, I do not see much value in this totally speculative discussion that is not anchored in any of the authors' results. Some of this returns on line 550 where again I do not see a strong connection.

Regarding the GTG lacI strain: The GTG lacI strain is consistently seen at slightly lower levels in nearly all in vivo competitions versus wild-type (Fig. 5A, Day 1 and Day 3, all but one point out of 12 are lower than WT; Fig. 5B, 6 out of 6 points on standard diet and 6 out of 6 points on germ-free diet are lower than WT). Could the authors explain this result?

Interpretation of the results of phylogenetic analysis (Fig 6, and lines 386-427): There are many other ways of modulating expression level of a protein, e.g. codon composition, ribosomal binding site, transcript expression level. However the authors suggest that the changes in ATG vs. GTG start codon along the phylogeny are the major observed modulator of LacI expression level. If selection is favoring lower levels of LacI, it is unclear why the same point mutation is seen along the phylogeny rather than any number of other mutations impacting the above processes which could also lower expression levels. Therefore I would like to ask the authors: is it possible that the ATG vs GTG choice observed in phylogenies is in fact playing another role? For example, the ATG allele of LacI seems to exhibit higher repression of the operon even at high IPTG levels, which suggests an allelic effect beyond modulating expression level. Is there any other possible effect of ATG versus GTG choice that is not exclusive to expression level?

Methods details: I could not find strain details regarding construction of the lacZ strain. Likewise, additional details regarding how genome editing for the ATGlacI and GTGlacI strains was performed would be helpful. Were the details provided somewhere?

Minor comments:

Line 236-7: A bit more detail regarding the previous studies would be helpful, i.e. what are the contrasting findings or discrepancy relevant to the current study.

Line 304, Fig. 4B: In addition, it looks like the lacI lag phase is shorter than the wild-type.

Line 312: "derepression" rather than "depression"

Line 317: I think this should be 5 uM rather than 5 mM.

Line 367: Indicates a 1:1:1:1:1 mixture (with five strains) but only four are mentioned in the same sentence. The lacZ mutant is only mentioned in the next sentence which is a bit confusing, it sounds like there was a separate control performed but in fact lacZ is part of the five strain mixture.

Line 430: Missing a).

Fig. 4C: Y-axis should be O.D. rather than D.O.

Reviewer #2 (Remarks to the Author):

The manuscript "A non-canonical start codon in the lactose utilisation repressor provides a context-dependent growth advantage to a commensal E. coli in the murine Gut" by Cherrak and colleagues demonstrates that the start codon GTG in genes encoding for repressors of genes involved in the utilisation of inconsistently available nutrients can be selectively advantageous. The authors also provide evidence that the selective advantage could stem from a shortened lag time in utilising the nutrients compared to species with stricter repression. The figures are generally well-presented, and the manuscript reads clearly. In general, the study is innovative and will be of interest to a broad readership as it spans microbial genetics, metabolism and ecology.

Main comments

Lines 167-169: "[...] attributable to selection and not spontaneous random mutations". Please rephrase. Spontaneous random mutations also precede selection processes as mutations generate the genetic diversity for natural selection to act on. Perhaps the authors meant "attributable to selection and not genetic drift"? Also, it remains unclear why a GTG genotype frequency of only >1% would already indicate underlying natural selection. Please elaborate on why even low frequencies of GTG suggest selection.

Evidence for the selective advantage of the non-canonical start codons to be uniquely tied to carbohydrate utilisation is currently weak. No formal statistical analysis was performed (Fig 6), and critically, only three non-carbohydrate-related global transcription regulators have been chosen to compare against the many local and global transcription regulators of carbohydrate-related genes. This line of evidence could be strengthened by including more

transcription factors not involved in carbohydrate utilisation. Characterisation of regulator genes into activators, repressors and bi-directional regulators could reveal important patterns not yet addressed in the text.

Figure 3C) Error bar function is not described, and the bar heights represent the mean in this case, not the median, which should be mentioned in the figure's legend. The same applies to Figures 2B and 4C (mean is explained, but not the error bars).
CD) The order of the diets should be the same between the graphs and would read easier if D was arranged by increasing/decreasing lactose concentration

Figure 4B and 4D: The legend says that the growth curves display the mean OD and error. The error bars are only displayed for some of the data points, and it is not mentioned which error function is used (SE, SD, CI?). Each growth curve is the mean of only three independent experiments. Thus, I suggest showing each individual growth curve in addition to the mean (perhaps the three individual curves in muted colours and mean curves in saturated colours).

Figure 4C: There is no statistical analysis, although the results section (lines 310-320) describes significant results from group comparisons.

Lines 235-236: The term "diet" is misleading as results show this is still related to lactose concentration.

Fig 2A: Please change/add the unit "kDa" (with a lowercase "a") for the three numbers (50 kDa, 37 kDa, 38.6 kDa). The same applies to Fig 4A.

Fig 2E: The significance of the two genotype groups to the WT levels should also be added (as it is done in Fig 2D).

Line 768: Typo. Remove 's' from "codons".

Reviewer #3 (Remarks to the Author):

In this manuscript the authors explore how the fine tuning of gene expression in the gut can influence the competitive ability of a commensal strain of *Escherichia coli*. Importantly, they focus on the repressor of a well-known operon, the lac operon. They hypothesize, that the non-canonical GTG start codon in lacI, which is close to universal among *E. coli* strains, confers the ideal amount of expression to colonize the gut. They do this by comparing its competitive ability with the canonical ATG version of lacI and with the constitutive lacI mutant. By colonizing mice with a mixture of 5 genotypes (lacI, WT, GTG lacI, ATG lacI and lacZ) they follow their frequencies for 1 or 3 (?) days. Since the WT and GTG lacI remain the most frequent they conclude they display the right amount of expression to subsist in the presence of lactose.

The manuscript is clearly written and the experiments are well thought and carefully designed.

Main concerns:

- Throughout the manuscript, the authors use the words "infection" for the colonization process, however they don't report any infection symptoms in recipient mice. In case no symptoms of infection are present in mice, I believe the term "colonization" better

describes the procedure.

- The authors say *E. coli* 8178 relies on lactose metabolism to better colonize the gut. However, Fig 3D shows that *lacZ* is neutral when mice are under a diet devoid of lactose. Therefore, this is not only true in the presence of lactose (Fig 3D).

- pg 8 L248, the authors say *E. coli* 8178 relied on lactose metabolism in C57BL/6, despite the competitive defect of *lacZ* being the same with or without lactose added to the food.

- The authors observe that *lacZ* is deleterious in C57BL/6 GF with or without added lactose. In contrast, Pinto saw that *lacZ* was neutral in GF animals, irrespectively of adding or not lactose to the diet. This could be related either 1) to the different strain of *E. coli* (Pinto et al. used MG1655) or 2) to the fact that the authors did not treat GF mice with streptomycin, whereas Pinto et al. did. Antibiotics can affect the host independently of the microbiota and therefore treating SPF animals with streptomycin could eventually have led to host modification that could impact *lac* competitions. (Fig 3B) and finally, 3) animals are transferred to SPF facility on day 0, whereas in Pinto et al. experiment, animals were kept in the GF facility. Transferring GF mice monocolonized to SPF facility could allow further uncontrolled colonization with foreigner microorganisms.

- The authors do a colonization where they start with a mixture of 1:1:1:1 of barcoded versions of the *lacI*-deficient strain, WT, GT*GlacI*, and AT*GlacI* strains plus delta *lacZ*. They then estimate the competitive index of each of these genotypes against the wt, both in the presence and absence of lactose. While the wt has the highest competitive ability in the presence of lactose, its is important to note that they all start at a relatively high frequency (20%), and that other constitutive mutants (e.g. *srR*, Fig 6 in DOI:10.1371/journal.pgen.1006420 or *gatR*, fig 5 in doi:10.1093/molbev/msx221) have shown to be deleterious at high frequency (and beneficial at low frequency), but were still able to coexist with the regulated version of the operon. Therefore, it looks a bit simplistic to say that *lacI* is deleterious. It would be important to follow the colonization with the 5 genotypes for some weeks and see which mutants (including the *lacI*) get extinct and which are able to persist. It is possible that such an experiment would reveal that no single optimal expression of the *lac* operon is uniquely beneficial but rather several levels of expression can coexist.

From fig 4B and D it looks like wt, AT*GlacI*, GT*GlacI* and *lacI* have similar growth rates in the presence of lactose, the main differences potentially being related with the bacterial readiness to consume lactose. Other experiments (Fig 3 in doi:10.1093/molbev/msx221) have shown polymorphism to be selected in expression of the *gat* operon, where the two phenotypes (*gat*-neg and *gat*-pos) coexist without fixation of any of the two. In fact, when the authors tried to fix any of the two, the other would always emerge de novo. Even in the case of the *lac* operon, constitutive mutants have been observed to be selected in the gut. So, in sum, I think it is fair to suspect that the complex and ever-changing environment of the gut will favour polymorphism in detriment of a single level of operon expression, depending on the microbiota colonizing the gut as well as on host diet.

Minor:

- Pg 8 L265 replace "where there" by "when they"
- Abstract, L33 replace "abondance" by "abundance".
- Pg 14 L541, Galactitol repressor is *gatR* and not *galR*

Ana Margarida Sousa

Author Rebuttal to Initial comments

Reviewer Comments:

Reviewer #1 (Remarks to the Author):

This manuscript describes a very interesting study of the effect of LacI expression level on competitive fitness of *E. coli* strains. The authors find that the non-canonical GTG start codon of the LacI protein (the repressor of the lac operon) causes lower expression of the repressor relative to the canonical ATG start codon. This lower expression level is shown to be beneficial in regulating the lac operon. Competition experiments conducted *in vivo* within mice demonstrate that the wild-type strain outcompetes a strain altered to use the canonical ATG codon for LacI as well as a strain that lacks the repressor altogether. Part of the importance of this work is the close connection of *in vitro* and *in vivo* results, which together demonstrate that a single base pair change in a regulator, which increases its expression levels marginally, has a pronounced effect on competitive fitness *in vivo*. The authors explore phylogenetically the importance of non-canonical start codons, and use the case of LacI as a model system for understanding phylogenetic changes and their selective underpinnings.

Overall, I would recommend publication of this manuscript at Nature Microbiology if the authors could substantially address the following comments:

Response: We would like to thank referee#1 for dedicating her/his time to review our manuscript. We are pleased to note the genuine interest expressed about our study and hope that our detailed point-by-point responses will be compelling to support the publication of our work.

1. Line 211, Fig. 2B: Is there a significant difference between wild type and *lacI* in the lactose condition? If so, why?

Response: Thank you very much for pointing out the difference between the WT and *lacI* strain in presence of lactose. To confirm this observation and test for its significance, we run out additional *in vitro* β -galactosidase assays (biological triplicate) and the results are presented in the revised Figure 2B. We observed that the WT and *lacI* mutant displayed a significant difference in presence of lactose at the concentration tested (10 mM). We hypothesized that this disparity could be attributed to the incomplete derepression of the

10lacZYA operon in the WT strain under this condition. To test that, we conducted an additional experiment with an elevated concentration of lactose (100 mM). At this higher lactose concentration, the β -galactosidase levels in the WT and *lacI* mutant background is similar, confirming that the initial difference was indeed due to an incomplete repression alleviation. We have updated Fig. 2B and modified the caption accordingly. The results have been changed accordingly to contextualize this observation. Specifically, we replaced:

Translation initiation from the commonly universal ATG codon leads to premature termination of the process while translation from the GTG reading frame generates a full and functional protein actively repressing the lac operon in absence of lactose (Extended Data Fig. 2C, Fig. 2A-B).

by:

[L206] Translation initiation from the commonly universal ATG codon leads to premature termination of the process (Extended Data Fig. 2C). Conversely, translation from the GTG reading frame generates a full and functional protein that actively represses the lacZYA operon in absence of lactose; with this repression relieved by increasing concentrations of lactose (Fig. 2A, B, Extended Data Fig. 2C).

2. Line 313-314: It is unexpected that the expression level of the lacZYA reporter is significantly lower in the ATG_{lacI} mutant regardless of IPTG concentration. I would expect that saturation with IPTG would cause all molecules of LacI to lose affinity for DNA and there should be no difference in expression levels at sufficiently high IPTG. Could the authors explain this finding? Or do they have data at higher IPTG levels where this residual repression is lost?

Response: We thank referee#1 for highlighting the different lacZ expression levels between the WT and ATG_{lacI} strain at all the IPTG concentrations tested in the previous version of our manuscript. As suggested, we have extended this experiment to higher concentrations of IPTG (200µM and 500µM) to achieve the full derepression of the lacZYA operon in the ATG_{lacI} strain. Our results indicate that at higher IPTG concentrations, the lacZ expression levels became comparable between the WT, ATG_{lacI}, GTG_{lacI} and lacI strains. This confirms that the residual repression observed at 100µM can be overcome by increasing IPTG concentrations. Figure 4C and its associated caption have been updated. We have adjusted the Results section to incorporate these findings and replaced:

In contrast to lacI, expression lacZYA reporter in the ATG_{lacI} mutant was significantly lower, regardless of the IPTG concentration (Figure 4C). Notably, we observed that gfp signal resulting from the lacZYA operon transcription in the WT and GTG_{lacI} strains was overall higher than the ATG_{lacI} mutant, even in the absence of IPTG. Furthermore, a slight increase in the IPTG concentration (5 µmM) was sufficient for the WT and GTG_{lacI} strains to achieve a similar high level of the lacZYA expression as the lacI mutant. From this experiment, we concluded that utilization of the GTG start codon in lacI was sufficient to enhance the sensitivity and basal expression of the lacZYA operon (Figure 4C).

by:

[L320] At the highest IPTG concentration tested (500 µM), the WT, GTG_{lacI} and ATG_{lacI} strains displayed reporter signals similar to the lacI background, indicating that the full derepression of the lacZYA operon is achieved under this condition. Notably, the gfp signal resulting from the lacZYA operon transcription in the WT and GTG_{lacI} strains was consistently higher than in

the ATG_{lacI} mutant at any tested intermediate IPTG concentration (1; 5; 20; 100 μ M). This trend held true even in the absence of IPTG, demonstrating that the utilization of the GTG start codon in lacI was sufficient to enhance the sensitivity and basal expression of the lacZYA operon (Fig. 4C).

3. Line 343-344: Could the authors point to a specific figure that demonstrates the more optimal growth of wild-type compare to ATG_{lacI} on lactose? It was unclear which of their experimental results they are referring to here.

Response: We apologize for the lack of clarity in this section. This sentence pertains to Figure 4D which demonstrates that, upon switching from LB to a Minimal Medium + Lactose condition, the WT strain displayed a shorter lag phase, and thus grows more optimally than the ATG_{lacI} mutant background. We have now clarified this part by (1) adding the related figure and (2) pointing out more clearly under which condition such lag phase difference is observed.

We replaced:

As a result of our in vitro experiments, we observed that the WT E. coli 8178 strain grows more optimally on lactose compared to its ATG_{lacI} counterpart.

by:

[L352] *As a result of our in vitro experiments, we observed that the WT E. coli 8178 strain displayed a shorter lag phase than the ATG_{lacI} background, particularly when transitioning from a complex to a minimal media (Figure 4D).*

4. Line 352: The authors state that wt outperforms by 100-fold but the horizontal bar on Day 3 in Fig. 5A indicates less, perhaps 20 or 30 fold? It is hard to tell exactly.

Response: The sentence in this section relates to Figure 5A showcasing the *in vivo* fitness defect associated with the canonical ATG start codon in *E. coli* 8178 *lacI* gene. The values of the 6 competitive indexes are 0.06, 0.3, 0.04, 0.025, 0.007 and 0.021, indicating that the WT outperforms the ATG_{lacI} by 16.6-, 3-, 25-, 40-, 142- and 47- fold respectively. We fully acknowledge referee#1's observation that the median (0.0325) is not reflecting the indicated 100-fold reduction. We apologize for this mistake and have replaced the following sentence:

Under these conditions, we found that the WT strain outperformed the isogenic ATG_{lacI} mutant by approximately 100- fold within 3 days (Figure 5A)

by:

[L361] Under these conditions, we found that the WT strain outperformed the isogenic ATG_{lacI} mutant by approximately 30- fold within 3 days (Figure 5A).

5. Lines 454-455: The observation that unusual start codons lead to different protein isoforms is interesting - is this possibility ruled out by the authors' results? Is there any possibility that ATG and GTG result in (slightly) different isoforms?

Response: We thank referee#1 for raising such an important question and we are happy to provide a clarification. In eukaryotic cells, alternative start codons positioned at different locations can give rise to distinct protein isoforms (Touriol et al., 2003). Whether this phenomenon also occurs in bacteria remains unknown. However, we agree with referee#1 that the presence of alternative start codons in proximity outlined by our bioinformatic analysis may indicate a potential function in protein isoforms.

In line with our initial objective to investigate the impact of canonical versus non canonical start codons in bacterial physiology, we deliberately excluded the possibility

of generating different protein isoforms. To achieve this, we capitalized on the extensive body of literature on the *lacZYA* operon (from both functional and structural perspectives: Farabaugh, 1978; Lewis *et al.*, 1996; Blattner *et al.*, 1997) to precisely replace the known *lacI* start codon, by the canonical ATG version, at the same location. The accuracy of our mutation was further validated by sequencing analysis. From that and based on the literature, we can confidently assert that the ATG- and GTG version of *lacI*, being positioned at the same location, yield to similar protein composition.

Additionally, it is important to note that in bacteria, the first amino acid incorporated in a protein is a formyl methionine, regardless of the nature of the start codon (ATG, GTG, TTG) (Adams *et al.*, 1996; Clark *et al.*, 1996; Milón *et al.*, 2012). This is further exemplified by earlier protein sequencing works highlighting the Methionine as the first amino acid of the LacI protein, despite starting with a GTG start codon (Beyreuther *et al.*, 1973; Farabaugh, 1978). Consequently, we can safely claim that the ATG_{*lacI*} and GTG_{*lacI*} strains yield lac-repressor proteins with identical N-termini.

To avoid any confusion, we have added the missing details and replaced:

*Mechanistically, the use of GTG or TTG start codon leads to a sub-optimal translation efficiency reflected by an 8-to-12-fold decrease in the expression rate compared to ATG (Sussman *et al.*, 1996; O'Donnell & Janssen, 2001).*

by:

[L128] *Mechanistically, and despite recruiting the same N-formyl methionyl-tRNA (Adams *et al.*, 1966; Clark *et al.*, 1996), the use of GTG or TTG start codon leads to a sub-optimal translation efficiency reflected by an 8-to-12-fold decrease in the expression rate compared to ATG (Sussman *et al.*, 1996; O'Donnell & Janssen, 2001).*

and:

*To this aim, we carefully selected the lacI start codon sequence based on the extensive body of functional and structural validation studies available in the literature (Farabaugh, 1978; Blattner *et al.*, 1997; Lewis *et al.*, 1996).*

by:

[L291] *To this aim, we carefully selected the lacI start codon sequence based on the extensive body of functional and structural validation studies available in the literature (Farabaugh,*

1978; Blattner et al., 1997; Lewis et al., 1996). *By specifically targeting the known *lacI* start codon, we expect to alter the protein expression level without generating different isoforms.*

and:

*Using a lactose-rich diet, we found that the growth of both the *lacI* and *ATG_{lacI}* mutants was impeded compared to the WT.*

by:

[L524] *To study the role of the atypical *lacI* GTG start codon in *E. coli* 8178, we introduced a single point mutation (G→A) precisely at the known *lacI* start codon position. By doing so, we*

anticipated that both the WT and ATG_{lacI} strains would produce identical LacI protein sequence but in variable amounts. Using a lactose-rich diet, we found that the growth of both the lacI and ATG_{lacI} mutants was impeded compared to the WT.

Nonetheless, with the exception of our *lacI* mutagenesis design and study framework, we cannot rule out the hypothesis that alternative start codons can additionally generate distinct protein isoforms. We thank referee#1 for bringing this to our attention and have replaced:

For instances, translation from unusual and differently placed start codons has been proposed to contribute to generate protein isoforms harbouring different N-termini (Touriol et al., 2003; Bazykin & Kochetov, 2011, Ivanov et al., 2017).

By

[L479] *For instances, translation from unusual and differently placed start codons in eukaryotic genomes has been proposed to contribute to generate protein isoforms harbouring different N-termini (Touriol et al., 2003; Bazykin & Kochetov, 2011, Ivanov et al., 2017). In bacteria, it is yet to be determined whether the presence of multiple alternative start codons in close proximity is associated with the generation of protein variants.*

6. Lines 511-526: I found the entire discussion of noise in relation to the authors' findings to be highly speculative and largely irrelevant for understanding their results.

Response: We agree that the discussion about the noise is not directly related to our work, and we are sorry to read that referee#1 found it irrelevant. Our decision to discuss about a potential link between the nature of the start codon nature and the expression noise was motivated by multiple reasons. One of them is related to the observed stochastic fluctuation of the *lacZYA* operon in *E. coli* (Elowitz et al., 2002; van Hoek & Hogeweg, 2007), suggesting that alternative start codons and their impact on translation may additionally modulate the level of *lacZYA* expression noise. Additionally, considering the fitness advantage conferred by the decreased expression of *lacI*, it remains unclear why *E. coli* strains bearing a GTG mutation on the LacI start codon is so strongly selected while mutations in the promoter can lead to similar outcomes. We believe that the answer to this question is related to the noise expression. As mentioned in the manuscript, recent studies have demonstrated that an infrequent transcription followed by an efficient translation generates a higher noise than an efficient transcription followed by an inefficient translation (Ozbudak et al., 2002). Consequently, a reduced

18expression due to a promoter mutation would result in a higher noise compared to an inefficient translation. We believe that mentioning this in the discussion, albeit speculative, can offer additional insights into the robust preference of *lacI* GTG start codons and ultimately open new research directions. We tried to emphasize this aspect on the discussion and have substituted:

In addition to conferring a metabolic advantage, the reduction of lacI protein levels at the translation stage may also directly impact the regulation of noise within this sugar utilization cluster.

by:

[L561] *Having demonstrated the metabolic benefit associated to the decrease of lacI expression, it remains unclear why the GTG start codon is so favourably selected, especially when in average, lacI downregulation can be similarly achieved at the transcriptional level. We believe that the selection of non-canonical start codons over promoter mutations may be associated with their respective contributions to noisy gene expression.*

However, we will be willing to remove this part from the main manuscript and add it in the “Additional Discussion” section, should the editor and other referees agree with this suggestion.

7. Lines 521-526: The authors state that there will be increased gene expression noise for the case of ATG. ATG causes higher expression levels of LacI. How does this lead to higher noise in LacI? Usually higher expression is associated with lower noise. It is not clear whether the authors are considering noise levels in LacI or downstream of LacI. Either way, I do not see much value in this totally speculative discussion that is not anchored in any of the authors’ results. Some of this returns on line 550 where again I do not see a strong connection.

Response: We thank referee for raising this important point and are pleased to provide a clarification. We agree that a high gene expression leads in general to a lower noise. However, modulation of gene expression can be achieved in two different ways, i.e. by enhancing the transcription or translation rate, each one of them contributing differently to the noise level. It is important to note that although both scenarios may lead to similar LacI protein concentrations, high transcription coupled with inefficient translation generates less noise compared to infrequent transcription followed by efficient translation (Ozbudak et al., 2002). Based on that, we proposed that inefficient translated regulatory genes (*i.e.* non-ATG start codons, mutation in the RBS) may be favoured over promoter mutations for the control of carbohydrate utilization operons due to their low-noise characteristics. We believe that this section is important as it allows us to elaborate on the potential benefit of alternative start codons vs promoter mutations. Nonetheless, as mentioned above, we will be happy to remove it if the editor and other referees concur with this suggestion.

8. Regarding the GTGLacI strain: The GTGLacI strain is consistently seen at slightly lower levels in nearly all in vivo competitions versus wild-type (Fig. 5A, Day 1 and Day 3, all but one point out of 12 are lower than WT; Fig. 5B, 6 out of 6 points on standard diet and 6 out of 6 points on germ-free diet are lower than WT). Could the authors explain this result?

Response: We thank referee#1 for highlighting the slight decreased in the load of the GTG_{lacI} mutant compared to WT strain. Figure 5A depicts the results obtained from the selective plating of the WT and GTG_{lacI} strains. In contrast to the WT strain, we used antibiotic-based selection (ampicillin) to exclusively enrich and detect the GTG_{lacI} clones. We believe that this procedure may have impacted the associated growth of the GTG_{lacI} strain, as selection based on antibiotic is often linked to a slight growth defect. To illustrate this, we grew a *E. coli* 8178 WT strain (which carries an ampicillin resistance cassette) and plated it in both antibiotic-free and ampicillin-supplemented plates. Despite incubating similar quantities, we consistently observed (n=6) more colonies on

the antibiotic-free plate. We have illustrated this plating bias, a well-known phenomenon in the field, by plotting the ratio of the counts determined on antibiotic vs antibiotic-free plates. We concluded that the difference between the WT and GTG_{lacI} strain in Figure 5A originates from the antibiotic selection applied to the latter.

In comparison to Figure 5A, Figure 5C results from a qPCR-based analysis. Although the load associated to the GTG_{lacI} strain appears to be lower than the WT, a direct comparison of the raw c.f.u between the WT and GTG_{lacI} strains reveals similar load with no statistically significant difference. We attribute these slight differences to potential sources of variation and error during the DNA extraction, plate preparation and analysis steps. To avoid any confusion, we have incorporated the results of the statistical test when comparing each barcoded mutant with the WT strain.

9. Interpretation of the results of phylogenetic analysis (Fig 6, and lines 386-427): There are many other ways of modulating expression level of a protein, e.g. codon composition, ribosomal binding site, transcript expression level. However the authors suggest that the changes in ATG vs. GTG start codon along the phylogeny are the major observed modulator of *LacI* expression level. If selection is favoring lower levels of *LacI*, it is unclear why the same point mutation is seen along the phylogeny rather than any number of other mutations impacting the above processes which could also lower expression levels. Therefore I would like to ask the authors: is it possible that the ATG vs GTG choice observed in phylogenies is in fact playing another role? For example, the ATG allele of *LacI* seems to exhibit higher repression of the operon even at high IPTG levels, which suggests an allelic effect beyond modulating expression level. Is there any other possible effect of

22ATG versus GTG choice that is not exclusive to expression level?

Response: This is a completely valid point, and we thank referee#1 for offering us the opportunity to expand on it. We first would like to comment on the assumption that “the changes in ATG vs. GTG start codon along the phylogeny are the major observed modulator of *lacI* expression level” raised by the referee. Our bioinformatics analysis highlighted a significant conservation of the GTG start codon over the canonical ATG version in the *lacI* gene of different species. Building on this observation, we posited [L431] that “*lactose utilization benefit associated to the non-canonical lacI GTG start codon can be extended – in a strain-dependent manner – to other genera from the Enterobacteriaceae family.* However, we wish to clarify that we did not assert that ATG vs GTG change seen in

our analysis is the major observed modulator of *lacI* expression. As mentioned by the referee, making such assumption would require additional sequence analysis, specifically on the promoter region, to be able to compare the occurrence of GTG start codons against any other mutations in the promoter. Given that our bioinformatic pipeline was designed to specifically look at the start codon region, we thus cannot (and did not) assume that the GTG start codon is the major *lacI* expression modulator. However, it is evident that the strong preference for GTG start codons indicates an important and conserved function in *lacI* downregulation, which we try to emphasize in the discussion section.

While the absence of further computational analysis (*i.e.* promoter sequence) prevent us from claiming that the ATG vs GTG mutation is the main contributor of *lacI* downregulation, the strong conservation of the GTG *lacI* version raises an important question: why a mutation in the start codon would be selected so favourably whereas a similar outcome could be achieved by accumulating mutations in the promoter? As mentioned by referee#1, one can hypothesize that alternative start codons may potentially have additional roles unrelated to gene expression. However, we would like to draw to the referee's attention that the ATG allele of *lacI* exhibits indeed a higher repression level that we finally could alleviate by further increasing the IPTG concentration (Figure 4C). Additionally, and to the best of our knowledge, there is no study emphasizing the function of non-canonical start codons in bacterial translation-independent processes. We attempted to offer a preliminary answer to this fundamental question by emphasizing the effect of an inefficient translation in gene expression noise. However, as already mentioned, this remains hypothetical. Answering this complex biological question would necessitate further works which unfortunately fall beyond the scope of our current study. Nevertheless, we agree with referee#1 that, in absence of any further evidence, the role of non-canonical start codon in translation-independent processes remains plausible, which we have explicitly mentioned now in the discussion.

We have clarified our statement on the contribution of non-canonical start codons in *lacI* expression modulation. We also acknowledge the translation-independent function hypothesis of non-canonical start codon. Specifically, we have replaced:

In addition to conferring a metabolic advantage, the reduction of lacI protein levels at the translation stage may also directly impact the regulation of noise within this sugar utilization cluster.

by:

[L561] *Having demonstrated the metabolic benefit associated to the decrease of lacI expression, it remains unclear why the GTG start codon is so favourably selected, especially when in average, lacI downregulation can be similarly achieved at the transcriptional level. We believe that the selection of non-canonical start codons over promoter mutations may be associated with their respective contributions to noisy gene expression.*

Additionally, we have added:

[L578] *Finally, it is worth mentioning that the strong preference for the GTG start codon in lacI and in certain other sugar utilization regulators may indicate a potential function of non-*

canonical start codons that expands beyond translation processes. Further work is required to investigate this hypothesis thoroughly.

10. Methods details: I could not find strain details regarding construction of the lacZ strain. Likewise, additional details regarding how genome editing for the ATG_{lacI} and GTG_{lacI} strains was performed would be helpful. Were the details provided somewhere?

Response: We apologize for any lack of clarity in the Methods details. All information regarding the strain design can be found within the section titled “Strain, media and chemicals”. The lacZ mutant in *E. coli* 8178 was generated using a modified version of the lambda red recombinase-dependent one-step inactivation procedure (Datsenko & Wanner, 2000). The design of the ATG_{lacI} and GTG_{lacI} strains were achieved through a suicide vector adapted from earlier works (Link et al., 1997). We have amended the section with additional details, and specifically replaced:

Gene deletion and fitness neutral barcode insertion were achieved in E. coli 8178 using a modified version of the lambda red recombinase-dependent one-step inactivation procedure (Datsenko & Wanner, 2000). Briefly, the antibiotic cassette (kanamycin for gene deletion, ampicillin for barcode insertion) was PCR-amplified using primer pairs carrying a 50-nucleotide extension homologous to the adjacent targeted region. Mutants were obtained following the electroporation of the PCR product into E. coli 8178 cells expressing the lambda red recombinase from the pSIM5 plasmid and incubated on selective media (Diner et al., 2011). Gene deletion was confirmed by colony PCR. LacI point mutation at the native locus were engineered by allelic replacement using the pKO3 suicide vector conferring chloramphenicol resistance and sucrose sensitivity (Link et al., 1997).

by:

[L643] *Insertion of fitness-neutral barcodes and deletion of the lacZ gene into E. coli 8178 were achieved using a modified version of the lambda red recombinase-dependent one-step inactivation procedure (Datsenko & Wanner, 2000). Briefly, the antibiotic resistance cassette (kanamycin for gene deletion, ampicillin for barcode insertion) was PCR-amplified using primer pairs carrying a 50-nucleotide extension homologous to the adjacent targeted region. Mutants were obtained following the electroporation of the PCR product into E. coli 8178 cells expressing the lambda red recombinase from the pSIM5 plasmid and incubated on selective media (Diner et al., 2011). Gene deletion was confirmed by colony PCR. The lacI point mutation at the native locus was conducted by allelic replacement using the pKO3 suicide vector (Link et al., 1997). Briefly, the lacI region including the start codon and 500 base pairs upstream and downstream was cloned into the pKO3 plasmid which carried the sacB gene for*

26counter-selection on sucrose-containing media (Gay et al., 1985). Following the mutagenesis of the start codon by PCR amplification, the plasmid was introduced into E. coli 8178 and we selected on ampicillin plates for mutant with a single-crossover integration of the plasmid into the chromosome. A sucrose-based counter selection, colony PCR and sequencing was applied to identify clones carrying the desired marker-less nucleotide exchange at the lacI start codon.

Minor comments:

11. Line 236-7: A bit more detail regarding the previous studies would be helpful, i.e. what are the contrasting findings or discrepancy relevant to the current study.

Response: We have further detailed this part and replaced:

Recent studies have presented contrasting findings regarding the role of lactose in E. coli gut colonization (Fabich et al., 2008; Conway & Cohen; 2015; Pinto et al., 2021). Despite the difference in the tested strains, we speculated that the host microbiota and diet were contributing to this discrepancy.

by:

[L236] *Recent studies have presented contrasting findings regarding the role of lactose in E. coli gut colonization (Fabich et al., 2008; Conway & Cohen; 2015; Pinto et al., 2021). Specifically, it was demonstrated that the E. coli UPEC and EPEC strains utilize lactose in vivo, whereas lactose metabolism was found dispensable in E. coli Nissle and EDL933 strains (Fabich et al., 2008; Maltby et al., 2013). More strikingly, contrasting findings were observed with E. coli MG1655, (Fabich et al., 2008; Pinto et al., 2021), suggesting that despite the differences in the tested strains, the host microbiota and diet composition may contribute to this discrepancy (Pinto et al., 2021).*

12. Line 304, Fig. 4B: In addition, it looks like the *lacI* lag phase is shorter than the wild-type.

Response: We have now specified it in the text and replaced:

*However, we observed distinct lag phases after the switch from glucose to lactose, which were markedly shortened (by ~ 60 minutes) in the WT, *lacI* and complemented GTG_{lacI} mutants compared to the ATG_{lacI} strain (Figure 4B).*

by:

[L311] *However, we observed distinct lag phases after the switch from glucose to lactose, which were markedly shortened (by ~ 60 minutes) in the WT and complemented GTG_{lacI} mutants, and even further in the *lacI* background, compared to the ATG_{lacI} strain (Fig. 4B).*

13. Line 312: “derepression” rather than “depression”

Response: The correction has been added.

14. Line 317: I think this should be 5 uM rather than 5 mM.

Response: The correction has been added.

15. Line 367: Indicates a 1:1:1:1:1 mixture (with five strains) but only four are mentioned in the same sentence. The lacZ mutant is only mentioned in the next sentence which is a bit confusing, it sounds like there was a separate control performed but in fact lacZ is part of the five strain mixture.

Response: We have corrected it and replaced:

To study the potential correlation between the bacterial growth fitness and the level of lacZYA repression, we infected I29S6/SvEvTac mice with a 1:1:1:1:1 mixture of barcoded versions of

the *lacI*-deficient strain, WT, *GTG_{lacI}*, and *ATG_{lacI}* strains. As an additional internal control, we added a *lacZ* mutant.

by:

[L394] To study the potential correlation between the bacterial growth fitness and the level of *lacZYA* repression, we infected 129S6/SvEvTac mice with a 1:1:1:1 mixture of barcoded versions of the *lacI*-deficient strain, WT, *GTG_{lacI}*, *ATG_{lacI}* and the *lacZ* mutant as an internal control.

16. Line 430: Missing a).

Response: The correction has been added.

17. Fig. 4C: Y-axis should be O.D. rather than D.O.

Response: The correction has been added

References

- Adams, J. M., & Capecchi, M. R. (1966). N-formylmethionyl-sRNA as the initiator of protein synthesis. *Proceedings of the National Academy of Sciences*, 55(1), 147-155.
- Beyreuther, K., Adler, K., Geisler, N., & Klemm, A. (1973). The amino-acid sequence of lac repressor. *Proceedings of the National Academy of Sciences*, 70(12), 3576-3580.
- Blattner, F. R., Plunkett III, G., Bloch, C. A., Perna, N. T., Burland, V., Riley, M., ... & Shao, Y. (1997). The complete genome sequence of Escherichia coli K-12. *science*, 277(5331), 1453-1462.
- Clark, B. F. C., & Marcker, K. A. (1966). The role of N-formyl-methionyl-sRNA in protein biosynthesis. *Journal of molecular biology*, 17(2), 394-IN7.
- Datsenko, K. A., & Wanner, B. L. (2000). One-step inactivation of chromosomal genes in Escherichia coli K-12 using PCR products. *Proceedings of the National Academy of Sciences*, 97(12), 6640-6645.
- Elowitz, M. B., Levine, A. J., Siggia, E. D., & Swain, P. S. (2002). Stochastic gene expression in a single cell. *Science*, 297(5584), 1183-1186.

- Farabaugh, P. J. (1978). Sequence of the *lacI* gene. *Nature*, 274(5673), 765-767.
- Lewis, M., Chang, G., Horton, N. C., Kercher, M. A., Pace, H. C., Schumacher, M. A., ... & Lu, P. (1996). Crystal structure of the lactose operon repressor and its complexes with DNA and inducer. *Science*, 271(5253), 1247-1254.
- Link, A. J., Phillips, D., & Church, G. M. (1997). Methods for generating precise deletions and insertions in the genome of wild-type *Escherichia coli*: application to open reading frame characterization. *Journal of bacteriology*, 179(20), 6228-6237.
- Milón, P., & Rodnina, M. V. (2012). Kinetic control of translation initiation in bacteria. *Critical reviews in biochemistry and molecular biology*, 47(4), 334-348.

- Ozbudak, E. M., Thattai, M., Kurtser, I., Grossman, A. D., & Van Oudenaarden, A. (2002). Regulation of noise in the expression of a single gene. *Nature genetics*, 31(1), 69-73.
- van Hoek, M., & Hogeweg, P. (2007). The effect of stochasticity on the lac operon: an evolutionary perspective. *PLoS computational biology*, 3(6), e111.

Reviewer #2 (Remarks to the Author):

The manuscript “A non-canonical start codon in the lactose utilisation repressor provides a context-dependent growth advantage to a commensal *E. coli* in the murine Gut” by Cherrak and colleagues demonstrates that the start codon GTG in genes encoding for repressors of genes involved in the utilisation of inconsistently available nutrients can be selectively advantageous. The authors also provide evidence that the selective advantage could stem from a shortened lag time in utilising the nutrients compared to species with stricter repression. The figures are generally well-presented, and the manuscript reads clearly. In general, the study is innovative and will be of interest to a broad readership as it spans microbial genetics, metabolism and ecology.

Response: We thank referee#2 for investing time in carefully reviewing our manuscript and appreciate the enthusiasm expressed. We hope our detailed point-by-point responses will be sufficient to strengthen our claim and support publication of our work.

1. Main comments

Lines 167-169: “[...] attributable to selection and not spontaneous random mutations”. Please rephrase. Spontaneous random mutations also precede selection processes as mutations generate the genetic diversity for natural selection to act on. Perhaps the authors meant “attributable to selection and not genetic drift”? Also, it remains unclear why a GTG genotype frequency of only >1% would already indicate underlying natural selection. Please elaborate on why even low frequencies of GTG suggest selection.

Response: We apologize to referee#2 for any confusion arising from this section. We confirm that our intended phrase was “attributable to selection and not genetic drift”. To further strengthen this argument, we have analysed the distribution of the *lacI* ATG/GTG start codon variant in a subset of *E. coli* genomes. For this purpose, we have randomly selected 76 genomes displaying exclusively a *lacI* GTG start codon and 26 genomes predicted to encode for a *lacI* ATG start codon. Notably, we found that genomes with a

predicted *lacI* ATG start codons were non-monophyletic but rather spread throughout the phylogenetic diversity. We concluded that the ATG and GTG variation in the *lacI* gene in *E. coli* is attributable to selection (and not genetic drift). The phylogenetic tree has been added as Extended Data Fig. 1B. We have modified the result section and replaced:

Out of the 32 regulator genes subjected to analysis, 13 featured non-ATG start codons in >1 % of all *E. coli* strains, which suggests that this is attributable to selection and not spontaneous random mutations (Figure 1A).

by:

[L190] *Genomes with a predicted lacI ATG start codon were spread throughout the phylogenetic diversity within a subset of E. coli strains (Extended Data Fig. 1B), suggesting that the preference for the GTG or ATG start codon in lacI is attributable to selection and not genetic drift.*

2. Evidence for the selective advantage of the non-canonical start codons to be uniquely tied to carbohydrate utilisation is currently weak. No formal statistical analysis was performed (Fig 6), and critically, only three non-carbohydrate-related global transcription regulators have been chosen to compare against the many local and global transcription regulators of carbohydrate-related genes. This line of evidence could be strengthened by including more transcription factors not involved in carbohydrate utilisation. Characterisation of regulator genes into activators, repressors and bi-directional regulators could reveal important patterns not yet addressed in the text.

Response: We thank referee#2 for recommending us to improve our statistical analyses, thereby strengthening our claim that regulators of carbohydrate utilization operons show a particularly high frequency of non-ATG start codons. In that perspective, we have selected additional carbohydrate-unrelated regulators (based on the RegulonDB database) and increased their numbers from 3 to 29. They are now being compared to 32 carbohydrate-related regulators, throughout the *Enterobacteriaceae* family. Notably, our analysis confirmed a significant enrichment of non-ATG start codons in carbohydrate-related regulators (Fig 6.B, C and Extended Data Fig 6A). This observation remained consistent across positive, negative, and bidirectional regulators (Extended Fig 6B).

We have incorporated the additional analysis in Fig. 6B, 6C, Extended Data Fig 6A and 6B. We updated the text accordingly and replaced:

Upon investigating the occurrence of non-ATG start codons at the Enterobacteriaceae family level, we found that most metabolic regulator genes displayed alternative start codons in at least some strains of most species. Amongst them, lacI was shown to have a moderate occurrence of GTG start codons, offering further support for the contextual and strain-dependent benefit of this mutation. The majority of the metabolic regulators examined in this study exhibited non-ATG start codons only in a subset of strains within specific genera. This observation suggested that various selective pressures play a role in determining the evolutionary choice between ATG and non-ATG start codons in most of the studied carbohydrate utilization regulators. However, this pattern did not apply to the global regulators cra and mlc, as they are almost exclusively initiated by the GTG-start codon (Fig. 6B). To confirm that this distinctive pattern was specific to global carbohydrate utilization regulators, we carried out an analysis of the start codon in additional global regulators that do

not appear to be directly involved in carbohydrate metabolism such as fur, lrp and oxyR. In contrast to cra and mlc, we observed that non-canonical start codons were largely absent amongst the fur, lrp and oxyR regulators encoded across the Enterobacteriaceae family (Fig. 6B). Taken together, our findings indicated that alternative start codons can have a broader fitness spectrum that is not limited to lactose utilization in E. coli 8178 but can be expanded to carbohydrate metabolic pathways within the Enterobacteriaceae family.

by:

[L442] Upon scrutinizing the occurrence of non-ATG start codons at the *Enterobacteriaceae* family level, we found that all examined metabolic regulator genes displayed alternative start codons in at least a subset of strains within specific genera. Notably, non-canonical start codons were further enriched in *cra* and *mlc*, starting exclusively with a GTG start codon in almost all the genera analysed (Fig. 6B). Given the potential benefit associated with non-canonical start codons in carbohydrate metabolism, we next wondered whether such distribution would display a similar pattern among carbohydrate-unrelated regulators. To explore this, we carried out an analysis of the start codons in 29 additional regulators that do not appear to be exclusively involved in carbohydrate metabolism (Table S2). The overall proportion of non-canonical start codons was significantly lower ($P = 1.5 \times 10^{-11}$) in carbohydrate-unrelated compared to carbohydrate-related regulators (Fig. 6C, Extended Data Fig. 6A). This trend persisted across positive, negative, and bidirectional regulators (Extended Data Fig. 6B). Taken together, these observations suggest that non-canonical start codons have a wider range of functional advantages beyond lactose utilization in *E. coli* 8178, extending to carbohydrate metabolic pathways within the *Enterobacteriaceae* family.

3. Figure 3C) Error bar function is not described, and the bar heights represent the mean in this case, not the median, which should be mentioned in the figure's legend. The same applies to Figures 2B and 4C (mean is explained, but not the error bars). CD) The order of the diets should be the same between the graphs and would read easier if D was arranged by increasing/decreasing lactose concentration.

Response: We thank referee#2 for highlighting the missing information and apologize for any lack of clarity. We have now incorporated the following details in the figure captions:

[L899] Figure 3D: Lactose proportion is heterogeneous across diets. SPF and germ-free facility chows ($n = 8$ pieces) were subjected to lactose measurement assays. The mean and error (standard deviation) are represented for each diet.

[L870] Figure 2B: Activities are the mean of 3 independent experiments. Error bars: standard deviation.

[L920] Figure 4C: The *gfp* signal resulting from *lacZ* expression at mid-exponential phase represents the mean of a biological triplicate and is expressed as the mean fluorescence intensity (MFI) normalized by the bacterial density (optical density unit: $u.O.D_{600}$). Error bars: standard deviation

We have also modified Figure 3D and E by organizing the various diets in decreasing order of lactose concentration.

4. Figure 4B and 4D: The legend says that the growth curves display the mean OD and error. The error bars are only displayed for some of the data points, and it is not mentioned which error function is used (SE, SD, CI?). Each growth curve is the mean of only three independent experiments. Thus, I suggest showing each individual growth curve in addition to the mean (perhaps the three individual curves in muted colours and mean curves in saturated colours).

Response: After several attempts, we found that displaying individual growth curves for each replicate across all four different strains lacked clarity. Alternatively, we have now

substituted the symbols and represented the mean curve with the standard deviation. We additionally corrected the correspond Figure 4B and 4D captions and replaced:

[L916] Figure 4B: *The mean and error (standard deviation) of a biological triplicate are represented*

[L920] Figure 4C: *The *gfp* signal resulting from *lacZ* expression at mid-exponential phase represents the mean of a biological triplicate and is expressed as the mean fluorescence intensity (MFI) normalized by the bacterial density (optical density unit: *u.O.D*₆₀₀). Error bars: standard deviation.*

In addition, we would like to point out, that we are depositing all primary data “open access” for further re-analysis by others. This would allow any interested reader to perform further tests directly on the primary datasets.

5. Figure 4C: There is no statistical analysis, although the results section (lines 310-320) describes significant results from group comparisons.

Response: We have now included the previously missing statistical analysis for Figure 4C. This analysis clearly demonstrates that the WT and GTG_{lacI} strains display a significantly higher *lacZ-gfp* signal than the ATG_{lacI} strain. However, we would like to bring to the referee’s attention that this difference is resolved at a higher IPTG concentration (500µM), indicating that this concentration is sufficient to alleviate the *lacZYA* repression in the ATG_{lacI} strain. We have adjusted the results section and replaced:

*In contrast to *lacI*, expression *lacZYA* reporter in the ATG_{lacI} mutant was significantly lower, regardless of the IPTG concentration (Figure 4C). Notably, we observed that *gfp* signal resulting from the *lacZYA* operon transcription in the WT and GTG_{lacI} strains was overall higher than the ATG_{lacI} mutant, even in the absence of IPTG. Furthermore, a slight increase in the IPTG concentration (5 mM) was sufficient for the WT and GTG_{lacI} strains to achieve a similar high level of the *lacZYA* expression as the *lacI* mutant. From this experiment, we concluded that utilization of the GTG start codon in *lacI* was sufficient to enhance the sensitivity and basal expression of the *lacZYA* operon (Figure 4C).*

by:

[L320] *At the highest IPTG concentration tested (500 µM), the WT, GTG_{lacI} and ATG_{lacI} strains displayed reporter signals similar to the *lacI* background, indicating that the full derepression*

of the lacZYA operon is achieved under this condition. Notably, the gfp signal resulting from the lacZYA operon transcription in the WT and GTG_{lacI} strains was consistently higher than in the ATG_{lacI} mutant at any tested intermediate IPTG concentration (1; 5; 20; 100 μ M). This trend held true even in the absence of IPTG, demonstrating that the utilization of the GTG start codon in lacI was sufficient to enhance the sensitivity and basal expression of the lacZYA operon (Fig. 4C).

6. Lines 235-236: The term “diet” is misleading as results show this is still related to lactose concentration.

Response: We thank referee#2 for raising this issue and are happy to provide a clarification. In the course of our study, we have observed that different diets can lead to divergent observations regarding the contribution of lactose to *E. coli* 8178 growth *in vivo*. Additional experiments demonstrated that the observed discrepancies were a result of the varying amount of lactose present in the tested diets which we stated in the text [L286] as: *These findings indicate that variation in lactose content between different diets can lead to different bacterial fitness outcomes when studying gut colonization phenotypes.*

Although these variations do indeed result from distinct lactose concentrations, as the lactose levels differ depending on the diet, we believe that the following title *The in vivo lactose-dependent fitness of E. coli 8178 is affected by the host microbiota and diet* nicely relates to our findings. To avoid any misunderstanding, we have replaced it now by:

[L234] *The in vivo lactose-dependent fitness of E. coli 8178 is affected by the host microbiota and diet composition.*

7. Fig 2A: Please change/add the unit “kDa” (with a lowercase “a”) for the three numbers (50 kDa, 37 kDa, 38.6 kDa). The same applies to Fig 4A.

Response: This has been corrected

8. Fig 2E: The significance of the two genotype groups to the WT levels should also be added (as it is done in Fig 2D).

Response: We have added statistical analysis.

9. Line 768: Typo. Remove ‘s’ from “codons”.

Response: The correction has been made.

Reviewer #3 (Remarks to the Author):

In this manuscript the authors explore how the fine tuning of gene expression in the gut can influence the competitive ability of a commensal strain of *Escherichia coli*.

Importantly, they focus on the repressor of a well-known operon, the lac operon. They hypothesize, that the non-canonical GTG start codon in lacI, which is close to universal

40

among E. coli strains, confers the ideal amount of expression to colonize the gut. They do this by comparing its competitive ability with the canonical ATG version of lacI and with the constitutive lacI mutant. By colonizing mice with a mixture of 5 genotypes (lacI, WT, GTGlacl, ATGlacl and lacZ) they follow their frequencies for 1 or 3 (?) days. Since the WT and GTGlacl remain the most frequent they conclude they display the right amount of expression to subsist in the presence of lactose.

The manuscript is clearly written and the experiments are well thought and carefully designed.

Response: We would like to express our gratitude to referee#3 for dedicating time carefully reviewing our manuscript. We hope the revised manuscript and subsequent point-by-point answers will be sufficient to strengthen our claims and support publication of our work.

Main concerns:

1. Throughout the manuscript, the authors use the words “infection” for the colonization process, however they don’t report any infection symptoms in recipient mice. In case no symptoms of infection are present in mice, I believe the term “colonization” better describes the procedure.

Response: We thank referee#3 for bringing this up and have now changed the words “infection”, “infected” and “infect” throughout the entire manuscript.

2. The authors say *E. coli* 8178 relies on lactose metabolism to better colonize the gut. However, Fig 3D shows that *lacZ* is neutral when mice are under a diet devoid of lactose. Therefore, this is not only true in the presence of lactose (Fig 3D).

Response: The initial assessment of *E. coli* 8178’s ability to consume lactose in the murine gut began with the utilization of a standard diet and a commonly employed specific pathogen-free (SPF) mouse model. Under these conditions, we found a competitive index < 1 for the *E. coli* 8178 *lacZ* mutant. From that, we concluded that *E. coli* 8178 consumes lactose to efficiently colonize the mouse gut, which is stated in our manuscript – Section 2 of our Results. Nonetheless, upon changing the mouse model or dietary conditions, we noted variations in the lactose-dependence growth of *E. coli* 8178. This forced us to re-adjust our interpretation and state, from Results-Section 3 until the end of the manuscript, that *E. coli* 8178’s reliance on lactose is context-dependent. We understand the potential source of confusion as Figure 3E (formerly Fig.3D in the previous version) depicts a slightly different outcome than the preceding section. We have now changed the conclusion of the section 2 to emphasize the context under which *E. coli* 8178 relies on lactose. Specifically, we have replaced:

Overall, these data showed that the lac-operon of E. coli 8178 is functional and facilitates gut-luminal growth in the murine gut.

by:

[L231] Overall, these data demonstrate that the *lacZYA* operon of *E. coli* 8178 functions effectively, promoting gut luminal growth in the murine gut under these conditions.

3. pg 8 L248, the authors say *E. coli* 8178 relied on lactose metabolism in C57BL/6, despite the competitive defect of *lacZ* being the same with or without lactose added to the food.

Response: The data presented on Figure 3B showed that the WT *E. coli* 8178 strain outperforms the isogenic *lacZ* mutant by 10-fold in C57BL/6 mice. This observation,

regardless of the effect of lactose supplementation, is sufficient to state that *E. coli* 8178 utilizes lactose for proficient growth in the murine gut of C57BL/6 mice. However, as the referee rightly pointed out, the competitive index of the *lacZ* mutant remains constant in this mouse model, even in the presence of lactose (3%) in the drinking water. We believe that the unchanged *lacZ* fitness witnessed in the 3% lactose supplemented condition is reflective of the host and/or microbiota effect on the intestinal lactose availability. More precisely, we think that, unlike 129S6/SvEvTac mice, the supplementation with a 3% lactose solution to C57BL/6 mice may not be substantial enough to increase the amount of available lactose in the gut and further foster the growth of the WT over the *lacZ* mutant. To test this hypothesis, we elevated the concentration of lactose in the drinking water of inoculated C57BL/6 mice to 5% and 8% w/v (which remains in the range of lactose concentrations found in mammalian milk). We found that the competitive index of the *lacZ* mutant was further exacerbated in the 8% lactose supplemented group. Notably, the fitness defect of the *lacZ* mutant under this condition (C.I.: ~ 0.01) is comparable to the defect observed in inoculated 129S6/SvEvTac mice supplemented with 3% lactose in the drinking water. This indicated that the impact of lactose supplementation varies depending on the mouse line, with 129S6/SvEvTac exhibiting a higher responsiveness compared to C57BL/6 mice.

We have illustrated our new findings in Fig. 3C and updated Extended Data Fig. 3A. Additionally, we have replaced,

However, the addition of lactose (3%) to the drinking water did not exacerbate the competitiveness of the WT over the *lacZ* strain in this mouse model, demonstrating a different response to lactose supplementation in C57BL/6 compared to 129S6/SvEvTac mice.

by:

[L251] However, the addition of lactose (3%) to the drinking water did not exacerbate the competitiveness of the WT over the *lacZ* strain in this mouse model. This was achieved at a higher lactose concentration (8%), demonstrating a different response to lactose supplementation in C57BL/6 compared to 129S6/SvEvTac mice (Fig. 3B).

4. The authors observe that *lacZ* is deleterious in C57BL/6 GF with or without added lactose. In contrast, Pinto saw that *lacZ* was neutral in GF animals, irrespectively of adding or not lactose to the diet. This could be related either 1) to the different strain of *E. coli* (Pinto et al. used MG1655) or 2) to the fact that the authors did not treat GF mice with streptomycin, whereas Pinto et al. did.

Antibiotics can affect the host independently of the microbiota and therefore treating SPF animals with streptomycin could eventually have led to host modification that could impact lac competitions. (Fig 3B) and finally, 3) animals are transferred to SPF facility on day 0, whereas in Pinto et al. experiment, animals were kept in the GF facility. Transferring GF mice monocolonized to SPF facility could allow further uncontrolled colonization with foreigner microorganisms.

Response: We thank referee#3 for pointing out key differences between our study and the one described in Pinto et al., 2023. We agree that many parameters may explain the

disparate outcomes observed, especially upon colonizing germ-free mice. Notably, (1) the hygienic conditions but also (2) the colonization setup and (3) model organism are crucial factors, and we would like to take this opportunity to elaborate separately on each of these plausible explanations:

- (1) Hygienic condition: To mitigate any risk of contamination, mouse inoculation cannot be conducted within our existing germ-free animal facility. Instead, germ-free animals are transferred to our SPF facility. Upon their arrival, germ-free animals are promptly infected and placed, throughout the entire experiment, into BCU-2 sealed negative pressure individually ventilated cages that maintains sterile conditions. This procedure was established to minimize the risk of contamination and to date, we have not observed any instances of contamination under these conditions (Gül et al., 2023; Laganenka et al., 2023; Hoces et al., 2022 Hausmann et al., 2020; Nguyen et al., 2020).
- (2) *In vivo* setup: In addition to the hygienic condition, referee#3 nicely pointed out the absence of antibiotic treatment upon inoculating germ-free mice. Antibiotic treatment is typically employed to disrupt the gut microbiota and foster intestinal colonization by exogenous bacterial species. As germ-free mice inherently lack any microbes, they are naturally highly permissive to bacterial colonization. Consequently, in our condition, germ-free animals are not pretreated with antibiotics. However, we recognize that the use of antibiotics in Pinto et al may contribute to divergent outcomes when compared to our study. To test this hypothesis, we inoculated streptomycin-pretreated germ-free with a 1:1 mixture of the *E. coli* 8178 WT and *lacZ* mutant. When subjected to lactose (+) condition (standard diet + 3% lactose in the drinking water), we found that the *E. coli* 8178 WT strain displayed a competitive advantage over the *lacZ* mutant that mirrored our findings in untreated germ-free mice. This indicates that the lactose-dependent fitness of *E. coli* 8178 in germ-free animals is not affected by the potential effect of streptomycin on the host. Additionally, the consistent fitness disadvantage of the *lacZ* mutant in germ-free mice, regardless of antibiotic pretreatment, allows us to exclude the hygienic hypothesis as any potential bacterial contaminants would have been eliminated by the antibiotic.
- (3) Finally, among the plausible explanations listed, we believe that the divergent results between our study and Pinto et al., primarily arise from the use of different model organisms. Indeed, several studies using different *E. coli* strains (EPEC, MG1655, UPEC, Nissle) have shown variable *lacZ*-associated fitness, indicating a strain-specific dependence on lactose for *in vivo* growth (Fabich *et al.*, 2008; Conway & Cohen; 2015).

We have amended the manuscript with the new control data resulting from our

46streptomycin-pretreated germ-free mice experiment. The new figures are presented in Extended Data Fig. 3C-D and the Result section has been modified accordingly. Specifically, we have changed:

In order to explore the potential role of the gut microbiota in the lactose-dependent fitness defect of a lacZ mutant, we used germ-free C57BL/6 mice (Figure 3A). Inoculated germ-free C57BL/6 mice displayed a competitive defect of the lacZ mutant which, in contrast to SPF colonized C57BL/6 animals, was significantly exacerbated upon lactose supplementation (Fig.

3B, orange symbols, Extended Data Fig. 3B). Combined, these observations demonstrate that *E. coli* 8178's reliance on lactose metabolism varies in a microbiota-dependent manner.

by:

[L255] *In order to explore the potential role of the gut microbiota in the lactose-dependent fitness defect of a lacZ mutant, we used germ-free C57BL/6 mice (Fig. 3A). Inoculated germ-free C57BL/6 mice displayed a competitive defect of the lacZ mutant which, in contrast to SPF colonized C57BL/6 animals, was significantly exacerbated upon 3% lactose supplementation (Fig. 3B, orange symbols, Extended Data Fig. 3B). As opposed to SPF C57BL/6 mice, germ-free mice are inherently permissive to gut colonization and, as a result, were not subjected to antibiotic pretreatment. To verify that the difference in the lacZ-associated fitness between germ-free and antibiotic-pretreated SPF C57BL/6 mice arises from the microbiota rather than the antibiotic effect on the host, we evaluated the competitive index of the lacZ mutant in streptomycin-pretreated germ-free mice. As expected, the competitive index of the lacZ mutant remained unchanged between antibiotic-pretreated and non-pretreated germ-free animals when exposed to 3% lactose in the drinking water (Extended Data Fig. 3C, D). Combined, these observations demonstrate that E. coli 8178's reliance on lactose metabolism varies in a microbiota-dependent manner.*

Additionally, we have further emphasized the strain-specific nature of the *lacZ*-associated phenotype *in vivo* and have replaced in the discussion section:

This conditional importance may result from lactose heterogeneity in diets and could thus provide an explanation for the varying outcomes observed in studies examining the importance of lactose in E. coli's intestinal colonization.

by:

[L521] *This conditional importance may result from lactose heterogeneity in diets which, along with the variations in the model organism, could explain distinct outcomes observed in studies examining the importance of lactose in E. coli's intestinal colonization.*

5. The authors do a colonization where they start with a mixture of 1:1:1:1 of barcoded versions of the *lacI*-deficient strain, WT, GTG*lacI*, and ATG*lacI* strains plus Δ *lacZ*. They then estimate the competitive index of each of these genotypes against the wt, both in the presence and absence of lactose. While the wt has the highest competitive ability in the presence of lactose, it is important to note that they all start at a relatively high frequency (20%), and that other constitutive mutants (e.g. srlR, Fig 6 in

48

DOI:10.1371/journal.pgen.1006420 or gatR, fig 5 in doi:10.1093/molbev/msx221) have shown to be deleterious at high frequency (and beneficial at low frequency), but were still able to coexist with the regulated version of the operon. Therefore, it looks a bit simplistic to say that lacI is deleterious. It would be important to follow the colonization with the 5 genotypes for some weeks and see which mutants (including the lacI) get extinct and which are able to persist. It is possible that such an experiment would reveal that no single optimal expression of the lac operon is uniquely beneficial but rather several levels of expression can coexist.

Response: We thank referee#3 for highlighting the significance of the initial mutant frequency and experiment duration in our conclusions. We agree that modulating these

factors may contribute to a more comprehensive understanding of the *in vivo* fitness of our various mutants, which we achieved in the revised manuscript.

Given that the primary focus of our manuscript centers on the *lacI* non-canonical start codon, we were particularly curious about the long-term dynamics of the ATG_{*lacI*} and WT strains. To answer this question, we prolonged the experiments depicted in Fig 5A from 3 days to 2 weeks, a timeframe during which *E. coli* 8178 can stably colonize the microbiota of streptomycin-pretreated 129/SvEvS6Tac. Beyond this period, the intestinal load of *E. coli* 8178 drastically drops, presumably due to the replenishment of the gut microbiota. Following its growth defect reported at days 1-3, we observed that the ATG_{*lacI*} strain remains stable from day 6- until day 14-post-inoculation. This indicates that the ATG_{*lacI*} mutant did not go extinct but rather coexisted (albeit at lower proportion) with the WT strain. Similar experiments were conducted under lactose-limited conditions (germ-free facility diet) or using the GTG_{*lacI*} strain, which demonstrated that this pattern resulted solely from the *lacI* start codon sequence and depended on lactose.

The new data for days 6-10-14 have been added as Extended Data Fig. 5A-B. Additionally, we have completed the results section and added:

[L363] *To gain a more comprehensive understanding of the prolonged impact of the lacI ATG start codon mutation, we extended the duration of the experiments to two weeks (Extended Data Fig. 5A). Beyond this period, the intestinal load of E. coli 8178 drastically drops, presumably due to the replenishment of the gut microbiota (Extended Data Fig. 5B). Although facing a significant fitness disadvantage in the early stages (days 1-3), the ATG_{lacI} mutant did not go extinct but rather stably coexisted (albeit at a lower proportion) with the WT strain (Extended Data Fig. 5A). In contrast, the complemented GTG_{lacI} mutant had a similar competitive fitness as the WT strain across the 2-week experiment (Fig. 5A, Extended Data Fig. 5A, B).*

As rightly pointed out by referee#3, we next investigated to which extent the nature of the *lacI* start codon can be beneficial at a lower frequency. For this aim, we inoculated streptomycin-pretreated 129S6/SvEvTac mice with a mixture containing the WT and ATG_{*lacI*} strains to a ratio of 0.001:1 (WT diluted to 1/1000). This setup, recently developed in the laboratory (Gül et al., 2023), enabled us to monitor the *in vivo* dynamics of the diluted strain which, we anticipated, would catch up the diluting strain through a more efficient lactose metabolism. In presence of lactose in the drinking water (3%) and throughout the 2-week experiment, we found that the WT strain can gradually catch up the ATG_{*lacI*} mutant. This emphasizes the benefit associated to the *lacI* GTG start codon, even at a lower frequency. In parallel, we have inoculated a second group of mice with the WT and ATG_{*lacI*} strains to a ratio of 1:0.001 (ATG_{*lacI*} diluted to 1/1000). In that context, the ATG_{*lacI*} mutant did not catch up the WT strain. More interestingly, the WT-to-ATG_{*lacI*}

strain ratio remains consistently close to ~ 1000 throughout the 2-week experiment. This suggests that the ATG mutation on the *lacI* start codon is not detrimental but rather appears to be neutral at a lower mutant frequency, under these conditions.

The outcome of this new experiment is depicted in Figure 5B and Extended Data Fig. 5C-D. We have changed the adjusted the Result section and added:

[L377] *We next explored how the initial mutant frequency influences the impact of the lacI ATG start codon mutation. This was motivated by prior studies that have demonstrated frequency-dependent selection of different gene regulators (Lourenço et al., 2016; Sousa et al., 2017;*

Turner et al., 2020). For this aim, we inoculated streptomycin-pretreated 129S6/SvEvTac mice with a mixture of the WT and ATG_{lacI} strains at varying ratios and evaluated the mutant fitness over a 2-week experiment course in presence of lactose (3 %) in the drinking water. When placed in excess, the ATG_{lacI} mutant was progressively outnumbered by the WT strain, highlighting a fitness disadvantage conferred by the ATG lacI start codon even at a high frequency (Fig. 5B, Extended Data Fig. 5C, D). Interestingly, the fitness of the ATG_{lacI} mutant remained unchanged when diluted a thousand-fold with the WT strain. This indicated that this mutation, at a lower frequency, appears to be no longer detrimental under our conditions (Fig. 5B, Extended Data Fig. 5C, D).

Finally, to strengthen our claim on the *in vivo* benefit associated to the GTG start codon, we have additionally extended the experiment presented in Fig. 5B from 3 days to 2 weeks. In presence of lactose, the *lacI*, *lacZ* and ATG_{lacI} mutant remained consistently low. This aligned with our previous observations and demonstrated the stable coexistence of multiple lactose metabolism variants (namely *lacI*, WT and ATG_{lacI}) at different ratios, with the WT dominating in our conditions.

The day 6-10-14 have been added as Extended Data Fig. 5F. We have additionally stated it in the Result as follows:

[L411] Remarkably, the fitness of the *lacI* and ATG_{lacI} mutants stabilized throughout the 2-week duration of the experiment, highlighting the coexistence of multiple lactose-consuming variants, with the WT prevailing under our conditions. Contrasting with the lactose-supplemented group, the fitness defect of the ATG_{lacI} strain was less pronounced under lactose-limited conditions (Fig. 5C, Extended Data Fig. 5F). Taken together, these observations highlight the frequency- and context-dependent metabolic benefit attributed to the *lacI* start codon in *E. coli* 8178.

Despite offering insights into the long-term fitness associated with the *lacI* start codon sequence and emphasizing the coexistence of multiple *lac* variants, we acknowledge that our data remains preliminary. Indeed, with our current mouse model, we were constrained to 2-weeks experiments, as beyond this period, *E. coli* 8178 gets displaced by the replenishment of the gut microbiota (Extended Data Fig. 5D). We would like to bring to the referee's attention that assessing the dynamic of *E. coli* 8178 variants over a longer period requires the utilization of alternative *in vivo* models (i.e. gnotobiotic, antibiotic-supplemented animals) for which we currently lack background knowledge. Although extremely relevant, we believe that setting up long-term *E. coli* 8178 inoculation experiments requires a substantial amount of work that will unfortunately extend beyond the timeframe of the revision.

6. From fig 4B and D it looks like wt, ATGlacl, GTGlacl and lacl have similar growth rates in the presence of lactose, the main differences potentially being related with the bacterial readiness to consume lactose. Other experiments (Fig 3 in doi:10.1093/molbev/msx221) have shown polymorphism to be selected in expression of the gat operon, where the two phenotypes (gat-neg and gat-pos) coexist without fixation of any of the two. In fact, when the authors tried to fix any of the two, the other would always emerge de novo. Even in the case of the lac operon, constitutive mutants have been observed to be selected in the gut. So, in sum, I think it is fair to suspect that the

complex and ever-changing environment of the gut will favour polymorphism in detriment of a single level of operon expression, depending on the microbiota colonizing the gut as well as on host diet.

Response: This is a highly pertinent point, and we thank referee#3 for giving us the opportunity to expand on it. In the previous manuscript version, we found that upon colonizing streptomycin-pretreated 129S6/SvEvTac mice with various lactose metabolism variants (ATG_{lacI}, *lacI* and *lacZ*), all of them faced a significant growth disadvantage in the initial 3 days. Contrary to our expectation of these mutants going extinct as the experiments progressed, we observed, in our extended time courses, that the *lacI* and ATG_{lacI} strains stabilized overtime. This allowed them to coexist (albeit at a lower proportion) with the WT strain for at least 2 weeks. In line with this, the ATG_{lacI} mutant remained unaffected when diluted to a thousand times with the WT strain. These findings underscore the stable (2-week long) co-existence of different *lac* variants, with the WT dominating under our conditions, due to its ability to adapt faster to lactose metabolism without suffering the cost of a constitutive *lacZYA* expression.

We agree with referee#3 that this polymorphism, which has become visible only in the revised manuscript, may have been selected to easily adapt to the highly complex and dynamic environment encountered in the gut microbiota. For instance, under lactose-limited conditions, the proportion of the *lacI* and ATG_{lacI} strains in regard to the WT are considerably different. We anticipate further modulations as the intestinal environment changes, such as the *lacI* variant potentially dominating in neonates where lactose is the main carbohydrate (Ghalayini et al., 2019). Regarding the ATG_{lacI} mutation, it is not yet clear under which conditions it can be favoured, but its low occurrence across *E. coli* strains suggest few instances where this can happen.

We have now emphasized the polymorphism observed by adding the following paragraph in the discussion.

[L544] *The advantage associated with such faster metabolic adaptation was primarily visible in the early stages of gut colonization (days 1-3 post inoculation). Beyond this period, the ATG_{lacI} strain did not go extinct but rather persisted at a stable proportion (Extended Data Fig. 5A). When coupled with the observed neutral fitness of the ATG_{lacI} mutant at a lower proportion (Fig. 5B, Extended Data Fig. 5C), we concluded that the lacI ATG start codon is subjected to a frequency-dependent selection. This phenomenon, which was similarly observed for the galactitol and sorbitol metabolism (Lourenço et al., 2016; Sousa et al., 2017), may provide a reasonable explanation for the relatively low frequency (1 %) of ATG lacI start codon observed in the analysed E. coli genomes (Fig. 1). Furthermore, the observed long-term coexistence of the WT and ATG_{lacI}, but also lacI strains indicated the maintenance of a polymorphism.*

Although the WT predominates under our conditions, the simultaneous presence of different lactose metabolism variants, albeit at different proportions, concurs with recent findings highlighting the coexistence of different genotypes through negative frequency-dependent selection (Delph & Kelly, 2014; Sousa et al., 2017; Kurbalija Novičić et al., 2020; Turner et al., 2020; Christie & McNickle, 2023). This suggests that, beyond gene regulation, the long-term preservation of different lactose metabolism variants may play an important role in the bacterial adaptation to the fluctuating conditions of the gastrointestinal tract.

7. Minor:

- Pg 8 L265 replace "where there" by "when they"
- Abstract, L33 replace "abondance" by "abundance".
- Pg 14 L541, Galactitol repressor is gatR and not galR

Response: The corrections have been made.

Ana Margarida Sousa

References

- Conway, T., & Cohen, P. S. (2015a). Commensal and pathogenic *Escherichia coli* metabolism in the gut. *Metabolism and bacterial pathogenesis*, 343-362.
- Fabich, A. J., Jones, S. A., Chowdhury, F. Z., Cernosek, A., Anderson, A., Smalley, D., ... & Conway, T. (2008). Comparison of carbon nutrition for pathogenic and commensal *Escherichia coli* strains in the mouse intestine. *Infection and immunity*, 76(3), 1143-1152.
- Ghalayini, M., Magnan, M., Dion, S., Zatout, O., Bourguignon, L., Tenailon, O., & Lescat, M. (2019). Long-term evolution of the natural isolate of *Escherichia coli* 536 in the mouse gut colonized after maternal transmission reveals convergence in the constitutive expression of the lactose operon. *Molecular Ecology*, 28(19), 4470-4485.
- Gül, E., Abi Younes, A., Huuskonen, J., Diawara, C., Nguyen, B. D., Maurer, L., ... & Hardt, W. D. (2023). Differences in carbon metabolic capacity fuel co-existence and plasmid transfer between *Salmonella* strains in the mouse gut. *Cell Host & Microbe*.

Decision Letter, first revision:

1Message: Our ref: NMICROBIOL-23102600A

10th June 2024

Dear Dr. Hardt,

Thank you for your patience as we've prepared the guidelines for final submission of your Nature Microbiology manuscript, "A non-canonical start codon in the lactose utilization repressor provides a context-dependent growth advantage to a commensal E. coli in the mouse gut" (NMICROBIOL-23102600A). Please carefully follow the step-by-step instructions provided in the attached file, and add a response in each row of the table to indicate the changes that you have made. Please also check and comment on any additional marked-up edits we have proposed within the text. Ensuring that each point is addressed will help to ensure that your revised manuscript can be swiftly handed over to our production team.

In recognition of the time and expertise our reviewers provide to Nature Microbiology's editorial process, we would like to formally acknowledge their contribution to the external peer review of your manuscript entitled "A non-canonical start codon in the lactose utilization repressor provides a context-dependent growth advantage to a commensal E. coli in the mouse gut". For those reviewers who give their assent, we will be publishing their names alongside the published article.

Nature Microbiology offers a Transparent Peer Review option for new original research manuscripts submitted after December 1st, 2019. As part of this initiative, we encourage our authors to support increased transparency into the peer review process by agreeing to have the reviewer comments, author rebuttal letters, and editorial decision letters published as a Supplementary item. When you submit your final files please clearly state in your cover letter whether or not you would like to participate in this initiative. Please note that failure to state your preference will result in delays in accepting your manuscript for publication.

Cover suggestions

COVER ARTWORK: We welcome submissions of artwork for consideration for our cover. For more information, please see our guide for cover artwork.

2Nature Microbiology has now transitioned to a unified Rights Collection system which will allow our Author Services team to quickly and easily collect the rights and permissions required to publish your work. Approximately 10 days after your paper is formally accepted, you will receive an email in providing you with a link to complete the grant of rights. If your paper is eligible for Open Access, our Author Services team will also be in touch regarding any additional information that may be required to arrange payment for your article.

Please note that *Nature Microbiology* is a Transformative Journal (TJ). Authors may publish their research with us through the traditional subscription access route or make their paper immediately open access through payment of an article-processing charge (APC). Authors will not be required to make a final decision about access to their article until it has been accepted. Find out more about Transformative Journals

Best regards,

Reviewer #1:

Remarks to the Author:

The authors have fully addressed my previous points, and only a few small things remain to be revised, as follows:

- Point 6: The addition on L561 explaining the relevance of start codons versus promoter mutations is helpful. However, on lines 574-576 the reasoning remains unclear. The

3authors propose that, “the reduction of LacI protein concentration at the start codon sequence level provides a strong metabolic advantage which is associated with a low-noise characteristics.” In the authors’ hypothesis, I do not follow how lower noise in LacI would be tied to a metabolic advantage, after all once lactose metabolism is underway, the repressor would not be bound to the promoter.

- Point 8: (Plating bias). The explanation in the response letter regarding why the GTG strain has slightly lower counts than wild type was helpful, but it should be mentioned somewhere in the manuscript, as some readers may not realize where this systematic bias comes from.
- Looking back at the method for computing the competitive index (lines 749 - 754), it is not 100% clear how the WT c.f.u. was determined. The mutant c.f.u. is determined by plating on ampicillin, but I could not determine which plating isolated the WT. Was WT c.f.u. determined directly, or was the total bacterial c.f.u. (mutant + WT) used as normalization? It would be useful to include this information as explicitly as possible so that the study can be repeated with ease. The specific resistances of each strain need to be given in the Supplementary Tables S3 - S5. Currently, resistances are not explicitly listed anywhere in the tables.
- Extended Fig. 5D, the y-axis label is not visible.

Reviewer #2:

Remarks to the Author:

The authors addressed all my previous comments, and I have no remaining concerns or suggestions. I especially appreciate the extended analysis of the distribution of start codons in carbohydrate metabolism-related and carbohydrate metabolism-unrelated regulators.

Reviewer #3:

Remarks to the Author:

I believe the authors have made an incredible effort to address all raised concerns and have succeeded in clarifying those. I would like just to add some final remarks.

In the bottom of pg 18 the authors say: “: The data presented on Figure 3B showed that the WT E. coli 8178 strain outperforms the isogenic lacZ mutant by 10-fold in C57BL/6 mice. This observation, regardless of the effect of lactose supplementation, is sufficient to state that E. coli 8178 utilizes lactose for proficient growth in the murine gut of C57BL/6 mice.”

I still don’t think this is technically correct, as the authors don’t measure the lactose metabolism in the gut, they cannot state this. What they can say is that lacZ mutant has a growth defect in the gut, which is not the same as saying that E. coli relies on lactose metabolism to colonize the gut.

4Other reasons besides lactose metabolism, could justify the differences between the lacZ mutant and the wt. For e.g.:

1- The permease from the lacZ mutant is still active and therefore lactose (or other molecule) is still being transported by the cell, and this is costly.

2- The lac operon can be used to consume other carbon sources, such as galactosyl-glycerol, a naturally occurring galactoside DOI: 10.1093/genetics/139.1.19, possibly present in the mouse food.

The authors state by the end of pg. 19 that "any potential bacterial contaminants would have been eliminated by the antibiotic.", this is not true as streptomycin is unable to eliminate many of the gut colonizers.

Ana Margarida Sousa

Reviewer #4:

Remarks to the Author:

The authors addressed all my previous comments, and I have no remaining concerns or suggestions. I especially appreciate the extended analysis of the distribution of start codons in carbohydrate metabolism-related and carbohydrate metabolism-unrelated regulators.

Author Rebuttal, first revision:

Reviewer Comments:

Reviewer #1

1. The addition on L561 explaining the relevance of start codons versus promoter mutations is helpful. However, on lines 574-576 the reasoning remains unclear. The authors propose that, "the reduction of LacI protein concentration at the start codon sequence level provides a strong metabolic advantage which is associated with a low-noise characteristics." In the authors' hypothesis, I do not follow how lower noise in LacI would be tied to a metabolic advantage, after all once lactose metabolism is underway, the repressor would not be bound to the promoter.

Response: This specific part of the discussion was intended to address a key question about the advantage of slowing down gene expression at the translational vs transcriptional level. We suggested that the advantage for bacteria to accumulate mutations at the start codon (i.e GTG in *lacI*) might be linked to the low-noise characteristics of mutations affecting the translation process, as compared to the

5high-noise characteristics of mutations occurring at the promoter level. Overall, this remains highly speculative, and in light with the comments previously made by referees 1 and 2, and the need to shorten our manuscript, we decided to remove this part from the discussion.

2. The explanation in the response letter regarding why the GTG strain has slightly lower counts than wild type was helpful, but it should be mentioned somewhere in the manuscript, as some readers may not realize where this systematic bias comes from.

Response: We have added the explanation in the results section and replaced:

In contrast, the complemented GTG_{lacI} mutant had a similar competitive fitness as the WT strain across the 2-week experiment (Fig. 5A, Extended Data Fig. 5A, B).

By

In contrast, the complemented GTG_{lacI} mutant had a similar competitive fitness as the WT strain across the 2-week experiment (Fig. 5A, Extended Data Fig. 5A, B). It's worth noting that the load of the GTG_{lacI} strain tends to be slightly lower than the WT. This can be attributed to the growth defect associated with antibiotic-based selection of the mutant, whereas the WT count is deducted from the total load on antibiotic-free plates.

3. Looking back at the method for computing the competitive index (lines 749 - 754), it is not 100% clear how the WT c.f.u. was determined. The mutant c.f.u. is determined by plating on ampicillin, but I could not determine which plating isolated the WT. Was WT c.f.u. determined directly, or was the total bacterial c.f.u. (mutant + WT) used as normalization? It would be useful to include this information as explicitly as possible so that the study can be repeated with ease. The specific resistances of each strain need to be given in the Supplementary Tables S3 - S5. Currently, resistances are not explicitly listed anywhere in the tables.

Response: We have further clarified this point in the method section and replaced:

The bacterial load was determined by plating the suspension on MacConkey or LB agar supplemented or not with antibiotics.

By

The bacterial load was determined by plating the suspension on MacConkey or LB agar supplemented or not with antibiotics. In competitive experiments with 2 strains (WT and mutant), the load of the mutant

6was directly determined on antibiotic plate. In contrast, the load of the WT strain was calculated from the total amount of bacteria on antibiotic-free plate ($c.f.u \text{ WT} = c.f.u \text{ Total} - c.f.u \text{ mutant}$). The load of every single mutant strain was then normalized to the inoculum and used to calculate the normalized competitive index (C.I) in order to compare their fitness to the WT. The C.I of a WT strain and respective mutant in a competitive experiment was determined as a ratio between $c.f.u$ (mutant) and $c.f.u$ (WT) divided by the ratio of both strains in the inoculum).

We have also added the antibiotic resistance of each strain in Table S3

4. Extended Fig. 5D, the y-axis label is not visible.

Response: We have increased the size of both axis labels.

Reviewer #3

1. In the bottom of pg 18 the authors say: “: The data presented on Figure 3B showed that the WT E. coli 8178 strain outperforms the isogenic lacZ mutant by 10-fold in C57BL/6 mice. This observation, regardless of the effect of lactose supplementation, is sufficient to state that E. coli 8178 utilizes lactose for proficient growth in the murine gut of C57BL/6 mice.”

I still don't think this is technically correct, as the authors don't measure the lactose metabolism in the gut, they cannot state this. What they can say is that lacZ mutant has a growth defect in the gut, which is not the same as saying that E. coli relies on lactose metabolism to colonize the gut.

Other reasons besides lactose metabolism, could justify the differences between the lacZ mutant and the wt. For e.g.:

- 1- The permease from the lacZ mutant is still active and therefore lactose (or other molecule) is still being transported by the cell, and this is costly.
- 2- The lac operon can be used to consume other carbon sources, such as galactosyl-glycerol, a naturally occurring galactoside DOI: 10.1093/genetics/139.1.19, possibly present in the mouse food.

Response: We agree with the referee that, without additional evidence, claiming that *E. coli* 8178 utilizes lactose *in vivo* is premature. We have revised and toned down our statement. Specifically, we have replaced:

Similar to our findings in 129S6/SvEvTac animals, E. coli 8178 reached a high level of gut colonization ($\sim 10^9$ - 10^{10} c.f.u/g feces), and relied on lactose metabolism in C57BL/6 mice.

By

*Similar to our findings in 129S6/SvEvTac animals, E. coli 8178 reached a high level of gut colonization ($\sim 10^9$ - 10^{10} c.f.u/g feces), with the *lacZ* mutant exhibiting a fitness defect (Fig. 3B; black symbols; Extended Data Fig. 3A).*

and

Combined, these observations demonstrate that *E. coli* 8178's reliance on lactose metabolism varies in a microbiota-dependent manner.

By

Combined, these observations demonstrate that *E. coli* 8178's reliance on the *lacZYA* operon varies in a microbiota-dependent manner.

2. The authors state by the end of pg. 19 that "any potential bacterial contaminants would have been eliminated by the antibiotic.", this is not true as streptomycin is unable to eliminate many of the gut colonizers.

Response: As we have mentioned in the previous rebuttal version, the experimental setup we used to maintain our germ-free mice has proven to be highly effective in preventing contamination. Although the streptomycin-pretreatment may not be optimal, it supports the same conclusion, indicating that the competitive index of the *lacZ* mutant in our germ-free animal is biologically relevant. As an alternative to a contamination issue, we believe that the divergent observation in germ-free animals between our study and the one published in Pinto et al., can instead be attributed to the different *E. coli* strains used.

Final Decision Letter:

Message: 4th July 2024

Dear Professor Hardt,

I am pleased to accept your Article "Non-canonical start codons confer context-dependent advantages in carbohydrate utilisation for commensal *E. coli* in the murine gut" for publication in Nature Microbiology. Thank you for having chosen to submit your work to us and many congratulations.

You may wish to make your media relations office aware of your accepted publication, in case they consider it appropriate to organize some internal or external publicity. Once your paper has been scheduled you will receive an email confirming the publication details. This is normally 3-4 working days in advance of publication. If you need additional notice of the date and time of publication, please let the production team know when you receive the proof of your article to ensure there is sufficient time to coordinate. Further information on

9our embargo policies can be found here:
<https://www.nature.com/authors/policies/embargo.html>

Please note that *Nature Microbiology* is a Transformative Journal (TJ). Authors may publish their research with us through the traditional subscription access route or make their paper immediately open access through payment of an article-processing charge (APC). Authors will not be required to make a final decision about access to their article until it has been accepted. Find out more about Transformative Journals

To assist our authors in disseminating their research to the broader community, our SharedIt initiative provides you with a unique shareable link that will allow anyone (with or

without a subscription) to read the published article. Recipients of the link with a subscription will also be able to download and print the PDF.

With kind regards,